# Transient loss of Polycomb components induces an epigenetic cancer fate

V. Parreno[1,9], V. Loubiere[1,2,9], B. Schuettengruber[1], L. Fritsch[1], C. C. Rawal[3], M. Erokhin[4], B. Győrffy[5,6], D. Normanno[1], M. Di Stefano[1], J. Moreaux[1,7,8], N. L. Butova[3], I. Chiolo[3], D. Chetverina[4], A.-M. Martinez[1✉] & G. Cavalli[1✉]

Although cancer initiation and progression are generally associated with the accumulation of somatic mutations[1,2], substantial epigenomic alterations underlie many aspects of tumorigenesis and cancer susceptibility[3–6], suggesting that genetic mechanisms might not be the only drivers of malignant transformation[7]. However, whether purely non-genetic mechanisms are sufficient to initiate tumorigenesis irrespective of mutations has been unknown. Here, we show that a transient perturbation of transcriptional silencing mediated by Polycomb group proteins is sufficient to induce an irreversible switch to a cancer cell fate in *Drosophila*. This is linked to the irreversible derepression of genes that can drive tumorigenesis, including members of the JAK–STAT signalling pathway and *zfh1*, the fly homologue of the *ZEB1* oncogene, whose aberrant activation is required for Polycomb perturbation-induced tumorigenesis. These data show that a reversible depletion of Polycomb proteins can induce cancer in the absence of driver mutations, suggesting that tumours can emerge through epigenetic dysregulation leading to inheritance of altered cell fates.

Genetic, epigenetic and environmental inputs are deeply intertwined, making it difficult to disentangle their respective contributions to cell fate decisions[8,9], and epigenetic reprogramming is a major contributor to tumour plasticity and adaptation[10,11]. Over recent decades, large-scale projects expanded the known repertoire of cancer-associated genetic mutations affecting epigenetic factors[12,13], including chromatin remodellers and modifiers, which regulate histone marks[14,15], DNA methylation[16], micro-RNAs[17] and 3D-genome folding[18], corroborating the role of epigenetic aberrations in the aetiology of haematological and solid malignancies[19,20]. Indeed, epigenetic modifications are used as biomarkers and are targeted by epi-drugs in cancer therapy[21]. Tumorigenesis is therefore associated with genetic as well as epigenetic determinants[22–25]. The fact that several hallmarks of human cancer[24,26] may be acquired through epigenome dysregulation suggests that epigenetic alterations play causal roles in cancer[4,27,28] and in metastatic progression[29–33]. In some paediatric cancers, such as posterior fossa ependymoma, low numbers of mutations were detected, consistent with the possibility that epigenetic changes may drive tumorigenesis[30]. These observations suggest that cancer is not solely a consequence of DNA mutations[34,35], but whether purely non-genetic reprogramming mechanisms are sufficient to initiate tumorigenesis remains an open question. Polycomb group (PcG) proteins are epigenetic factors forming two main classes of complexes called Polycomb Repressive Complex 1 and 2 (PRC1 and PRC2, respectively), which are highly conserved from fly to human and play a critical role in cellular memory by repressing developmental genes throughout development[36]. PcG dysregulation leads to cell fate changes[37], developmental transformations and is

associated with cancer[38]. PRC2 deposits the H3K27me3 repressive mark, whereas PRC1, which contains the PH, PC, PSC and the SCE subunits in flies, is responsible for H2AK118Ub deposition[36]. Contrasting with the redundancy found in mammals[36], most PcG components are encoded by a single gene in *Drosophila*, making this system more tractable for functional studies[39].

## Epigenetic perturbations initiate tumours

Null mutations or constant RNAi (RNA interference) knock-down (KD) targeting both *ph* homologues (*ph-p* and *ph-d*, which we refer to as *ph* for simplicity) can induce growth defects, loss of differentiation and cell overproliferation[40–43]. To test whether a transient epigenetic perturbation might initiate an irreversible change in cell fate, we set up a thermosensitive *ph*-RNAi system enabling the reversible KD of *ph* in the developing larval eye imaginal disc (ED) (Fig. 1a,b and Extended Data Fig. 1a–d). The PH protein is depleted in 24 h at 29 °C and is restored within 48 h of recovery at 18 °C (Extended Data Fig. 1e).

As expected, on constant PH depletion throughout development, 100% of EDs collected at the third larval stage (L3) are transformed into tumours (Fig. 1c,d and Methods), resulting in reduced viability (Extended Data Fig. 1f). A transient 24 h depletion of PH at the L1 stage, during which the ED starts developing, is also sufficient to trigger tumour formation in L3 EDs, characterized by overgrowth, loss of apico-basal cell polarity and of the ELAV differentiation marker (Fig. 1c–e and Extended Data Fig. 1g–i). These tumours show normal concentrations of PH protein in L3 EDs, both at day 9 (transient *ph*-KD d9) and day

[1]Institute of Human Genetics, CNRS, University of Montpellier, Montpellier, France. [2]Research Institute of Molecular Pathology, Vienna BioCenter, Vienna, Austria. [3]University of Southern California, Los Angeles, CA, USA. [4]Institute of Gene Biology, Russian Academy of Sciences, Moscow, Russia. [5]Semmelweis University Department of Bioinformatics, Budapest, Hungary. [6]Department of Biophysics, Medical School, University of Pécs, Pécs, Hungary. [7]Department of Biological Hematology, CHU Montpellier, Montpellier, France. [8]UFR Medicine, University of Montpellier, Montpellier, France. [9]These authors contributed equally: V. Parreno, V. Loubiere. ✉e-mail: anne-marie.martinez@igh.cnrs.fr; giacomo.cavalli@igh.cnrs.fr

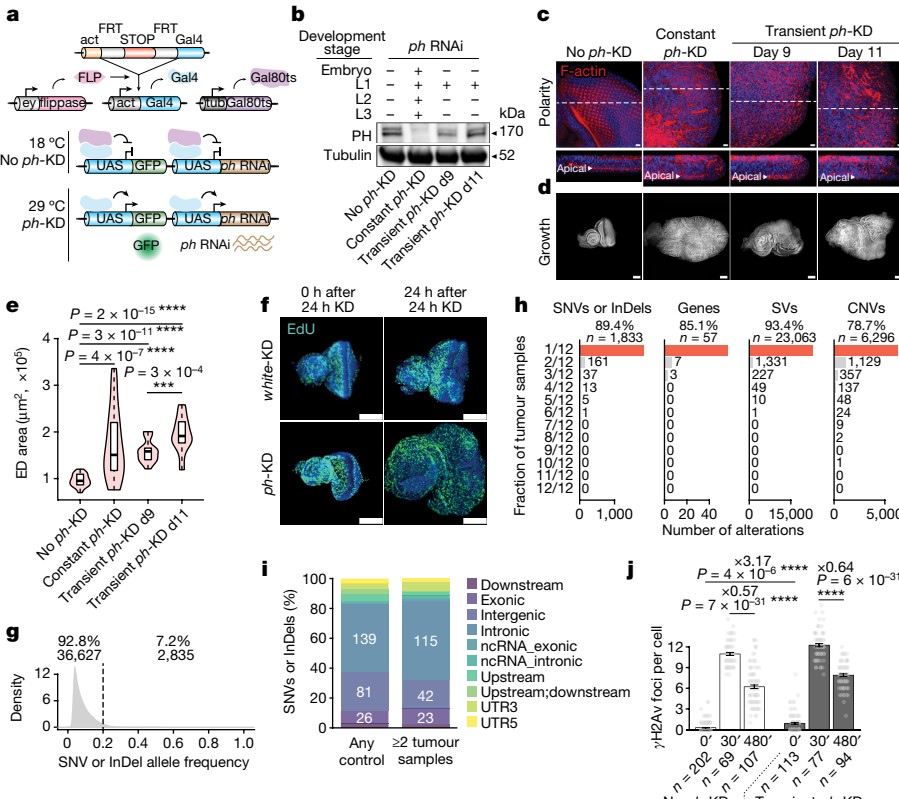

**Fig. 1 | Transient PRC1 depletion is sufficient to initiate tumours. a**, Scheme depicting the conditional *ph*-KD system (Methods). **b**, Western blot analysis of PH protein concentrations in the EDs of L3 larvae subjected to no *ph*-KD (control), constant or transient *ph*-KD at L1 stage. **c**, Representative confocal images of F-actin staining (red) showing a polarized epithelium with apical F-actin (*xz* cross-sections at the bottom) in no *ph*-KD (control, left), whereas polarity is disrupted on constant or transient *ph*-KD EDs (dissected at L3 stage). DNA is stained with DAPI (blue). **d**,**e**, DAPI staining (**d**) is used to measure ED areas (**e**) under no *ph*-KD (control), constant or transient *ph*-KD conditions (*n* = 30 EDs per condition; two-sided Wilcoxon test: ***$P < 1 \times 10^{-3}$, ****$P < 1 \times 10^{-5}$; box plots show the median (line), upper and lower quartiles (box) ±1.5× interquartile range (whiskers); outliers are not shown). **f**, EdU staining (green) imaged at 0 h (left) and 24 h (right) after 24 h of *w*-KD (control, top) or *ph*-KD (bottom). **g**, Distribution of somatic SNVs or InDel allele frequencies detected in all samples. **h**, Number of tumour samples in which each SNVs or InDels, gene with deleterious SNVs or InDels, structural variants (SVs) and CNVs were found. **i**, Feature distribution of SNVs or InDels found in any of the control samples (no *ph*-KD, left bar) or shared between at least two tumour samples (right bar). **j**, Number of γH2Av foci per cell before (0 min; indicated as 0′) and after (30 and 480 min, indicated as 30′ and 480′) exposure to 5 Gy irradiation in control (no *ph*-KD, left) or transient *ph*-KD EDs (right). Individual data points are shown in grey and bars correspond to the mean ± standard error (whiskers). Two-sided *t*-test ****$P < 1 \times 10^{-5}$. Scale bars, 10 μm (**c**), 100 μm (**d**,**f**).

11 (transient *ph*-KD d11) after egg laying (AEL) (Fig. 1b and Extended Data Fig. 1c,d). EDs continue to grow after PH recovery (Fig. 1e) and cannot differentiate (Extended Data Fig. 1i), suggesting that the tumour state is stable and maintained independently of its epigenetic trigger. Likewise, PH depletion at L2 or early L3 stage induces tumours (Extended Data Fig. 1g–i), suggesting that PRC1 is required throughout development to prevent tumorigenesis. Transient depletion of PSC-SU(Z)2, another core PRC1 subunit for which null mutations drive neoplastic transformation[44], is also sufficient to induce tumorigenesis (Extended Data Fig. 1j–m).

Transient PH depletion induces tumours with 100% penetrance within 2 days, as illustrated by the early L3 PH depletion experiment (Extended Data Fig. 1g–i). To assess whether such tumours may arise from a clonal subpopulation of cells, we performed EdU (5-ethynyl-2′-deoxyuridine) staining after 24 h *ph*-KD in early L3 EDs (Fig. 1f and Supplementary Videos 1 and 2). Aberrant replication was observed throughout the tissue within 24 h, indicating that most or all cells undergo malignant transformation. For DNA mutations to drive these tumours, they should simultaneously occur in many cells to trigger overproliferation in the whole tissue. Given the low frequency of deleterious mutations per cell generation (about 1.2 per genome[45]) and the limited number of genes that can act as cancer drivers in *Drosophila*[46], this scenario seemed unlikely. Nevertheless, we sequenced

whole cancer genomes by collecting eggs from several independent crosses of mated females and subjecting them to transient KD, constant KD or no *ph*-KD (control condition), before sequencing their genomic DNA (gDNA). In total, we sequenced four independent control samples as well as 12 independent tumour samples (Methods). When using batch-matched control tissues (no *ph*-KD) to identify single nucleotide variants (SNV) or small insertions and deletions (InDels)[46], we found that 68.1% of the identified variants are present in only one of the samples and that 7 out of 12 tumour samples contained fewer SNVs or InDels than at least one of the control samples (Extended Data Fig. 2a), ruling out that PH depletion induces a massive increase in mutation rates and consistent with previous data[47]. Moreover, 92.8% of the identified SNVs or InDels had an allele frequency below 0.2, precluding them from driving whole-tissue tumours (Fig. 1g). Regarding SNVs or InDels with an allele frequency higher than 0.2, none of them was shared among the 12 tumour samples (Fig. 1h). Instead, 89% were found in only one sample and the 217 variants shared between at least two tumours had similar feature distributions compared to the variants found in control samples, without bias towards exons (Fig. 1i). No genes contained deleterious SNVs or InDels in all tumour samples, and similar results were found when considering structural variants or copy number variations (CNVs) (Fig. 1h and Methods). Together, these results argue strongly against the presence of recurrent driver mutations in these tumours.

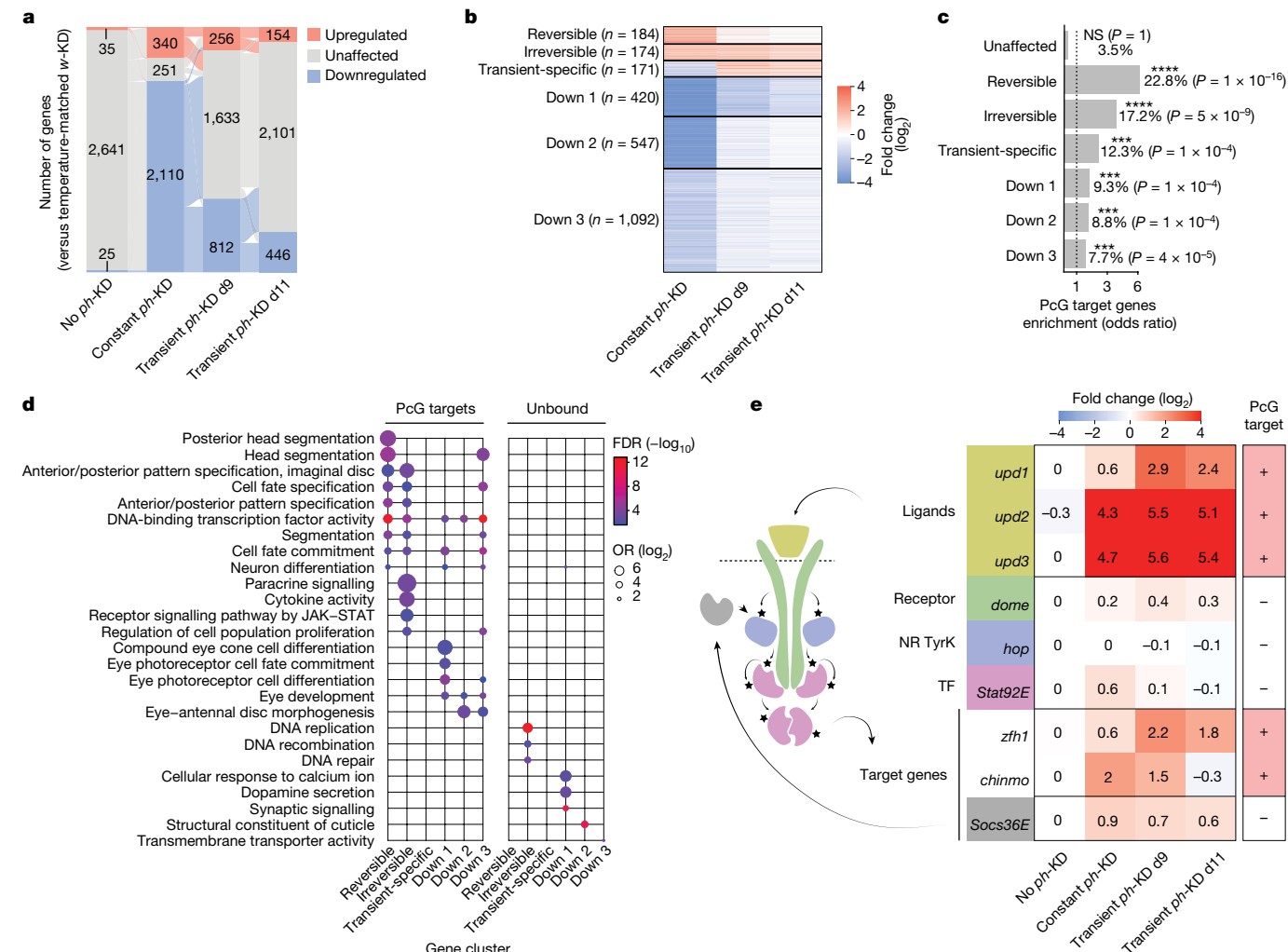

**Fig. 2 | EICs show irreversible transcriptional changes. a,** Alluvial plot showing differentially expressed genes after no *ph*-KD (control), constant and transient *ph*-KD. Transitions between upregulated (orange), unaffected (grey) and downregulated (blue) states are indicated by thin lines of the same respective colours. **b,** Clustering of differentially expressed genes after constant or transient *ph*-KD. **c,** Over-representation of direct PcG target genes (defined as more than or equal to 50% of the gene body overlapping a H3K27me3 repressive domain in control condition). One-sided Fisher's exact test *P* values were corrected for multiple testing using FDR: ***FDR < 1 × 10⁻³, ****FDR < 1 × 10⁻⁵; NS, *P* > 0.05. **d,** Representative Gene Ontology terms enriched for each gene cluster, further stratified as being direct PcG targets (left) or not (right). The full chart is available in Extended Data Fig. 3d. **e,** Transcriptional fold changes of genes involved in the JAK–STAT signalling pathway on *ph*-KD. Direct PcG targets (+) are indicated in the right column.

To test whether transient PH depletion could induce genome instability, we counted the number of phospho-H2AvD foci (γH2Av) per cell in control (no *ph*-KD) and transient *ph*-KD tumours before and during a time course after irradiation. Despite a slightly higher number of foci before irradiation, probably due to the higher fraction of cells engaged in DNA replication, tumour and control samples showed a similar decrease in the number of γH2Av foci between 30 and 480 minutes after irradiation (Fig. 1j and Extended Data Fig. 2b,c), suggesting that these tumours can efficiently repair DNA breaks to prevent the accumulation of mutations. Finally, karyotype analysis of the tumours collected on transient *ph*-KD did not show significant differences in chromosomal rearrangements compared to control samples (Extended Data Fig. 2d).

In summary, transient depletion of PRC1 components is sufficient to switch cells into a neoplastic state that is maintained even after normal PcG protein concentrations are re-established. As the same genotype can generate both a normal phenotype or a tumour depending on a transient gene regulatory modification in the absence of DNA driver mutations, we defined these tumours as epigenetically initiated cancers (EICs).

## JAK–STAT signalling activation in EICs

We compared the transcriptomes of the control condition (no *ph*-KD), transient and constant *ph*-KD tumours to temperature-matched controls, generated with a similar RNAi system targeting the *white* (*w*) gene, which is dispensable for normal eye development (differential transcriptome analyses are available in Supplementary Table 1). As expected, the *ph*-RNAi and the *w*-RNAi lines are hardly distinguishable at 18 °C, as well as in the transient *w*-KD condition (Fig. 2a and Extended Data Fig. 3a). Consistent with our previous work[41,42], constant *ph*-KD is associated with the upregulation of 340 genes—including canonical PcG targets such as Hox and developmental transcription factor genes—and the down-regulation of 2,110 genes, including most key regulators of ED development (Fig. 2a and Extended Data Fig. 3b). Only a subset of these genes was also differentially expressed in transient *ph*-KD at d9 AEL (256 and 812, respectively), and even less at later d11 AEL (154 and 446, respectively), suggesting a progressive yet incomplete rescue of the transcriptome (Fig. 2a and Extended Data Fig. 3a–c). Therefore, most (75%) of the transcriptional defects observed on constant *ph*-KD can be restored on reinstating normal levels of PH.

Hierarchical clustering of differentially expressed genes identified three clusters that are upregulated in at least one condition, and three downregulated clusters (Fig. 2b; clustering results available in Supplementary Table 2 and Methods). The upregulated clusters show stronger and significant over-representation of PcG target genes covered with H3K27me3 in control EDs (Fig. 2c). This suggests that their upregulation is a direct consequence of compromised PcG repression, although they retain distinct patterns. The 'reversible' cluster includes canonical PcG target genes such as *en*, *eve*, *Ubx* and *Scr*, that are upregulated on constant *ph*-KD but recover control levels of expression after transient *ph*-KD, precluding them from being required for the maintenance of EICs (Fig. 2b and Extended Data Fig. 3b). The same is true for 'transient-specific' genes, whose upregulation is dispensable for tumour growth after constant *ph*-KD.

The 'irreversible' cluster is of particular interest, as it contains a high fraction of PcG target genes that remain upregulated despite PH restoration and therefore represents candidate genes involved in the development of EICs (Fig. 2b,c). Whereas PcG target genes from the reversible and irreversible clusters share ontologies associated with developmental transcription factors, irreversible genes show specific enrichments for paracrine signalling and cytokine activity (Fig. 2d and Extended Data Fig. 3d), including the JAK–STAT ligands (*upd1*, *upd2*, *upd3*), which were shown to be associated with various tumours, including those depending on PcG mutations[43,44,48] (Fig. 2e). In addition, *chinmo* and *zfh1* are direct PcG targets that have been described to act downstream of the JAK–STAT pathway[49] and are accordingly upregulated on PH depletion (Fig. 2e). The transcriptional repressor ZFH1 is of particular interest, because it remains upregulated at d11 AEL, is known to be involved in self-renewal and tumour growth[50–52] and is conserved in mammals, in which its homologue ZEB1 can induce epithelial-to-mesenchymal transition[53]. Consistent with its transcriptional upregulation, ZFH1 protein is increased on constant PH depletion and even more on transient PH depletion (Extended Data Fig. 3e,f), suggesting that it might support the development of EICs.

Finally, we noted that irreversible genes that are not PcG targets are enriched for Gene Ontology (GO) terms related to DNA replication and repair (Fig. 2d and Extended Data Fig. 3d), suggesting that their upregulation may be a consequence of the proliferation of tumour cells. Together, these results indicate that EICs are driven by a restricted set of irreversibly upregulated genes, including major members of the JAK–STAT signalling pathway, rather than by the vast pleiotropic dysregulation of cancer genes that is observed on constant PH depletion. Therefore, we sought to investigate why this subset of genes remains irreversibly upregulated after restoration of normal PH levels and to test whether they are required for the development of EICs. For simplicity, unless explicitly stated, further investigations of transient *ph*-KD EDs were conducted on tissues collected at d11 AEL after a 24 h KD at the L1 stage, representing the condition with the smallest number of differentially expressed genes.

## Chromatin analysis at irreversible genes

To identify their unique chromatin features, we focused on irreversible (*n* = 30) and reversible (*n* = 42) genes that are direct PcG targets and are covered with the H3K27me3 repressive mark in control EDs (for a full list of PcG target genes, see Supplementary Table 2). Both groups show similar H3K27me3 levels in control tissues (Extended Data Fig. 4a), where they are transcribed at similarly low levels (Fig. 3a). They are also induced at comparable levels on constant *ph*-KD, ruling out the possibility that weaker PcG repression and/or higher transcriptional levels are the reason for irreversible genes being unable to recover normal transcription after transient *ph*-KD (Fig. 3a).

We then explored the possibility that chromatin might not be correctly re-established at irreversible genes in EICs, by performing chromatin immunoprecipitation combined with sequencing (ChIP–seq) for

PH and CUT&RUN for several histone marks after no *ph*-KD (control), constant and transient *ph*-KD. Whereas most reversible and irreversible genes lost the H3K27me3 repressive mark on constant *ph*-KD, H3K27me3 domains were notably recovered after transient *ph*-KD (Fig. 3b,d). Most H3K27me3 domains and overlapping PH peaks are erased on constant PH depletion, but are recovered after transient depletion (Extended Data Fig. 4b). The same applies to the H2AK118Ub repressive mark deposited by PRC1 (Fig. 3d and Extended Data Fig. 4b). H3K27me3 loss on constant *ph*-KD is accompanied by a reciprocal gain of H3K27Ac peaks, its activating counterpart, at both reversible and irreversible genes (Fig. 3c,d). Nevertheless, both groups show similar H3K27me3 and H3K27Ac levels after transient *ph*-KD, suggesting that comparable chromatin landscapes may promote distinct transcriptional outcomes (Extended Data Fig. 4a). Inspection of individual loci showed that recovery of chromatin composition is similar at the level of reversible and irreversible genes, as evidenced by the *upd* locus, which does not contain H3K27Ac peaks after transient *ph*-KD although it is irreversibly upregulated (Figs. 2e and 3d).

Nevertheless, we noted some exceptions, such as the *zhf1* gene that retains low but significantly higher levels of H3K27Ac compared to control tissues on transient depletion of PH (Fig. 3d), suggesting that a fraction of irreversible loci might retain small quantitative differences. Differential analyses indicated that most H3K27me3 domains showed a steep decrease on constant *ph*-KD but overall recovered to normal levels under transient conditions (Fig. 3e). Similar trends were found at H3K27Ac peaks and H2AK118Ub domains, whereby transient *ph*-KD showed weaker and fewer significant differences compared to constant *ph*-KD (Fig. 3f and Extended Data Fig. 4c, respectively). This approach again identified the *zfh1* locus as an outlier showing significantly increased H3K27Ac peaks after transient *ph*-KD (Fig. 3f). To precisely assess whether small differences in terms of H3K27me3 or H3K27Ac fold changes would be predictive of irreversible transcriptional changes, we classified H3K27me3 domains based on whether they contain irreversible or reversible genes and interestingly found that genes from the two groups are usually found in different domains (Extended Data Fig. 4d). Domains overlapping irreversible versus reversible genes showed small differences in H3K27me3 or H3K27Ac fold changes (Fig. 3g,h), which are unlikely to explain the clear-cut difference between reversible and irreversible genes. Therefore, irreversible transcriptional changes drive tumorigenesis despite the re-establishment of an essentially normal chromatin landscape at PcG target genes.

## Heritable chromatin accessibility changes

The analysis of PH binding levels at PH peaks located ±25 kb from the transcription start sites (TSS) of reversible (*n* = 113) or irreversible (*n* = 91) genes revealed no significant differences either in control EDs (no *ph*-KD) or after transient *ph*-KD (Extended Data Fig. 4e). This is consistent with the levels of H3K27me3 and H2AK118Ub repressive marks, which are also similar (Extended Data Fig. 4a). We therefore wondered whether the irreversible transcriptional changes found in EICs might be due to the binding of specific transcription factors to specific chromatin targets on *ph*-KD, preventing re-repression on restoration of PH. In this scenario, one would expect the opening of specific sites at irreversible gene loci. To test this hypothesis, we performed ATAC-Seq in control EDs (no *ph*-KD) or after constant or transient *ph*-KD, and found 1,220 reversible peaks showing a stark increase in accessibility after constant PH depletion but returning to normal levels after transient KD (Fig. 4a). By contrast, 446 ATAC-Seq peaks increased accessibility both on constant as well as on transient PH depletion (Fig. 4a). We named these ATAC-Seq regions irreversible peaks (clusters are fully available in Supplementary Table 3).

To assess whether reversible and irreversible peaks correlate with transcriptional changes, we assigned them to the closest TSS (±25 kb, Methods). Reversible and irreversible ATAC-Seq peaks were

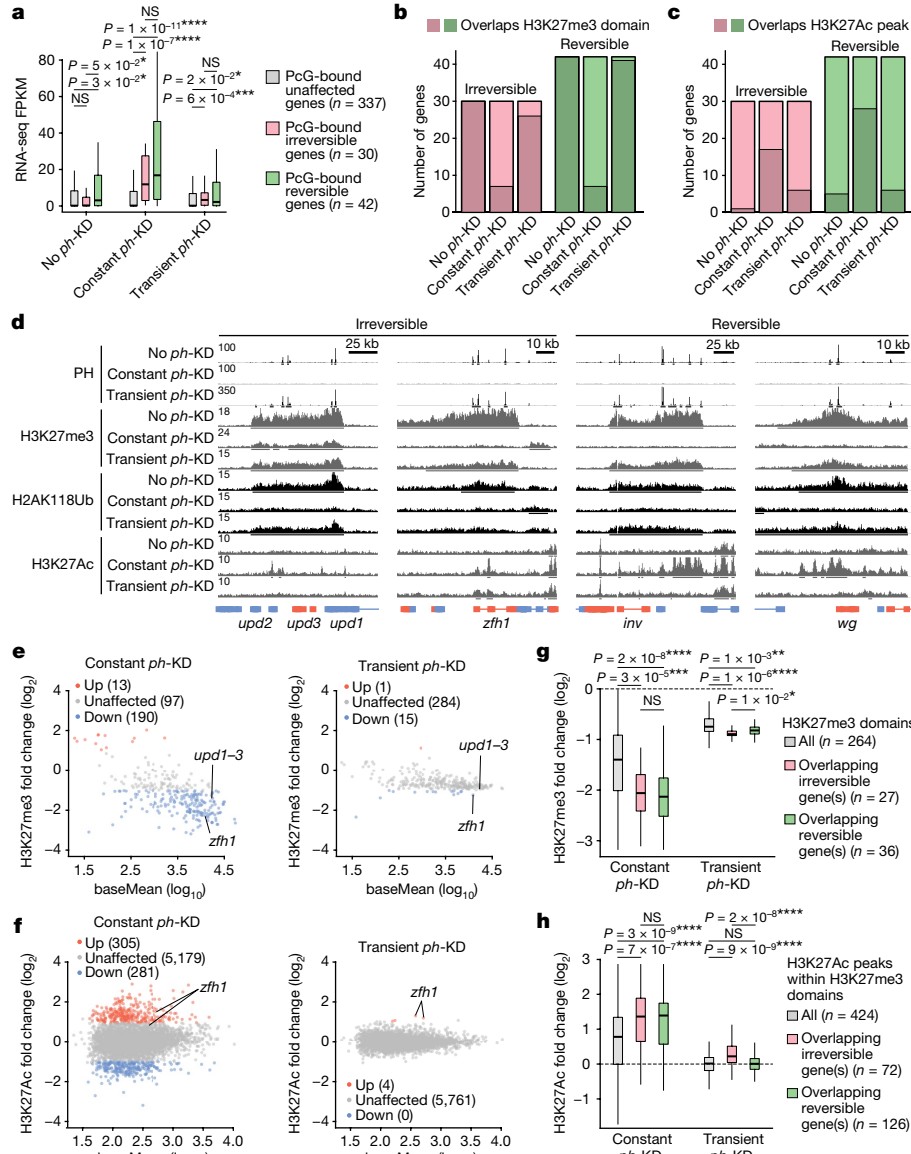

**Fig. 3 | PcG repressive landscape is restored after transient *ph*-KD.**
**a**, Fragments per kilobase of transcript per million mapped reads (FPKM) of irreversible (pink), reversible (green) and unaffected (grey) genes that are direct PcG targets. Two-sided Wilcoxon test: $*P < 5 \times 10^{-2}$, $***P < 1 \times 10^{-3}$, $****P < 1 \times 10^{-5}$, NS, $P > 0.05$. Box plots show the median (line), upper and lower quartiles (box) ±1.5× interquartile range (whiskers), outliers are not shown. **b**, Number of irreversible (pink) and reversible (green) genes overlapping an H3K27me3 domain (more than or equal to 50% of the gene body) after no *ph*-KD (control), constant or transient *ph*-KD. **c**, Number of irreversible (pink) and reversible (green) genes overlapping at least one H3K27Ac peak (in the gene body or up to 2.5 kb upstream of the TSS) after no *ph*-KD (control), constant or transient *ph*-KD. **d**, Screenshot of PH ChIP–seq, H3K27me3, H2AK118Ub and H3K27Ac CUT&RUNs tracks at representative irreversible (left) or reversible (right) loci under the indicated conditions (left). **e**,**f**, For H3K27me3 domains (**e**) and H3K27Ac peaks (**f**), fold changes are shown as a function of their average-normalized counts across all samples (baseMean) for constant (left) or transient (right) *ph*-KD conditions. Significant changes are highlighted using a colour code (colour legend). **g**, The H3K27me3 fold changes (between constant or transient *ph*-KD and no *ph*-KD conditions) at H3K27me3 domains that are found in the control sample (no *ph*-KD) and overlap irreversible (pink) or reversible (green) genes. All H3K27me3 domains are shown for reference (grey). Two-sided Wilcoxon test: $*P < 5 \times 10^{-2}$, $**P < 1 \times 10^{-2}$, $***P < 1 \times 10^{-3}$, $****P < 1 \times 10^{-5}$, NS, $P > 0.05$. Box plots show the median (line), upper and lower quartiles (box) ±1.5× interquartile range (whiskers), outliers are not shown. **h**, The H3K27Ac fold changes at H3K27Ac peaks overlapping the H3K27me3 domains found in control sample (no *ph*-KD) and overlapping the irreversible (pink) or reversible (green) genes. All H3K27Ac peaks overlapping control H3K27me3 domains are shown for reference (grey). Two-sided Wilcoxon test: $****P < 1 \times 10^{-5}$, NS. $P > 0.05$. Box plots show the median (line), upper and lower quartiles (box) ±1.5× interquartile range (whiskers), outliers are not shown.

significantly associated with the reversible and irreversible genes identified by RNA sequencing (RNA-seq) analysis in Fig. 2b, respectively (Fig. 4b). This suggests that a substantial fraction of these peaks might correspond to enhancer elements that activate the transcription of cognate TSSs from a distance. Consistently, roughly 70% of reversible and irreversible peaks are found more than 1 kb away from the closest TSS (Fig. 4c,d). For example, the *upd3* gene is irreversibly upregulated after transient *ph*-KD and is surrounded by several promoter-distal irreversible ATAC-Seq peaks, whereas the reversible gene *Ubx* shows reversible ATAC-Seq peaks that can be observed only on constant *ph*-KD (Fig. 4d). In parallel, 604 peaks show reduced accessibility and are associated with downregulated genes (Figs. 2b and 4a,b).

To understand which transcription factors might cause these differences in accessibility, we searched for DNA binding motifs in ATAC-Seq

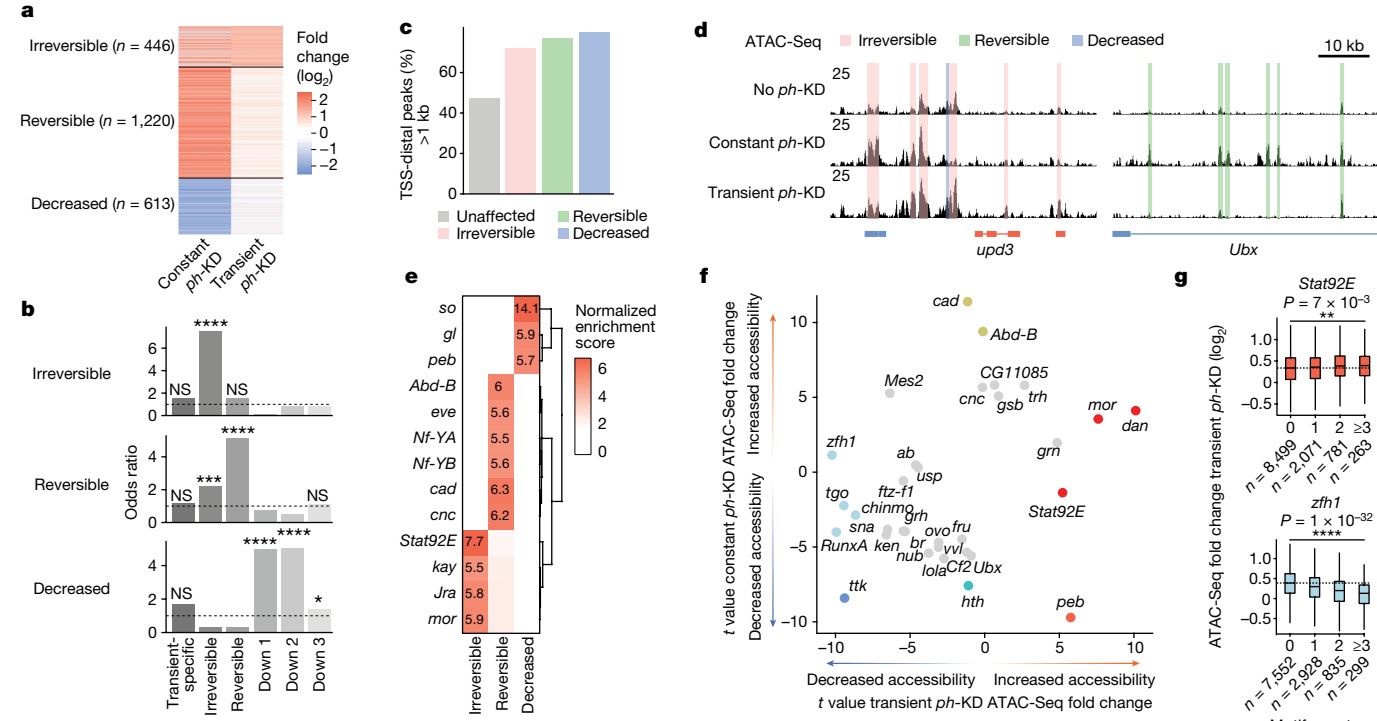

**Fig. 4 | Chromatin accessibility changes underlie reversible and irreversible transcriptional changes. a**, Clustering of ATAC-Seq peaks showing significant changes after constant or transient *ph*-KD. **b**, Over-representation of genes associated with irreversible (top), reversible (middle) or decreased (bottom) ATAC-Seq peaks, for each of the six RNA-seq clusters defined in Fig. 2b. One-sided Fisher's exact test *P* values were corrected for multiple testing using FDR: *FDR < $5 \times 10^{-2}$, ***FDR < $1 \times 10^{-3}$, ****FDR < $1 \times 10^{-5}$, NS, *P* > 0.05. Exact FDR values: $2 \times 10^{-1}$, $4 \times 10^{-23}$, $1 \times 10^{-1}$, $1 \times 10^{0}$, $1 \times 10^{0}$, $1 \times 10^{0}$ (irreversible); $4 \times 10^{-1}$, $2 \times 10^{-5}$, $2 \times 10^{-34}$, $1 \times 10^{0}$, $3 \times 10^{-1}$ (reversible); $8 \times 10^{-2}$, $1 \times 10^{0}$, $1 \times 10^{0}$, $2 \times 10^{-21}$, $5 \times 10^{-32}$, $1 \times 10^{-2}$ (decreased). **c**, Fraction of TSS-distal peaks per cluster (greater than 1 kb). **d**, Screenshot of ATAC-Seq tracks after no *ph*-KD (control, top), constant (middle) or transient (bottom) *ph*-KD, at the irreversibly upregulated

*upd3* gene (left) and the reversibly upregulated *Ubx* gene (right). **e**, Normalized enrichment scores of DNA binding motifs found at each cluster of ATAC-Seq peaks (±250 bp, *x* axis). **f**, Linear model *t* values of DNA binding motifs associated with increased (positive *t* values) or decreased (negative *t* values) accessibility after transient (*x* axis) or constant *ph*-KD (*y* axis). Only motifs with a significant *P* < $1 \times 10^{-5}$ in at least one of the two linear models are shown. **g**, Fold changes at ATAC-Seq peaks (*y* axis) on transient *ph*-KD, as a function of the number of *Stat92E* (left, in orange) or *zfh1* (right, in blue) motifs that they contain (*x* axis). Two-sided Wilcoxon test: **$P < 1 \times 10^{-2}$, ****$P < 1 \times 10^{-5}$. Box plots show the median (line), upper and lower quartiles (box) ±1.5× interquartile range (whiskers), outliers are not shown.

peaks. Reversible and irreversible peaks show distinct motif signatures (Fig. 4e). Reversible peaks are enriched for *Abd-B*, *cad* and *eve* motifs, three different PcG canonical targets involved in antero-posterior patterning that are strongly upregulated after constant *ph*-KD compared to transient *ph*-KD (Extended Data Fig. 3b). By contrast, irreversible peaks are enriched for *Jra* and *kay* motifs, the *Drosophila* homologues of AP-1, which are the main transcription factors of the oncogenic JNK signalling pathway[54]. Furthermore, they were strongly and specifically enriched for *Stat92E* motifs, the key effector of the JAK–STAT pathway[55]. Finally, decreased peaks are enriched in *glass* (*gl*) and *sine oculis* (*so*) motifs, two key regulators of eye development that are irreversibly downregulated (the down 1 cluster in Fig. 2b and Extended Data Fig. 3b). This latter point indicates that the activation of the retinal determination gene network is compromised in the absence of PcG, consistent with our previous work[42].

These results indicate that the *Abd-B*, *cad* and *eve* genes are responsible for the pleiotropic transcriptional defects observed on constant PH depletion, but are unlikely to be required for the progression of EICs. On the other hand, recruitment of AP-1 and STAT92E at irreversible peaks could maintain irreversible genes in an active state, potentially by maintaining open chromatin at their *cis*-regulatory regions. To tackle this latter point, we sought to predict ATAC-Seq changes using transcription factor motif counts (Methods). *cad* and *Abd-B* motifs are associated with increased accessibility after

constant PH depletion but not in a transient condition (Fig. 4f and Extended Data Fig. 5a), suggesting that their effect on chromatin and transcription is dispensable for the growth of EICs. Conversely, STAT92E and ZFH1 motifs were among the best predictors of increased and decreased accessibility after transient *ph*-KD, respectively (Fig. 4f,g).

## Tumorigenesis requires STAT92E and ZFH1

To assess whether the STAT92E activator and the ZFH1 repressor are necessary for the development of EICs, we set up dual RNAi systems allowing the depletion of each of the two factors in combination with *white* or *ph*. As a control, we combined *gfp* (green fluorescent protein) and *white*-RNAi (*gfp* + *w*-KD), which had no impact on ED growth or differentiation, whereas *gfp*+*ph*-KD induced tumours as expected (Fig. 5a and Extended Data Fig. 5b). Both on constant and on transient depletion, *Stat92E* and *zfh1*-KD alone had no visible effect. However, when combined with *ph*-KD, they both significantly reduced *ph*-dependent tumour growth and partially restored cell polarity and photoreceptor differentiation (Fig. 5a and Extended Data Fig. 5b–f), indicating that they are both bona fide drivers of the tumour phenotype. These rescues are also associated with an overall rescue of constant *gfp* + *ph*-KD transcriptomes, with 50% of differentially expressed genes returning to control levels on *gfp*+*zfh1*-KD (Fig. 5b). Consistent with previous

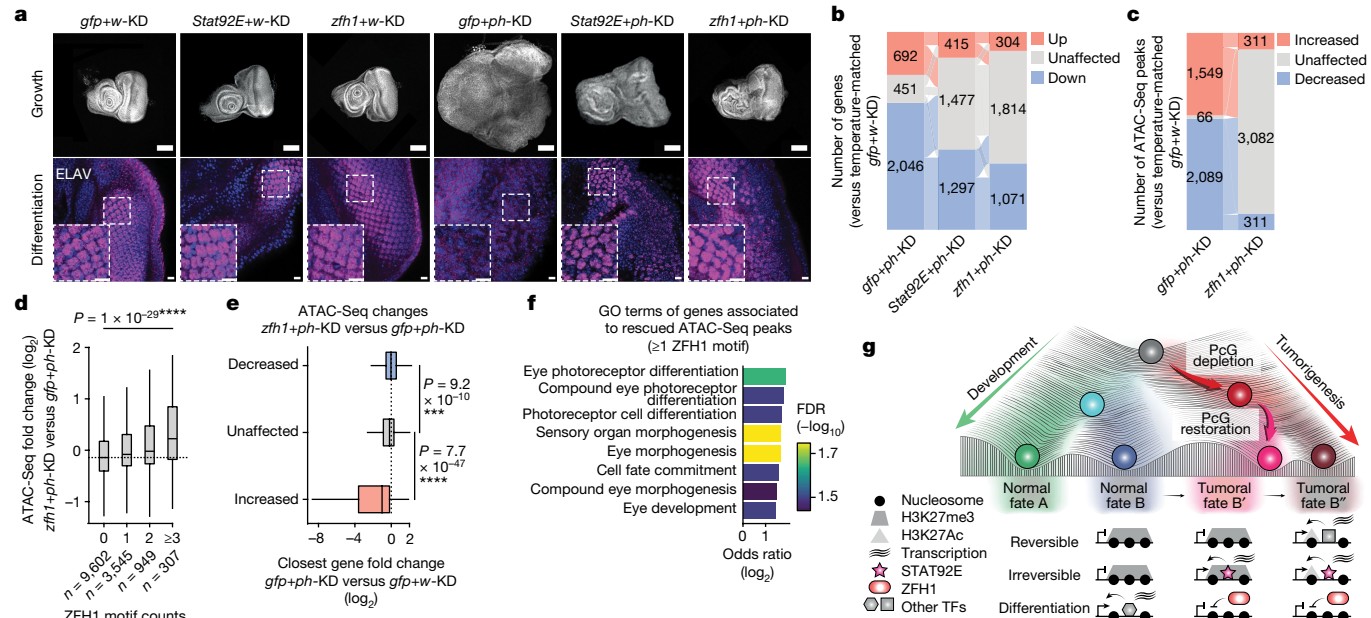

**Fig. 5 | Tumour development requires STAT92E and ZFH1. a**, DAPI (top, in grey) and neuronal differentiation marker ELAV (bottom, in magenta) stainings of EDs after constant KD of the following components: *gfp+w*, *Stat92E+w*, *zfh1+w*, *gfp+ph*, *Stat92E+ph* and *zfh1+ph* (top labels). Two independent biological replicates were performed with similar results. Scale bars: 100 μm (DAPI), 10 μm (ELAV). **b**, Number of differentially expressed genes after *gfp+ph*-KD (tumours), *Stat92E+ph*-KD and *zfh1+ph*-KD. Transitions between upregulated (orange), unaffected (grey) and downregulated (blue) states are indicated by thin lines of the same respective colours. **c**, Number of ATAC-Seq peaks showing significant accessibility changes after *gfp+ph*-KD or *zfh1+ph*-KD. Transitions between increased (orange), unaffected (grey) and decreased (blue) states are indicated by thin lines of the same respective colours. **d**, Fold changes at ATAC-Seq peaks between *zfh1+ph*-KD and *gfp+ph*-KD, depending on the number of ZFH1 motifs they contain (*x* axis). Two-sided Wilcoxon test,

$****P < 1 \times 10^{-5}$. Box plots show the median (line), upper and lower quartiles (box) ±1.5× interquartile range (whiskers), outliers are not shown. **e**, RNA-seq fold changes on *gfp+ph*-KD (*x* axis) of genes associated with ATAC-Seq peaks that are decreased (in blue), unaffected (in grey) or increased (in orange) after *zfh1+ph*-KD compared to *gfp+ph*-KD (*y* axis). Two-sided Wilcoxon test: $****P < 1 \times 10^{-5}$. Box plots show the median (line), upper and lower quartiles (box) ±1.5× interquartile range (whiskers), outliers are not shown. **f**, Top enriched Gene Ontology (GO) terms for genes associated with ATAC-Seq peaks containing at least one ZFH1 motif and showing significantly increased accessibility after *zfh1+ph*-KD compared to *gfp+ph*-KD. **g**, Schematic illustration showing that PcG depletion triggers an epigenetic switch to a cancer fate. Resulting cancers persist after the PcG protein is restored, and their maintenance is associated with stable transcriptional changes supported by the STAT92E activator and the ZFH1 repressor.

studies showing that *zfh1* is a target of STAT92E (ref. 50), the *zfh1* gene returned to control levels in *Stat92E+ph-KD* (differential analyses are available in Supplementary Table 4). Thus, ZFH1 seems to play a master role in shaping the tumour transcriptome.

Therefore, we sought to investigate its impact on chromatin by performing comparative ATAC-Seq experiments in *gfp+ph*-KD, *gfp+zfh1*-KD and *zfh1+ph*-KD. Consistent with our previous result showing that *zfh1* motifs are associated with decreased accessibility in tumours compared to control tissues (Fig. 4g), *zfh1*-KD in combination with *ph*-KD was found to be associated with the reopening of roughly 1,700 peaks showing decreased accessibility in *gfp+ph*-KD tumours (Fig. 5c). Moreover, *zfh1* motif counts are predictive of an increase in ATAC-Seq signal between *gfp+ph*-KD and *zfh1+ph*-KD tissues (Fig. 5d). These results indicate that *zfh1* represses transcription by reducing the accessibility of a subset of regulatory elements. Thus, we classified ATAC-Seq peaks based on their fold change between *gfp+ph*-KD and *zfh1+ph*-KD and assigned them to the closest TSS (±25 kb). Peaks with increased accessibility on *zfh1+ph*-KD were associated with genes that were aberrantly downregulated on *gfp+ph*-KD (Fig. 5e) and are involved in eye development and differentiation (Fig. 5f), reminiscent of the genes identified in the Down 1 RNA-seq cluster (Fig. 2b).

Altogether, these results indicate a multistep model (Fig. 5g) in which transient disruption of PcG-mediated silencing irreversibly activates the JAK–STAT pathway, which induces cell proliferation as well as the *zfh1* gene. In turn, ZFH1 represses genes required for ED development, thereby preventing cell differentiation in EICs.

## EICs are autonomous immortal tumours

Most EIC-bearing larvae die after day 11 AEL, preventing the study of tumour development over time. To circumvent this limitation, allografts of imaginal disc tissue into the abdomen of adult *Drosophila* hosts are commonly used to assess the tumorigenic potential of a tissue, and we previously showed that *ph* mutant EDs continuously grow until they eventually kill the host[43]. To be able to track transplanted EICs, we developed a variant of our thermosensitive system that constitutively expresses GFP in the eye, whereas an upstream activation sequence-red fluorescent protein (UAS-RFP) cassette can be used as a reporter of continuing *ph*-KD (Extended Data Fig. 6a,f–i). This system induces EICs with similar penetrance, morphological and transcriptional defects, showing that EICs can be obtained in different genetic backgrounds (Extended Data Fig. 6b–e). The differential analyses of the corresponding transcriptomes are available in Supplementary Table 5. We then performed allografts using this line (Extended Data Fig. 7), keeping host flies at a restrictive temperature after transplant (18 °C) to preclude activation of *ph*-RNAi in transplanted tissues.

Constant *ph*-KD primary tumours grew in a high fraction of the injected host flies within 20 days of transplantation (Extended Data Fig. 7a–c). Transient *ph*-KD primary EICs behaved similarly, indicating that their overgrowth results from an autonomous, stably acquired state (Extended Data Fig. 7a–c). To measure tumour growth over time, we set up a scheme allowing us to trace the tumour of origin (Extended Data Fig. 7d). Tumours derived from both constant or transient PH

depletion maintained their ability to expand in host flies more than ten rounds of transplantation. Tumour growth penetrance, defined as the percentage of host flies bearing GFP-positive cells 20 days after transplantation, increased over generations of transplantation (Extended Data Fig. 7b), whereas the survival of host flies decreased (Extended Data Fig. 7c,e,f). Furthermore, tumours metastasized to regions and organs far from the injection site, with increasing penetrance with the number of transplants (Extended Data Fig. 7g,h). Finally, allografts originating from tissues injected after a transient *ph*-KD at the late L3 stage also gave rise to tumours of increasing penetrance over the number of transplantations (Extended Data Fig. 7i,j).

Together, these results indicate that the tumorigenic potential of EICs is maintained autonomously, increases over time and can propagate months after *ph*-RNAi has been removed. This progression might suggest that EICs acquire secondary modifications, either epigenetic or genetic, that increase their aggressiveness over time.

## Discussion

It is difficult to discriminate among genetic, environmental and cell-intrinsic epigenetic contributions to tumorigenesis[33]. The system described here shows that on transient depletion of PRC1 subunits cells undergo neoplastic transformation (Fig. 5g and Extended Data Fig. 8), associated with the irreversible activation of genes including key JAK–STAT pathway members that sustain cell growth, proliferation, loss of cell polarity, cell migration and cytokine activity. One main difference between these irreversibly activated genes and reversible PcG target genes is the presence of different sets of transcription factor binding motifs in their vicinity. We posit that, even if PRC1 is wiped out from both classes of genes on depletion, the preferential binding of JAK–STAT related transcription factors in the vicinity of irreversible genes might specifically foster their transcription after transient perturbation of PcG, dampening their re-repression and inducing a self-sustaining aberrant cell state (Extended Data Fig. 8). One of these JAK–STAT targets, *zfh1* plays an important role by blocking cell differentiation. Altogether, this cascade of events results in a self-sustaining mechanism that drives tumorigenesis even after recovery of normal PcG protein concentrations and in the wake of the rescue of their chromatin function at most of the PcG binding sites.

Previous work showed that self-sustaining alternative cell states can be triggered by transient perturbations in a sensitized *Drosophila* system[56], as well as in immortalized breast cells[57] or other cultured cells[58], including neural progenitor cells subjected to transient inhibition of the PRC2 complex[59]. PRC2 impairment in mouse striatal neurons induces progressive neurodegeneration by triggering a self-sustaining transcription derailment programme over time[60]. Furthermore, knock-out or transient chemical inhibition of PRC2 also led cells to enter a quasi-mesenchymal state that depends on ZEB1, the mouse homologue of fly *zfh1*, which is highly metastatic and associated with poor patient survival[53]. Therefore, epigenetic events might play a major role at early stages of oncogenesis or during tumour progression in some mammalian cancers[61]. Our survey of a large database of different types of solid cancer (Extended Data Fig. 9) as well as of data from several cohorts of patients with multiple myeloma (Extended Data Fig. 10) indicates that low expression levels of genes encoding canonical PRC1 subunits is associated with poor patient prognosis, consistent with a putative suppressive role for PRC1 in these tumour types. Future work might address the role of epigenetic perturbations in these tumours and in other physiological processes.

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

# Methods

## *Drosophila* strains and genetics

Flies were raised on a standard cornmeal yeast extract medium at 25 °C unless otherwise indicated. Fly lines and crosses performed to deplete PRC1 subunits or to perform control experiments were generated from stocks provided by the Bloomington *Drosophila* Stock Center (BL) and the Vienna *Drosophila* Resource Center (VDRC), as indicated below for each experiment. The work with transgenic strains of *Drosophila* was performed under the ethical approval no. n6906C2 of the Ministère de l'Enseignement Supérieur, de la Recherche et de l'Innovation, issued on 8 April 2020.

For KD experiments of PRC1 subunits and generation of EICs, Gal80$^{ts}$ was used to control the temporal *ph* or *Psc/Su(z)2* down-regulation by switching the temperature from 18 to 29 °C. KDs are generated in the larval EDs using the *ey*-FLP system. The rationale of the reversible KD system is the following: *ph*-RNAi, as well as the GFP marker, are under control of UAS sequences. Cells expressing *ey*-FLP (in pink in Fig. 1a) induce FLP-out of a transcriptional stop (located between two FRT sites and indicated in orange in Fig. 1a) in EDs, leading to expression of *act*-Gal4 (in light blue in Fig. 1a). *tub*-Gal80$^{ts}$ (in purple in Fig. 1a) encodes a ubiquitously expressed, temperature-sensitive Gal4 repressor. At restrictive temperature (29 °C), Gal80$^{ts}$ is inactivated. Gal4 activates UAS sequences, expressing *ph*-RNAi and GFP (as readout of *ph*-KD).

To perform KDs, flies were reared and crossed at 18 °C to inhibit Gal4 activity. A total of 80 virgin females were crossed with 20 males for each genotype and experiment. In all conditions (no, constant or transient KDs), flies were allowed to lay eggs at 18 °C for 4 h to synchronize embryonic and larval stages. As the timing of *Drosophila* development is temperature dependent, we adapted the timing for each KD condition to carry out phenotypic and molecular analyses at comparable developmental times. The genotypes of the flies on which we carried out the different KDs are listed below.

For *ph*-KD: *ey*-FLP, Act-gal4 (FRT.CD2 STOP) (BL#64095); *TubGal80$^{ts}$* (BL#7019); *UAS-ph-RNAi* (VDRC#50028)/*UAS-GFP* (BL#64095).

For *Psc-Su(z)2*-KD: *ey*-FLP, Act-gal4 (FRT.CD2 STOP) (BL#64095); *UAS-Psc-Su(z)2 RNAi* (BL#38261, VDRC#100096); *TubGal80$^{ts}$* (BL#7018)/*UAS-GFP* (BL#64095).

For control *white*-KD: *ey*-FLP, Act-gal4 (FRT.CD2 STOP) (BL#64095); *TubGal80$^{ts}$* (BL#7019); *UAS-w-RNAi* (BL#33623)/*UAS-GFP* (BL#64095).

All dissections were performed on female larvae at the L3 stage. For the no *ph*-KD (no depletion), flies were kept at 18 °C throughout development and dissected 10 days AEL. For the constant *ph*-KD (constant depletion), flies were kept at 29 °C throughout development and dissected 5 days AEL. For the larval depletion (from L1 to L3) flies were kept at 18 °C for 48 h and shifted at 29 °C until dissection 5 days AEL. For the transient *ph*-KD at the L1 stage, flies were kept at 18 °C for 48 h, then shifted at 29 °C for 24 h and returned to 18 °C until dissection 9 or 11 days AEL. For the transient *ph*-KD at the L2 stage, flies were kept at 18 °C for 96 h, shifted at 29 °C for 24 h and returned to 18 °C until dissection 8 days AEL. For the transient *ph*-KD at the L3 early stage, flies were kept at 18 °C for 120 h, shifted at 29 °C for 24 h and returned to 18 °C until dissection 8 days AEL. For the transient *ph*-KD at the L3 late stage, flies were kept at 18 °C for 168 h, shifted at 29 °C for 24 h and returned to 18 °C until dissection 8 days AEL. For the transient *Psc-Su(z)2*-KD at the L1 stage, flies were kept at 18 °C for 48 h, shifted at 29 °C for 48 h and returned to 18 °C until dissection 8 days AEL. For all conditions, a minimum of three biological replicates was performed. For each replicate, 150 discs were scored in PH depletions and more than 30 discs were scored for PSC depletions. Constant and transient depletions of PH (PH-d and PH-p) or PSC-SU(Z)2 generated tumours in 100% of dissected tissues.

To assess viability, we measured adult hatching rate. For this purpose, after 4 h of egg laying, we applied the treatments described above to produce *ph*-KD at the desired times. The vials were maintained at 18 °C and the number of pupae was counted for each condition. The adult hatching rate was calculated by dividing the number of male and female adults hatched from pupae by the number of pupae.

For the *zfh1*-RNAi and *Stat92E*-RNAi rescue experiments under constant *ph*-KD, *ey-FLP, Act5C-gal4 (FRT.CD2 STOP); + ; UAS-GFP* (BL#64095) females were crossed with males of various genotypes. For negative control experiments, females were crossed with *UAS-gfp-RNAi* (BL#9331); *UAS-w-RNAi* (BL#33623) males. To confirm that the *zfh1*-RNAi and *Stat92E*-RNAi do not induce any significant change in the eye development we crossed female to *UAS-zfh1-RNAi* (VDRC#103205); *UAS-w-RNAi* (BL#33623) and *UAS-Stat92E-RNAi* (VDRC#43866)*; UAS-w-RNAi* (BL#33623) males. Positive control experiments were conducted by crossing females with *UAS-gfp-RNAi* (BL#9331); *UAS-ph-RNAi* (VDRC#50028) males. For the rescue condition we crossed females to *UAS-zfh1-RNAi* (VDRC#103205); *UAS-ph-RNAi* (VDRC#50028) and *UAS-Stat92E-RNAi* (VDRC#43866)*; UAS-ph-RNAi* (VDRC#50028) males. This systematic breeding strategy facilitated the investigation of the specific roles of *zfh1* and *Stat92E* genes under constant *ph*-KD conditions.

Flies were reared and crossed at 18 °C and tumours were scored in the progeny reared at 18 °C. Note that in this genetic background there is no Gal80$^{ts}$ and therefore the KDs are obtained independently of the temperature. In the case of the *ph*-KD positive control, a tumour phenotype with 100% penetrance was observed in the progeny.

For the *zfh1*-RNAi and *Stat92E*-RNAi rescue experiments under transient *ph*-KD, *ey-FLP, Act5C-gal4 (FRT.CD2 STOP)* (BL#64095); + ; *Tub-Gal80$^{ts}$* (BL#7018)/TM6BTb females were crossed with males of various genotypes. For negative control experiments, females were crossed with *UAS-gfp-RNAi* (BL#9331); *UAS-w-RNAi* (BL#33623) males. To confirm that the *zfh1*-RNAi and *Stat92E*-RNAi do not induce any significant change in the eye development, we crossed female to *UAS-zfh1-RNAi* (VDRC#103205); *UAS-w-RNAi* (BL#33623) and *UAS-Stat92E-RNAi* (VDRC#43866)*; UAS-w-RNAi* (BL#33623) males. Positive control experiments were conducted by crossing females with *UAS-gfp-RNAi* (BL#9331); *UAS-ph-RNAi* (VDRC#50028) males. For the rescue condition we crossed females to *UAS-zfh1-RNAi* (VDRC#103205); *UAS-ph-RNAi* (VDRC#50028) and *UAS-Stat92E-RNAi* (VDRC#43866)*; UAS-ph-RNAi* (VDRC#50028) males. This systematic breeding strategy facilitated the investigation of the specific roles of the *zfh1* and *Stat92E* genes under transient *ph*-KD conditions.

Flies were reared and crossed at 18 °C and flies were allowed to lay eggs overnight at 18 °C. For transient depletion, flies were kept at 18 °C for 48 h, then shifted at 29 °C for 24 h and returned to 18 °C until dissection 10 days AEL.

Allografts were performed according to the protocol described previously[62]. The following fly line was used: *ey-FLP* (BL#5580), *Ubi-p63E(FRT.STOP)Stinger* (BL#32249)*; Tub-Gal80$^{ts}$* (BL#7019)*; Act5C-Gal4(FRT.CD2), UAS-RFP* (BL#30558)/*UAS-ph-RNAi* (VDRC#50028). Briefly, GFP-positive EDs from no-*ph*-KD, constant *ph*-KD or transient *ph*-KD L3 female larvae were dissected in PBS, cut into small pieces and injected into the abdomen of adult female hosts (BL#23650). The whole experiment was performed at 18 °C to avoid reactivation of *ph*-RNAi expression. To score tumour progression in allografts, flies were imaged every 2 days using Leica MZ FLIII to verify GFP as a readout of tumour growth. Tumours were dissected and re-injected when the host abdomen was fully GFP. Injected *Drosophila* pictures were taken using Ximea USB 3.1 Gen1 camera with a Sony CMOS-xiCAll sensor.

## Immunostaining procedures

EDs from L3 female larvae were dissected at room temperature in 1× PBS and fixed in 4% formaldehyde for 20 min. Tissues were permeabilized for 1 h in 1× PBS + 0.5% Triton X-100 on a rotating wheel. Permeabilized tissues were blocked for 1 h in 3% BSA PBTr (1× PBS + 0.1% Triton X-100), and incubated O/N on a rotating wheel at 4 °C with primary antibodies diluted in PBTr + 1% BSA. For double-strand break staining, larvae

were dissected at room temperature in 1× PBS, fixed in 4% paraformaldehyde for 30 min and primary antibodies were incubated for 2 h at room temperature. The following primary antibodies were used: goat anti-PH[63] (1:500), mouse anti-ELAV (1:1,000, DSHB, catalogue no. 9F8A9), mouse anti-ABD-B (1:1,000, DSHS, catalogue no. 1A2E9), chicken anti-GFP (1:500, Invitrogen, catalogue no. A10262), rabbit anti-ZFH1 (ref. 49) (1:2,000) and rabbit anti-histone H2AvD pS137 (1:500, Rockland, catalogue no. 600-401-914). Then, samples were washed in PBTr three times before adding secondary antibodies in PBTr for 2 h at room temperature on a rotating wheel. The following secondary antibodies were used: donkey anti-goat Alexa Fluor 555 (1:1,000, Invitrogen, catalogue no. A-21432), donkey anti-mouse Alexa Fluor 647 (1:1,000, Invitrogen, catalogue no. A-31571), donkey anti-chicken (1:1,000, Clinisciences, catalogue no. 703-546-155), donkey anti-rabbit Alexa Fluor 555 (1:1,000, Invitrogen, catalogue no. A-31572), donkey anti-rabbit Alexa Fluor 488 (1:1,000, Invitrogen, catalogue no. A-21206). F-actin was stained by adding rhodamine phalloidin Alexa Fluor 555 (1:1,000, Invitrogen, catalogue no. R415) or Alexa Fluor 488 (1:1,000, Invitrogen, catalogue no. A12379). Tissues were washed three times in PBTr. DAPI (4,6-diamidino-2-phenylindole) staining was performed at a final concentration of 1 µg ml$^{-1}$ for 15 min. Then discs were washed in PBTr and mounted in Vectashield medium (Eurobio Scientific, catalogue no. H-1000-10) or ProLong Gold antifade agent (Life Technologies, P36930). Image acquisition was performed using a Leica SP8-UV confocal microscope. ED areas were measured using Fiji[64] by drawing contour lines around the DAPI-labelled tissue and measuring their surface. A minimum of 30 EDs was considered to measure average ED areas in each condition. Images for quantification of double-strand break foci were taken with a DeltaVision deconvolution microscope using a ×60 oil immersion objective and a CoolSNAP HQ2 camera. Images were processed using Deconvolution through SoftWoRx v.6.0. All experiments were performed in biological duplicates.

## EdU staining

EdU experiments were performed using Click-iT Plus EdU Alexa fluor 555 Imaging kit (Invitrogen, catalogue no. C10638). The EDs of L3 female larvae were dissected at room temperature in Schneider medium. Then, EdU incorporation was performed for 15 min with 25 µM EdU solution on a rotating wheel at room temperature. After washing with PBS, tissues were fixed in 4% formaldehyde 30 min and washed three times with PBS. The imaginal discs were permeabilized for 1 h in 1× PBS + 0.5% Triton X-100 on a rotating wheel then blocked for 1 h in 1× PBS + 0.1% Triton X-100 + 3% BSA. EdU detection was performed according to the manufacturer's instructions for 30 min on a rotating wheel at room temperature away from light. Next, 500 µl of Click-iT reaction cocktail were prepared per tube containing 20 EDs. After 1× PBS + 0.1% Triton wash DAPI staining was performed at a final concentration of 1 µg ml$^{-1}$ for 15 min. Tissues were washed in 1× PBS + 0.1% Triton and discs were mounted in Vectashield medium. Image acquisition was performed using a Leica SP8-UV confocal microscope. Images of EdU stained EDs shown in Supplementary Videos were acquired using a Zeiss LSM980 Airyscan microscope in 4Y modality. Airyscan images of EdU stained EDs were processed with ZEN (v.3.6 Blue Edition, Zeiss) using default settings. Videos were created using Imaris (v.10.1, Oxford Instruments). All experiments were performed in biological duplicates.

## Analysis of chromosomal abnormalities

Chromosome preparation and FISH were performed as previously described[65,66]. EDs from L3 stage larvae were dissected in 0.7% NaCl solution and incubated in Colchicine solution (3 ml of 0.7% NaCl + 100 µl of 10$^{-3}$ M Colchicine) for 1 h at room temperature away from light. EDs were incubated in 0.5% sodium acetate for 7 min, followed by fixation (freshly prepared 2.5% PFA in 45% acetic acid) for 4 min on coverslip. EDs were pressed onto poly-lysine coated slides using manual force and snap frozen in liquid nitrogen. Slides were washed in 100% ethanol

for 5 min, air dried and stained with fluorescence in situ hybridization (FISH) probes for AACAC, AATAT and 359 base pair (bp) repeats as previously described[65]. Probe sequences are: 5′-6-FAM-(AACAC)₇, 5′-Cy3-TTTTCCAAATTTCGGTCATCAAATAATCAT and 5′-Cy5-(AATAT)₆. FISH staining was used to help identify chromosomes in rearranged conditions. Microscopy acquisition was performed on a DeltaVision deconvolution microscope using a ×60 oil immersion objective and a CoolSNAP HQ2 camera. Images were processed for Deconvolution using SoftWoRx v.6.0.

## Damage induction by X-ray exposure

L3 early-stage female larvae were transferred into a petri dish containing standard food medium, and were exposed to 5 Gy of X-rays using a Precision X-RAD iR160 irradiator. After irradiation, larval heads were dissected at indicated timepoints at room temperature in 1× PBS and fixed in 4% paraformaldehyde for 30 min before immunostaining. Microscopy and image analysis were performed as described above.

## RT−qPCR experiments

L3 female larvae were dissected in Schneider medium on ice. Total RNA was extracted from EDs using TRIzol reagent. RNA purification was performed using the RNA Clean & Concentrator kit (Zymo Research, catalogue no. R1015). Reverse transcription was performed using Maxima First Strand complementary DNA synthesis kit (Invitrogen, catalogue no. K1642). Quantitative PCR (qPCR) was performed using LightCycler 480 SYBR Green I Master Mix (Roche, catalogue no. 04707516001). qPCR with reverse transcription (RT−qPCR) experiments were analysed using LightCycler and GraphPad Prism software. All experiments were performed in biological triplicates.

## RNA-seq experiments

L3 female larvae were dissected in Schneider medium on ice. Total RNA was extracted from EDs using TRIzol reagent. RNA purification was performed using the RNA Clean & Concentrator kit (Zymo Research, catalogue no. R1015). Finally, poly-A RNA selection, library preparation and Illumina sequencing (20 M paired-end reads, 150 nt) were performed by Novogene (https://en.novogene.com/). All experiments were performed in triplicates.

## gDNA sequencing

gDNA was isolated using QIAamp DNA Micro Kit (Qiagen) following the manufacturer's instructions. For each biological replicate, roughly 70 EDs from wandering female larvae were dissected. In total, we sequenced four biological replicates for control samples (no *ph*-KD condition, that is, larvae of the crosses used for transient depletion that were reared at constant permissive temperature of 18 °C). Furthermore, 12 tumour samples were sequenced, that is, two biological replicates for six different depletion conditions as follows: (1) constant *ph*-KD; (2) transient *ph*-KD d9; (3) transient *ph*-KD d11; (4) early L3 *ph*-KD, 24 h recovery; (5) early L3 *ph*-KD, 96 h recovery and (6) early L3 *ph*-KD, 144 h recovery. All these conditions result in tumour formation. The gDNAs of all samples were processed for library preparation by Novogene (https://en.novogene.com/). Briefly, gDNA was fragmented to an average size of roughly 350 bp and then processed for DNA library preparation according to the manufacturer's (Illumina) paired-end protocols. Sequencing was performed using the Illumina Novaseq 6000 platform to generate 150 bp paired-end reads with a coverage of at least ten times for 99% of the genome.

## Western blot

Roughly 150 EDs were dissected in Schneider medium on ice per replicate. To collect sufficient material, EDs were dissected in batches, snap frozen in liquid nitrogen and stored at −80 °C. Discs were homogenized with a Tenbroeck directly in radioimmunoprecipitation assay lysis buffer (50 mM Tris pH 7.5, 150 mM NaCl, 1% NP40, 0.5%

Na-deoxycholate, 0.1% SDS, 2× protease inhibitor) and incubated on ice for 10 min. If necessary, a second round of mechanical dissociation was performed. Samples were centrifuged for 10 min at 10,000$g$ at 4 °C and the supernatant was transferred to a fresh tube. Proteins were quantified using BCA protein assay and 10 μg were used per gel lane, before 40 min of migration at 200 V in MES 20× migration buffer and 1 h of transfer (1 A). Membranes were blocked for 1 h in PBS + 0.2% Tween + 10% milk powder at room temperature, incubated O/N with primary antibodies in PBS + 0.2% Tween at 4 °C on a shaker and washed in PBS + 0.2% Tween. The following primary antibodies were used: rabbit anti-PH (1:200), rabbit anti-zfh1 (ref. 49) (1:2,000), mouse anti-beta tubulin (1:5,000, DSHB, catalogue no. AA12.1). HRP-conjugated secondary antibodies were incubated with the membrane for 2 h at room temperature. The following secondary antibodies were used: goat antirabbit (1:15,000, Sigma, catalogue no. A0545), rabbit antimouse (1:15,000, Sigma, catalogue no. A9044). Membranes were washed in PBS + 0.2% Tween and revealed using Super Signal West Dura kit (Pierce) and Chemidoc Bio-Rad. Western blots were analysed using ImageLab software v.6.1 from Bio-Rad. The full-size raw blot images are provided in the Supplementary Fig. 1.

## ChIP–seq experiments

ChIP–seq on L3 EDs were performed as described previously[41], with minor modifications, and 400 EDs were used per replicate. If necessary, several dissection and/or collection batches were frozen in liquid nitrogen and stored at −80 °C to collect sufficient material. Chromatin was sonicated using a Bioruptor Pico (Diagenode) for 10 min (30 s on, 30 s off). PH antibodies[67] were diluted 1:100 for immunoprecipitation. After decrosslinking, DNA was purified using MicroChIP DiaPure columns from Diagenode. DNA libraries for sequencing were prepared using the NEBNext Ultra II DNA Library Prep Kit for Illumina. Sequencing (paired-end sequencing 150 bp, roughly 4 Gb per sample) was performed by Novogene (https://en.novogene.com/). All experiments were performed in biological duplicates.

## CUT&RUN experiments

CUT&RUN experiments were performed as described by Kami Ahmad in protocols.io (https://doi.org/10.17504/protocols.io.umfeu3n) with minor modifications. We dissected 50 EDs in Schneider medium, centrifuged them for 3 min at 700$g$ and washed them twice with wash+ buffer before adding concanavalin A-coated beads. MNase digestion (pAG-MNase Enzyme from Cell Signaling) was performed for 30 min on ice. After ProteinaseK digestion, DNA was recovered using SPRIselect beads and eluted in 50 μl of Tris-EDTA. DNA libraries for sequencing were prepared using the NEBNext Ultra II DNA Library Prep Kit for Illumina. Sequencing (paired-end sequencing 150 bp, roughly 2 Gb per sample) was performed by Novogene (https://en.novogene.com/). The following antibodies were used: H3K27me3 (1:100, Active Motif, catalogue no. 39155), H3K27Ac (1:100, Active Motif, catalogue no. 39133), H2AK118Ub (1:100, Cell Signaling, catalogue no. 8240). All experiments were performed in biological duplicates.

## ATAC-Seq experiments

ATAC-Seq experiments were performed using the ATAC-Seq kit from Diagenode (catalogue no. C01080002). Ten EDs were used as starting material for each replicate and condition. Tagmented DNA was amplified by PCR using 13 cycles and the purified DNA libraries were sequenced (paired-end sequencing 150 bp, roughly 2 Gb per sample) by Novogene (https://en.novogene.com/). All experiments were performed in biological duplicates.

## Statistics and reproducibility

ChIP–seq, CUT&RUN and ATAC-Seq were performed in duplicates, following Encode's standards (https://www.encodeproject.org/chip-seq/transcription_factor/#standards; https://www.encodeproject.org/atac-seq/#standards). RNA-seq were performed in triplicates, following Encode's recommendations (https://www.encodeproject.org/data-standards/rna-seq/long-rnas/).

In general, immunostaining experiments were performed in biological duplicates. Each biological replicate was obtained from independent genetic crosses. The only exception was the phospho-H2AV staining shown in Fig. 1j and Extended Data Fig. 2c, which was performed once, but scoring tissues that came from six independent genetic crosses. For sample sizes of immunostaining experiments, see the sheet named 'All IF sample numbers' in Supplementary Table 6. For transcriptomic, RT–qPCR and western blot analysis, experiments were performed in biological triplicates. ATAC-Seq, CUT&RUN, ChIP–seq and immunostaining experiments were performed in biological duplicates. Each biological replicate was obtained from independent genetic crosses.

For experiments presented in Figs. 1 and 5, as well as Extended Data Figs. 1, 2, 3, 5 and 6, involving genetic crosses with different lines and in different conditions, followed by tissue area measurements and immunofluorescence, two independent biological replicates were performed with similar results. Measured areas and the number of tissues analysed in imaging are reported in Supplementary Table 6.

Allograft experiments were performed in two independent biological replicates. In the first replicate, one starting tumour obtained on constant PH depletion and one tumour obtained from transient PH depletion were used. In the second replicate, two constant PH depletion and two transient PH depletion tumours were injected. Results were similar for both replicates. The total number of injected host flies is reported in the graphs of the Extended Data Fig. 7b,c.

## Bioinformatic analyses on *Drosophila* datasets

All in-house bioinformatic analyses were performed in R v.3.6.3 (https://www.R-project.org/). Computations on genomic coordinate files and downstream computations were conducted using the data.table R package (data.table: Extension of 'data.frame'. https://r-datatable.com, https://Rdatatable.gitlab.io/data.table, https://github.com/Rdatatable/data.table, v.1.14.2). In all relevant panels of figures and Extended Data figures, box plots depict the median (line), upper and lower quartiles (box) ±1.5× interquartile range (whiskers) and outliers are not shown. For each relevant panel, the statistical test that was used is specified in the caption: NS denotes not significant ($P > 0.05$), $*P < 5 \times 10^{-2}$, $**P < 1 \times 10^{-2}$, $***P < 1 \times 10^{-3}$, $****P < 1 \times 10^{-5}$.

## gDNA processing and mapping of somatic variants

gDNA variant calling was performed by Novogene (https://en.novogene.com/). Briefly, base calling was performed using Illumina pipeline CASAVA v.1.8.2, and subjected to quality control using fastp with the following parameters: -g -q 5 -u 50 -n 15 -l 150 --min_trim_length 10 --overlap_diff_limit 1--overlap_diff_percent_limit 10. Then, sequencing reads were aligned to the dm6 version of the *Drosophila* genome using Burrows–Wheeler aligner with default parameters and duplicate reads were removed using samtools and PICARD (http://picard.sourceforge.net). Raw SNP and InDel sets were called using GATK with the following parameters: --gcpHMM 10 -stand_emit_conf 10 -stand_call_conf 30. Then, SNPs were filtered using the following criteria: SNP QD < 2, FS > 60, MQ < 30, HaplotypeScore > 13, MappingQualityRankSum < −12.5, ReadPosRankSum < −8. For INDEL variants, the following criteria were used: QD < 2, FS > 200, ReadPosRankSum < −20. UCSC known genes were used for gene and region annotations. Finally, the variants were compared to a batch-matched control sample (no *ph*-KD), in the search for bona fide SNVs and InDels using the MuTect2 module of the GATK package. Only SNVs and InDels variants that passed Mutect2 filtering (FILTER = "PASS") were considered for downstream analyses. Structural variants and CNVs were detected using breakdancer (https://github.com/genome/breakdancer) and CNVnator (https://github.com/abyzovlab/CNVnator) software packages, respectively.

Then, called variants were imported in R for downstream analyses. When looking at the fraction of tumour samples that contained a given alteration (Fig. 1h), we only retained SNVs or InDels with an allelic fraction greater than 0.2, structural variants that were supported by at least five reads and CNVs with an allelic fraction bigger than 1.5 (duplication) or smaller than 0.66 (deletion).

### RNA-seq processing and differential analysis

After initial quality checks of the newly generated data using fastqc (http://www.bioinformatics.babraham.ac.uk/projects/fastqc/), the paired-end reads were aligned to a custom index consisting of the dm6 version of the *Drosophila* genome together with GFP, EGFP and mRFP1 sequences, using the align function from the Rsubread R package[68] (v.2.0.1) with the following parameters: maxMismatches = 6, unique = TRUE. Next, aligned reads were counted for each *D. melanogaster* transcript (dmel_r6.36 annotation) using the featureCounts function from the Rsubread R package (v.2.0.1, isPairedEnd = TRUE) and differential expression analysis was performed using the DESeq2 R package[69] (v.1.26.0, design = -replicate + condition). The tables corresponding to the different comparisons are available in Supplementary Tables 1, 4 and 5.

For the differential analysis of the transcriptomes after no *ph*-KD (control), constant and transient *ph*-KD, each *ph*-RNAi sample was compared to temperature-matched *w*-RNAi controls (Fig. 2a and Extended Data Fig. 8b). DESeq2 outputs are available in Supplementary Tables 1 and 5. For the differential analysis of the transcriptomes after *zfh1*+*w*-KD, *Stat92E*+*w*-KD, *gfp*+*ph*-KD, *zfh1*+*ph*-KD and *Stat92E*+*ph*-KD, all were compared to temperature-matched *gfp*+*w*-KD (Supplementary Table 4).

### Clustering of differentially expressed genes

For the clustering, we selected the genes that were differentially expressed ($P_{adj} < 0.05$ and $|\log_2 fold_2$ fold change $| > 1$) after constant or transient *ph*-KD (d9 or d11 AEL). In addition, we only considered the genes that did not show significant changes after no *ph*-KD (control). Then, $\log_2$ fold change values were clipped at the 5th and 95th percentiles and clustered using the supersom function from the kohonen R package[70] (v.3.0.10). As day 9 and day 11 transient *ph*-KD yielded substantially similar transcriptomes, a two-layer self-organizing map was trained (layer 1, constant *ph*-KD; layer 2, D9 and D11 transient *ph*-KD) with similar weights for the two layers, using a 3 × 2 grid (topology = hexagonal, toroidal = TRUE). Clustering output is in Supplementary Table 2.

### CUT&RUN, ChIP−seq and ATAC-Seq processing, peak calling and differential analysis

After initial quality checks of the newly generated data using fastqc, the reads were aligned to the dm6 version of the *Drosophila* genome using bowtie 2 (ref. 71, v.2.3.5.1) with the following parameters: --local --very-sensitive-local --no-unal --no-mixed --no-discordant --phred33 -I10 -X 700, and low mapping quality reads were discarded using samtools[72] (-q 30, v.1.10, using htslib v.1.10.2-3).

PH, H3K27me3, H3K27Ac, H2AK118Ub and ATAC-Seq peaks and/ or domains were called for each replicate separately and on merged reads using macs2 (ref. 73, v.2.2.7.1) with the following parameters: --keep-dup 1 -g dm -f BAMPE -B --SPMR. For PH ChIP−seq, the input sample was used as control. For H3K27me3, H3K27Ac and H2AK118Ub CUT&RUN, the IgG sample was used as control. Only peaks detected in both replicates (enrichment greater than 0 AND *q* value less than 0.05) and using merged replicates (enrichment greater than 2 AND *q* < 0.01) were retained for further analyses, after being merged with a minimum gap size of 250 bp for narrow peaks (PH, H3K27Ac and ATAC-Seq) and 2.5 kb for broad marks (H3K27me3 and H2AK118Ub). The macs2 bedgraph files were used for visualization purposes.

For the differential analysis of H3K27me3, H3K27Ac, H2AK118Ub CUT&RUN and ATAC-Seq, peaks and/or domains were first merged across all conditions (maximum gap of 250 bp for H3K27Ac and ATAC-Seq peaks; 2.5 kb for H3K27me3 and H2AK118Ub domains) and overlapping reads were counted using the featureCounts function from the Rsubread R package (v.2.0.1, isPairedEnd = TRUE). Differential analysis was then performed using the DESeq2 R package (v.1.26.0, size factors, total number of aligned reads; design, -replicate + condition). The same procedure was used for the differential analysis of ATAC-Seq peaks between *zfh1*+*ph*-KD and *gfp*+*ph*-KD.

### Clustering of differentially accessible ATAC-Seq peaks

For the clustering of ATAC-Seq peaks, we only considered the peaks showing a significant difference ($P_{adj} < 1 \times 10^{-3}$ and $|\log_2 fold change| > 1$) after constant or transient *ph*-KD (day 11 AEL) and with a minimum $\log_{10}$ base mean of 1.25 to avoid noisy peaks. The $\log_2$ fold change values were clipped at the 5th and 95th percentiles and clustered using the supersom function from the kohonen R package[70] (v.3.0.10) using a four-layer self-organizing map (layer 1, $\log_2$fold change constant *ph*-KD; layer 2, $\log_2$fold change transient *ph*-KD; layer 3, $P_{adj}$ constant *ph*-KD; layer 4, $P_{adj}$ transient *ph*-KD) with similar weights for the four layers, using a 1 × 3 grid (topology = hexagonal, toroidal = TRUE). Full clustering output is available in Supplementary Table 3.

### Classification of PcG target genes and peaks-to-gene assignment

To define PcG target genes, we defined a clean set of H3K27me3 domains in the control (no *ph*-KD) condition by removing artefactual splits due to sequencing gaps (github), resulting in 241 domains. Then, only the genes for which at least 50% of the gene body was overlapping with a H3K27me3 domain were considered as direct PcG target. When relevant, only irreversible, reversible and unaffected genes that were direct PcG targets when considered (Fig. 3). PcG target gene assignment is available in Supplementary Table 2 (PcG_bound and class columns).

To assess whether a gene was overlapping a H3K27me3 domain or a H3K27Ac peak in a given condition, we used different criteria. For H3K27me3 (Fig. 3b), only the genes for which at least 50% of the gene body was overlapping a confident H3K27me3 domain ('CUT&RUN, ChIP−seq and ATAC-Seq processing, peak calling and differential analysis' section above) were considered as hits. For H3K27Ac (Fig. 3c), only the genes containing a confident peak ('CUT&RUN, ChIP−seq and ATAC-Seq processing, peak calling and differential analysis' section above) in the gene body or up to 2.5 kb upstream of the TSS were considered as hits.

To assign PH peaks (Extended Data Fig. 6e) or ATAC-Seq peaks (Fig. 4b), peaks were assigned to the closest TSS with a maximum genomic separation of 25 kb (peaks that were located further away were not considered).

### Gene Ontology terms enrichment

Gene Ontology terms associated with the genes of interest and a background set of genes, consisting of all the genes that passed DESeq2 initial filters, were retrieved using the AnnotationDbi R package (https://bioconductor.org/packages/AnnotationDbi.html, v.1.48.0). For each Gene Ontology term, over-representation was assessed using a one-sided Fisher's exact test (alternative = 'greater'). Obtained *P* values were corrected for multiple testing using false discovery rate (FDR).

### Motif enrichment

To search for DNA binding motifs enriched at each ATAC-Seq cluster, we used the centre of corresponding peaks ±250 bp (500 bp total). Resulting regions were analysed with the i-cisTarget online tool[74], using v.6.0 of the position weight matrix database (consisting of 24,453 position weight matrics). Only top scoring motifs with a normalized enrichment score greater than 5.5 and a rank less than 50 were considered (Fig. 4e).

To search for motifs associated with increased or decreased accessibility after constant or transient *ph*-KD, we used a collection of

non-redundant transcription factor motifs[75] and counted their occurrences across all ATAC-Seq peaks ±250 bp, using the matchMotifs function from the motifmatchR package (v.1.18.0; https://doi.org/10.18129/B9.bioc.motifmatchr) with the following parameters: $P_{cutoff} = 5 \times 10^{-4}$, bg = 'genome', genome = 'dm6'. Of note, only motifs associated with a *Drosophila* transcription factor gene that passed initial DESeq2 initial filters were considered. Then, we fitted two LASSO regressions using the cv.glmnet and the glmnet functions from the glmnet package in R (v.4.1.4), with the following parameter: lambdas = $10^{seq}(2, -3, by = -0.1)$, standardize, TRUE; nfolds, 5), aiming at predicting $\log_2$ fold changes after constant or transient *ph*-KD. The top 25 motifs with the strongest |s0| coefficients in any of the two models were used to train two linear models to predict $\log_2$ fold changes after transient or constant *ph*-KD. Only the motifs with a significant coefficient in at least one of the two linear models ($P < 1 \times 10^{-5}$) were considered (Fig. 4f).

### Analysis of human solid tumours

The differential gene expression analysis was carried out by using a Mann–Whitney test and the TNMplot database, which contains transcriptome-level RNA-seq data for different tumour samples from The Cancer Genome Atlas (TCGA) and The Genotype-Tissue Expression (GTEx) repositories[76].

The survival analysis was carried out using the Pan-Cancer (Bladder, Lung adenocarcinoma and Rectum adenocarcinoma) or gene array (Breast, Ovarian and Prostate) datasets[77,78] of the online tool www.kmplot.com (accessed on 22 December 2022). The Pan-Cancer dataset is based on TCGA data generated using the Illumina HiSeq 2000 platform with survival information derived from the published sources[79]. The gene-array samples were obtained using Affymetrix HGU133A and HGU133plus2 gene chips. The samples were MAS5 normalized and the mean expression in each sample was scaled to 1,000. The most reliable probe sets to represent single genes were identified usNAiing JetSet[80].

In the survival analysis, each cut-off value between the lower and upper quartiles of expression was analysed by Cox proportional hazards regression and FDR was computed to correct for multiple hypothesis testing. Then, the best performing cut-off was used when drawing the Kaplan–Meier survival plots that were generated to visualize the survival differences. Hazard rates with 95% confidence intervals were computed to numerically assess the survival time difference between the two cohorts. The statistical analysis was performed in the R statistical environment (www.r-project.org). The analysis results for single genes can be validated using the platforms at www.kmplot.com and www.tnmplot.com.

### Analysis of cohorts of patients with multiple myeloma

For gene expression profiling data from patients with multiple myeloma, we used six cohorts that included Affymetrix gene expression data (HGU133plus2) of purified multiple myeloma cells from the TT2 (ref. 81) (Gene Expression Omnibus, accession number GSE2658), TT3 (ref. 82) (accession number E-TABM-1138 accession number GSE4583) and Hovon[83] (accession number GSE19784) cohorts (345, 158 and 282 newly diagnosed patients with multiple myeloma who were treated with high-dose melphalan and autologous haematopoietic stem cell transplantation); the Mulligan cohort[84] (188 patients at relapse treated by proteasome inhibitor in monotherapy); the Mtp cohort non-eligible for HDT[85] (63 newly diagnosed patients with multiple myeloma who were not eligible for high-dose melphalan and autologous haematopoietic stem cell transplantation) and the Mtp Dara cohort[85,86] (51 patients at relapse treated by anti-CD38 monoclonal antibody (Daratumumab)). Gene expression data were normalized with the MAS5 algorithm and processing of the data was performed using the webtool genomicscape (http://www.genomicscape.com), as done previously[87,88], using the R environment (www.r-project.org). The prognostic values of PHC1, PHC2, PHC3, CBX2, CBX7 and BMI1 gene expression was investigated using the Maxstat R function and Kaplan–Meier survival curves as

previously described[89]. The differential gene expression analysis between normal bone marrow plasma cells from healthy donors and multiple myeloma cells from patients was carried out by using the Mann–Whitney test. The prognostic value of *PHC1*, *PHC2*, *PHC3*, *CBX2*, *CBX7* and *BMI1* genes was combined using our previously published methodology[89] (sum of the Cox *b* coefficients of each of the six genes, weighted by ±1 if the patient's multiple myeloma cell signal for a given gene is above or below the probe set Maxstat value of the gene). Clustering was performed using the Morpheus software (https://software.broadinstitute.org/morpheus) and violin plots using GraphPad Prism software (http://www.graphpad.com/scientific-software/prism/).

### Reporting summary

Further information on research design is available in the Nature Portfolio Reporting Summary linked to this article.

### Data availability

The NGS datasets generated in this study were made publicly available in the Gene Expression Omnibus (accession number GSE222193). A UCSC browser to visualize the data is available at http://genome-euro.ucsc.edu/s/cavalli/EpiCancer.

### Code availability

All custom scripts that were generated for this study were made publicly available at https://github.com/vloubiere/Parreno_Loubiere_2023.

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

**Acknowledgements** We thank Montpellier Resources Imagerie facility as well as the *Drosophila* facilty (both affiliated to BioCampus University of Montpellier, CNRS, INSERM, Montpellier, France). We thank A.-M. Popmihaylova for help with immunostaining of *Drosophila* tissues. We thank J. Drouin for discussions and advice on the manuscript. We thank E. Soler for discussions on the function of the ZEB1 protein in cancer. V.P. was supported by the EpiGenMed cluster of Excellence funding (Programme d'Investissements d'Avenir of the French Ministry of Higher Education and Research) and by la Ligue Nationale Contre le Cancer. V.L. was supported by the EpiGenMed cluster of Excellence funding (PIA of the French Ministry of Higher Education and Research). A.-M.M. was supported by the University of Montpellier and a grant from the Fondation ARC (contract no. 216574, acronym 'Epicancer'). B.S. was supported by INSERM. G.C. was supported by CNRS. I.C. was supported by National Institutes of Health grant no. R01GM117376 and National Science Foundation Career no. 1751197.

Research in the G.C. laboratory was supported by grants from the European Research Council (Advanced Grant 3DEpi), the European CHROMDESIGN ITN project (Marie Skłodowska-Curie grant agreement no. 813327), the European E-RARE NEURO DISEASES grant 'IMPACT', by the Agence Nationale de la Recherche (PLASMADIFF3D, grant no. ANR-18-CE15-0010), by the Fondation pour la Recherche Médicale (grant no. EQU202303016), by the MSD Avenir Foundation ((Project GENE-IGH) and by the French National Cancer Institute (INCa, PIT-MM grant no. INCA-PLBIO18-362). M.E. was supported by RSF grant no. 20-74-10099.

**Author contributions** V.L., V.P., A.-M.M. and G.C. initiated and led the project. V.P., L.F. and V.L. performed genetic experiments. V.P. performed immunostaining, molecular biology and genomic experiments. V.L. and M.D.S. performed computational analysis of genomic datasets. V.P. and A.-M.M. performed allograft experiments. B.S. performed ChIP–seq, ATAC-Seq and CUT&RUN experiments. D.N. helped with EdU imaging. M.E., B.G. and D.C. performed computational analysis of different tumour types. J.M. performed computational analysis of multiple myeloma samples. C.C.R. performed irradiation experiments and N.L.B. performed karyotyping under the guidance of I.C. V.L., V.P., A.-M.M. and G.C. wrote the manuscript. All the authors discussed the data and reviewed the manuscript.

**Competing interests** The authors declare no competing interests.

**Additional information**
**Correspondence and requests for materials** should be addressed to A.-M. Martinez or G. Cavalli.

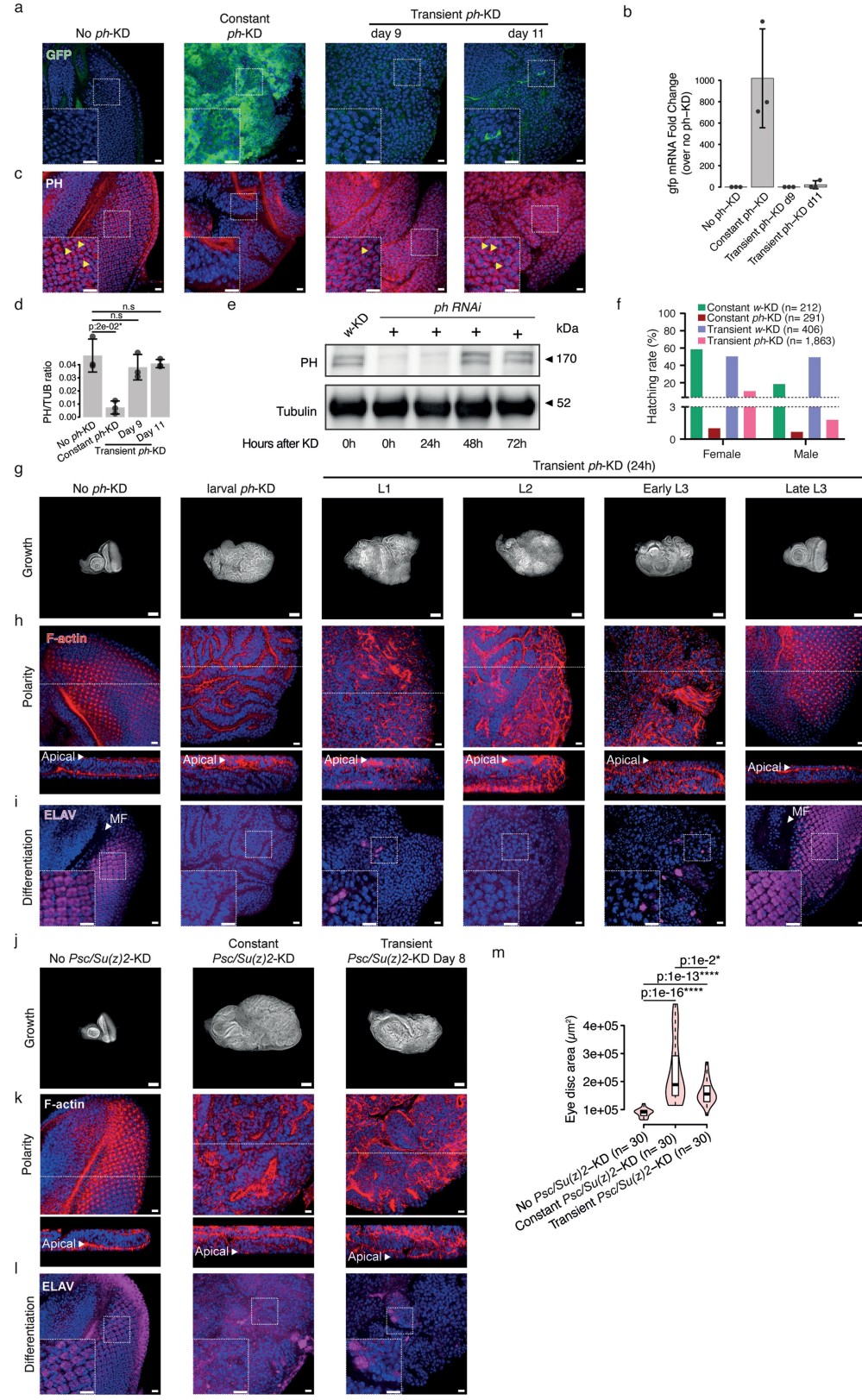

**Extended Data Fig. 1** | See next page for caption.

**Extended Data Fig. 1 | Transient PRC1 depletion generates neoplastic tumours that persist after PH protein recovery. a-** GFP staining (in green) used as a readout of the conditional *ph* knockdown (*ph*-KD) system described in Fig. 1a after no *ph*-KD (control), constant or transient *ph*-KD. The tissues were counterstained with DAPI (blue). Two independent experiments were performed with similar results. **b-** *gfp* mRNA fold change RT-qPCR measurement after no *ph*-KD (control), constant or transient *ph*-KD. Bars correspond to the mean ± standard deviation (whiskers) inferred from three biological replicates (grey dots). **c-** PH immunostaining (in red) after no *ph*-KD (control), constant or transient *ph*-KD. The tissues were counterstained with DAPI (blue). Two independent experiments were performed with similar results. **d-** Quantification of the western blot illustrated in Fig. 1b and of two other biological replicates. Bars correspond to the mean ratio ±standard deviation (whiskers) of the signal of PH over that of TUBULIN (PH/TUB) calculated from three biological replicates (grey dots). Two-sided unpaired t.test: *pval < 0.05, ns = pval > 0.05 (not significant). Error bars represent the standard error of the mean for three biological replicates. Dunnet's test: ns = not significant, **pval < 0.01. **e-** Western blot showing the PH protein after early L3 EDs were subjected to 24 h of *white*-KD (*w*-KD, control) or *ph*-KD followed by 0 h, 24 h, 48 h and 72 h of recovery at 18 °C (see bottom axis). This time course illustrates acute depletion and allows visualization of the kinetics of PH recovery after *ph*-KD. **f-** Hatching rate after constant or transient *ph*-KD. **g-i-** DAPI (in grey, g), F-actin (in red, h) and ELAV (in magenta, i) stainings of L3 EDs after no *ph*-KD (control), *ph*-KD throughout the three larval stages (L1, L2, L3) or transient (24 h) *ph*-KD during the first (L1), second (L2), early (Early L3) or late (Late L3) of the L3 stage, respectively. DAPI staining is used to assess ED growth, F-actin for apico-basal polarity, and the neuronal marker ELAV for differentiation. Note that late L3 tissues look normal immediately after the end of the *ph*-KD. Nevertheless, their cells are reprogrammed into a malignant state, as indicated by the fact that allografts of these tissues induce tumours, as shown in Extended Data Fig. 7i, j. Two independent experiments were performed with similar results. Scale bars: 10 µm (a, c, h, i), 100 µm (g). **j-l-** DAPI (in gray, j), F-actin (in red, k) and ELAV (in magenta, l) stainings of EDs after no *Psc/Suz(2)*-KD (control, left), constant (middle) or transient *Psc/Suz(2)*-KD (right), respectively. DAPI staining is used to assess growth, F-actin for apico-basal polarity, and the neuronal marker ELAV for differentiation. Two independent experiments were performed with similar results. **m-** ED sizes quantified as overall area of DAPI staining after no *Psc/Suz(2)*-KD (control), constant or transient *Psc/Suz(2)*-KD conditions. n = 30 for each condition. Two-sided Wilcoxon test: *pval < 5e-2, ****pval<1e-5. Box plots show the median (line), upper and lower quartiles (box) ±1.5x interquartile range (whiskers), outliers are not shown. Scale bars: 100 µm (j), 10 µm (k, l).

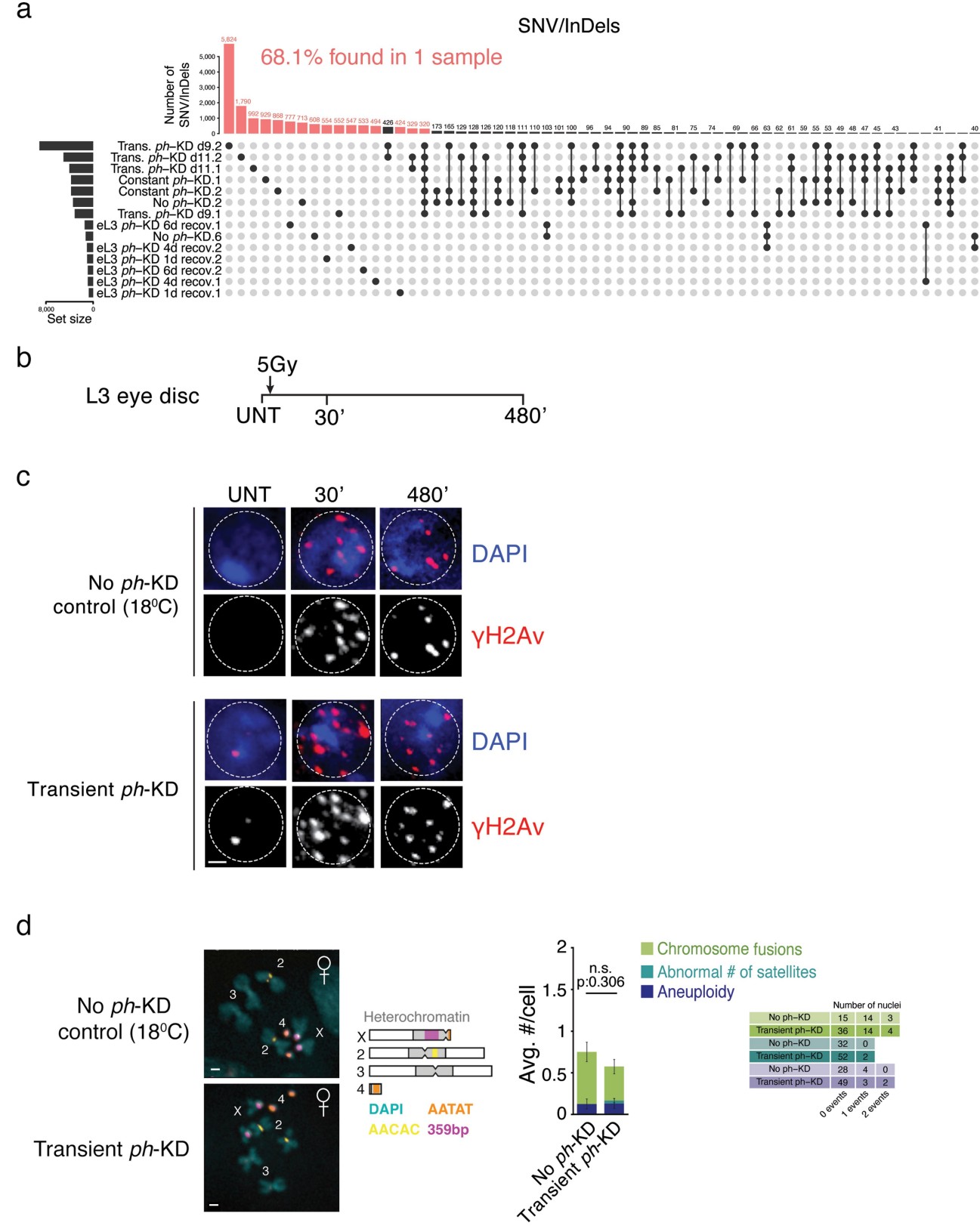

**Extended Data Fig. 2** | See next page for caption.

**Extended Data Fig. 2 | *ph*-KD does not induce the accumulation of mutations or aneuploidy. a-** SNV/InDels overlaps between all sequenced gDNA samples. Each vertical bar corresponds to an intersection (corresponding samples are shown below) and horizontal bars (bottom left) indicate the total number of SNV/InDels found in each sample. Only intersections containing ≥40 SNV/InDels are shown and SNV/InDels that are specific to one sample are shown in orange (68.1% of all SNV/InDels detected). **b-** Schematic view of the repair kinetic experiments. γH2Av foci were counted before or 30 min and 480 min after ionizing radiation (IR). **c-** Representative γH2Av staining in no *ph*-KD (control, top) and transient *ph*-KD EDs (bottom). Nuclei were counterstained with DAPI (in blue). **d-** Representative karyotypes (left) and quantification of chromosome abnormalities in EDs after no *ph*-KD (control, top) and transient *ph*-KD (bottom). The schematic representation shows the position of the satellites stained by FISH. Abnormalities were quantified from two biological replicates per condition (bar plot on the right, n = 32 for No *ph*-KD and n = 53 for *ph*-KD karyotypes). Bars correspond to the mean number of aberrations per cell ±standard error (whiskers). Two-sided t.test: ns = pval>0.05 (not significant). For each type of type of abnormality (see colour legend), the number of counted events are shown on the right (tables). Scale bars = 1 µm (c, d).

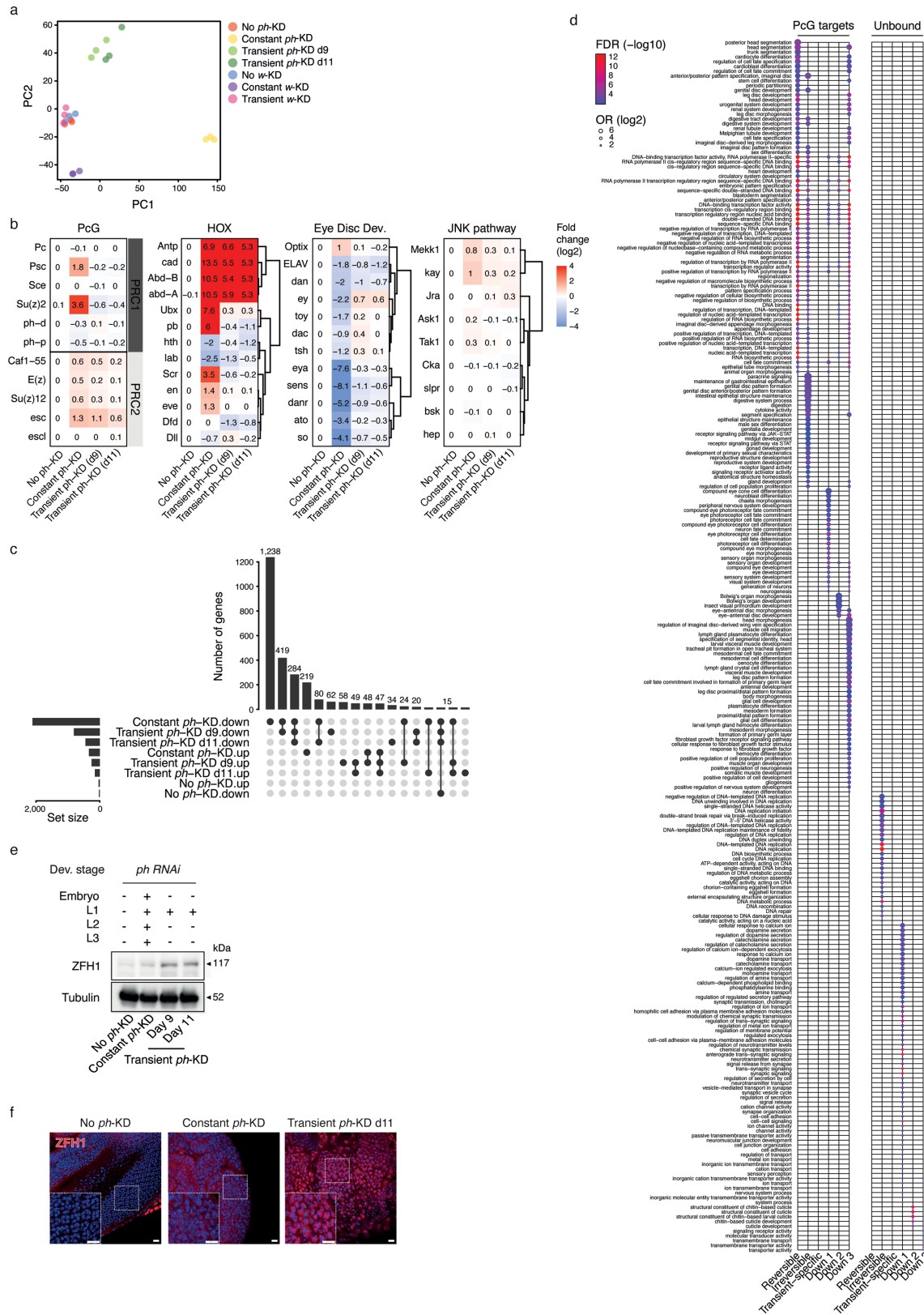

**Extended Data Fig. 3** | See next page for caption.

**Extended Data Fig. 3 | Transcriptional defects after constant or transient *ph*-KD include induction of ZFH1. a-** Principal component analysis (PCA) of normalized RNA-Seq read counts for different conditions. Each dot corresponds to one biological replicate. A close distance between samples reflects their similarity, showing that the control samples (no *w*-KD, no *ph*-KD) and the transient *w*-KD are very similar. **b-** Transcriptional fold changes after no *ph*-KD (control), constant or transient *ph*-KD (see x-axis) of the PcG core components, Hox genes (canonical targets of PcG repression), key genes that regulate ED development and the JNK pathway core members. **c-** Overlaps of differentially expressed genes between indicated RNA-Seq samples. Each vertical bar corresponds to an intersection (corresponding samples are shown below) and horizontal bars (bottom left) indicate the total number of differentially expressed genes in each sample. **d-** GO terms enriched for each gene cluster, then stratified as being direct PcG targets ( ≥ 50% of the gene body overlaps a H3K27me3 repressive domain) in control condition (left) or not (right). **e-** Western blot showing ZFH1 levels in EDs after no *ph*-KD (control), constant or transient *ph*-KD. Three independent experiments were performed with similar results. **f-** ZFH1 immunostaining (in red) after no *ph*-KD (control), constant or transient *ph*-KD. Tissues were counterstained with DAPI (in blue). Two independent experiments were performed with similar results. Scale bars: 10 μm.

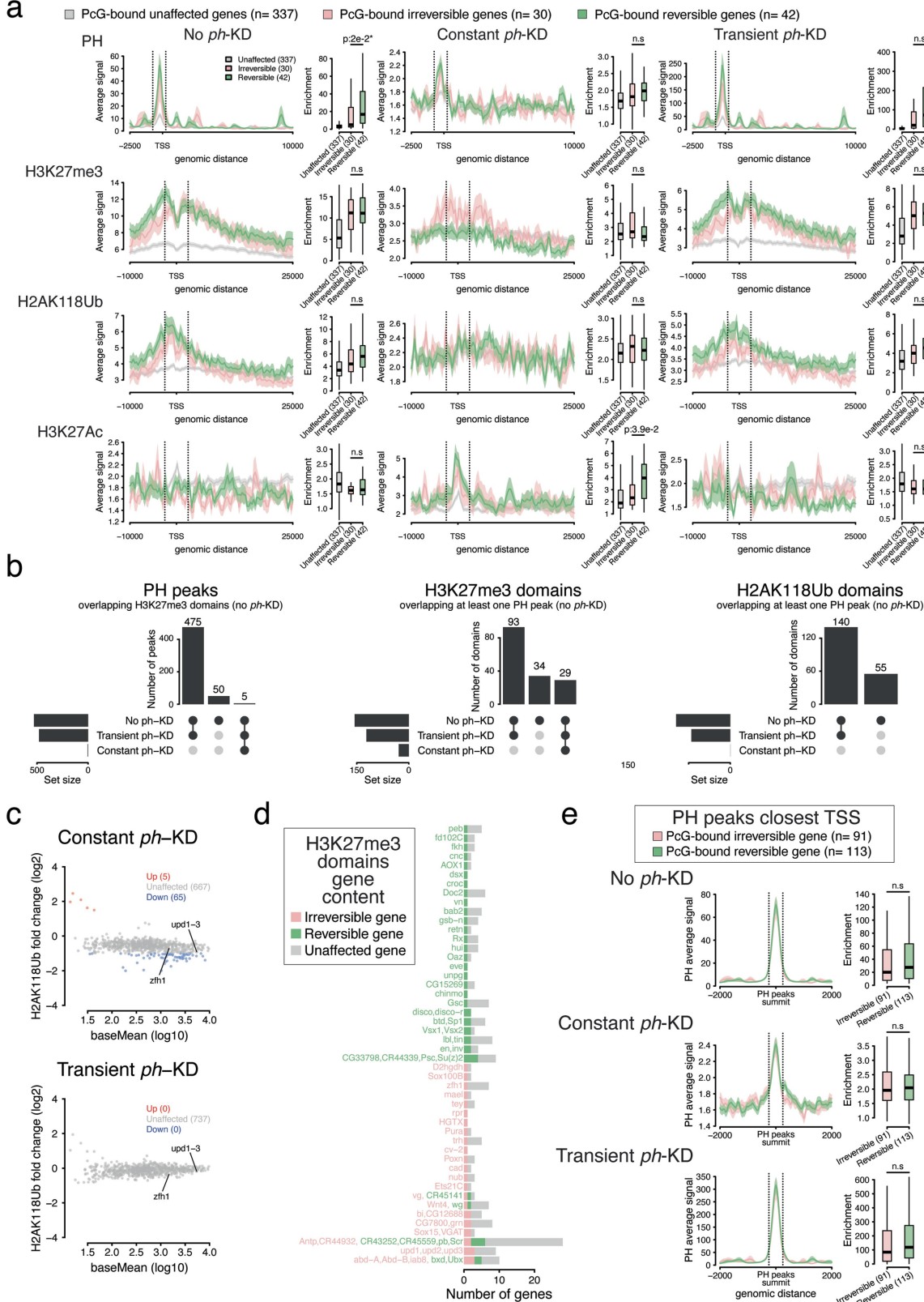

**Extended Data Fig. 4** | See next page for caption.

**Extended Data Fig. 4 | The PcG epigenetic landscape is globally re-established after transient *ph*-KD. a-** PH ChIP-Seq (top row), H3K27me3 (2nd row), H2AK118Ub (3rd row) and H3K27Ac (bottom row) CUT&RUN average tracks, anchored at the TSS of the PcG-bound irreversible (in pink), reversible (in green) and unaffected genes (in gray) after no *ph*-KD (control, left), constant (middle) or transient *ph*-KD (right). For each condition, the average signal is shown (solid line) ± standard error (shaded area). The distance to the TSS is shown on the x-axis. The signal was quantified at the regions highlighted by dashed lines (see corresponding boxplots on the right). Box plots show the median (line), upper and lower quartiles (box) ±1.5x interquartile range (whiskers), outliers are not shown. Two-sided Wilcoxon test: *pval < 0.05; n.s = pval > 0.05 (not significant). Although PH binding is significantly stronger at TSSs of reversible genes in control conditions, this small difference is not reflected in significant changes in the H3K27me3 and H2AK118Ub repressive marks. Binding is strongly reduced in constant depletion but it is restored after a transient depletion. **b-** The PH peaks, the H3K27me3 (PRC2-deposited repressive mark) and H2AK118Ub (PRC1-deposited repressive mark) domains overlaps are shown, after no *ph*-KD (control), constant or transient *ph*-KD. Each vertical bar corresponds to an intersection (the corresponding conditions are shown below) and the horizontal bars (bottom left) indicate the total number of peaks/domains detected in each sample. To avoid weak and noisy peaks/domains, we focused on domains containing at least one PH peak and on PH peaks overlapping H3K27me3 domains in control sample. **c-** Differential analysis of H2AK118Ub domains that show unaffected (gray), decreased (blue) or increased (orange) enrichment upon constant (top) or transient *ph*-KD (bottom). **d-** Each bar corresponds to an H3K27me3 domain containing at least one irreversible (pink) or reversible (green) gene. For each domain, the number of irreversible (pink), reversible (green) and unaffected genes (gray) are shown. Generally, domains containing reversible genes do not contain irreversible ones and *vice versa*. **e-** Average PH ChIP-Seq signal around PH peak summits (x-axis) after no *ph*-KD (top), constant (middle) or transient *ph*-KD (top). PH peaks were stratified based on the closest TSS with a maximum of 25 kb distance. Peaks assigned to irreversible and reversible peaks are shown in pink and in green, respectively. For each condition, the average signal is shown (solid line) ± standard error (shaded area). The distance to the TSS is shown on the x-axis. The signal was quantified at the regions highlighted by dashed lines (see corresponding boxplots on the right). Box plots show the median (line), upper and lower quartiles (box) ±1.5x interquartile range (whiskers), outliers are not shown. Two-sided Wilcoxon test: n.s = pval > 0.05 (not significant).

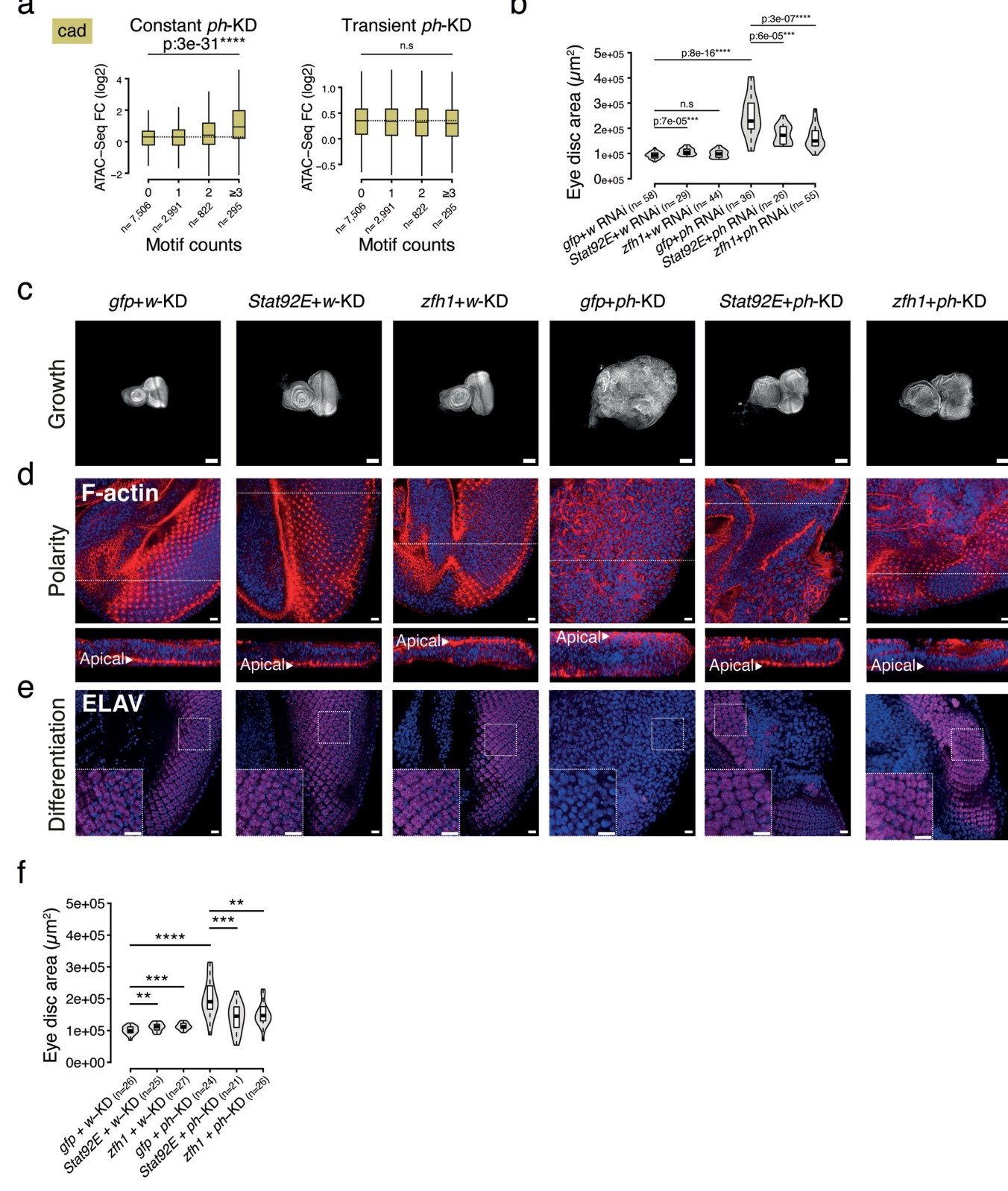

**Extended Data Fig. 5** | See next page for caption.

**Extended Data Fig. 5 | Analysis of transcription factors in EICs shows that transient *Stat92E*-KD and *zfh1*-KD are sufficient to substantially rescue transient *ph*-KD neoplastic signatures. a-** Fold changes at ATAC-Seq peaks (y-axis) upon constant (left) or transient (right) *ph*-KD, as a function of the number of *caudal* (*cad*) motifs they contain (x-axis). Box plots show the median (line), upper and lower quartiles (box) ±1.5x interquartile range (whiskers), outliers are not shown. Two-sided Wilcoxon test: ****pval < 1e-5, n.s = pval>0.05 (not significant). **b-** ED areas upon constant depletion of STAT92E and ZFH1, alone or in addition to PH depletion. Areas are measured using DAPI-stained tissues (number of measured EDs is reported in brackets). Box plots show the median (line), upper and lower quartiles (box) ±1.5x interquartile range (whiskers), outliers are not shown. **c-e** DAPI (gray, **c**), F-actin (red, **d**) and ELAV (magenta, **e**) stainings of EDs after transient *gfp*-KD (control), *Stat92E*-KD and *zfh1*-KD in the presence of a concomitant, transient depletion of *white* (*w*-KD, control, first three columns) or *ph* (*ph*-KD, last three columns). DAPI staining was used to assess growth, F-actin staining was used for apico-basal polarity and the neuronal marker ELAV for differentiation. Two independent experiments were performed with similar results. **f-** ED sizes quantified as overall DAPI staining area for different conditions, showing that transient *Stat92E*-KD or *zfh1*-KD decreased ED overgrowth associated with transient *ph*-KD. Box plots show the median (line), upper and lower quartiles (box) ±1.5x interquartile range (whiskers), outliers are not shown. Two-sided Wilcoxon test: **pval < 1e-2, ***pval < 1e-3, ****pval < 1e-5. Scale bars: 100 µm (**c**), 10 µm (**d**, **e**).

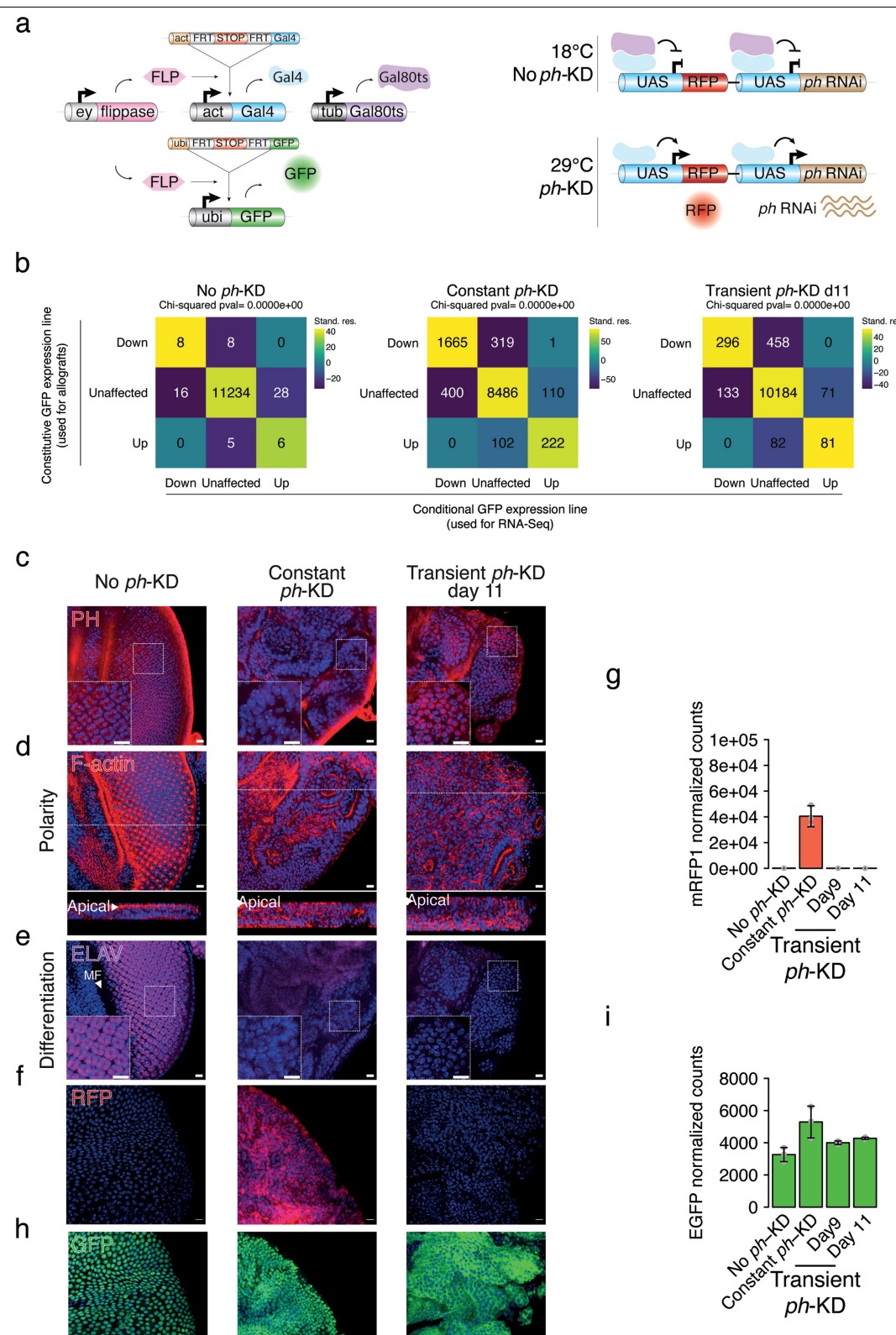

**Extended Data Fig. 6** | See next page for caption.

**Extended Data Fig. 6 | Description and validation of the conditional genetic tool allowing long-term tracking of cells subjected to constant or transient _ph_-KD. a-** Schematic overview of the thermosensitive _ph_-RNAi genetic system used in allograft experiments. Unlike the system described in Fig. 1a (conditional GFP expression), this one ubiquitously expresses GFP under the control of the Ubi-p63E promoter (constitutive GFP expression), while RFP expression is a readout of ongoing RNAi KD. **b-** Comparison of differentially expressed genes after no _ph_-KD (control), constant or transient _ph_-KD between the two genetic systems, allowing either the conditional (x-axis) or the constitutive (y-axis) expression of GFP. For each intersection, the corresponding number of genes is indicated (see numbers). Similarity between the two systems was assessed using chi-squared tests (see p.values on top) and chi-squared standardized residuals are shown using heat maps' colour code. The more an intersection exceeds the size that would be expected by chance, the higher the standard residuals. **c-f-** PH (in red, c), F-actin (in red, d), ELAV (in magenta, e) and RFP (in red, f) stainings of EDs after no _ph_-KD (control, left), constant (middle) or transient _ph_-KD (right), respectively. PH staining was used to visualize PH loss and recovery under constant and transient conditions. F-actin is used to analyse apico-basal polarity, the neuronal marker ELAV for differentiation and RFP as a conditional marker for induction of the RNAi system. The tissues were counterstained with DAPI (blue). Two independent experiments were performed with similar results. **g-** Normalized read counts of mRFP1 mRNAs after no _ph_-KD (control), constant or transient _ph_-KD, showing that transcriptional expression of RFP occurs at constant 29 °C exposure but returns to basal levels after transient _ph_-KD. For each condition, mean normalized read counts ±standard deviation (whiskers) were inferred from three biological replicates of RNA-Seq (grey dots). **h-** GFP staining (in green) after no _ph_-KD (control), constant or transient _ph_-KD. GFP is constitutively expressed after transient _ph_-KD. The tissues were counterstained with DAPI (in blue). **i-** Normalized read counts of GFP mRNAs after no _ph_-KD (control), constant or transient _ph_-KD, showing that GFP expression is irreversibly induced after transient _ph_-KD. For each condition, mean normalized read counts ±standard deviation (whiskers) were inferred from three biological replicates of RNA-Seq (grey dots). Scale bars: 10 μm (c, d, e, f, h).

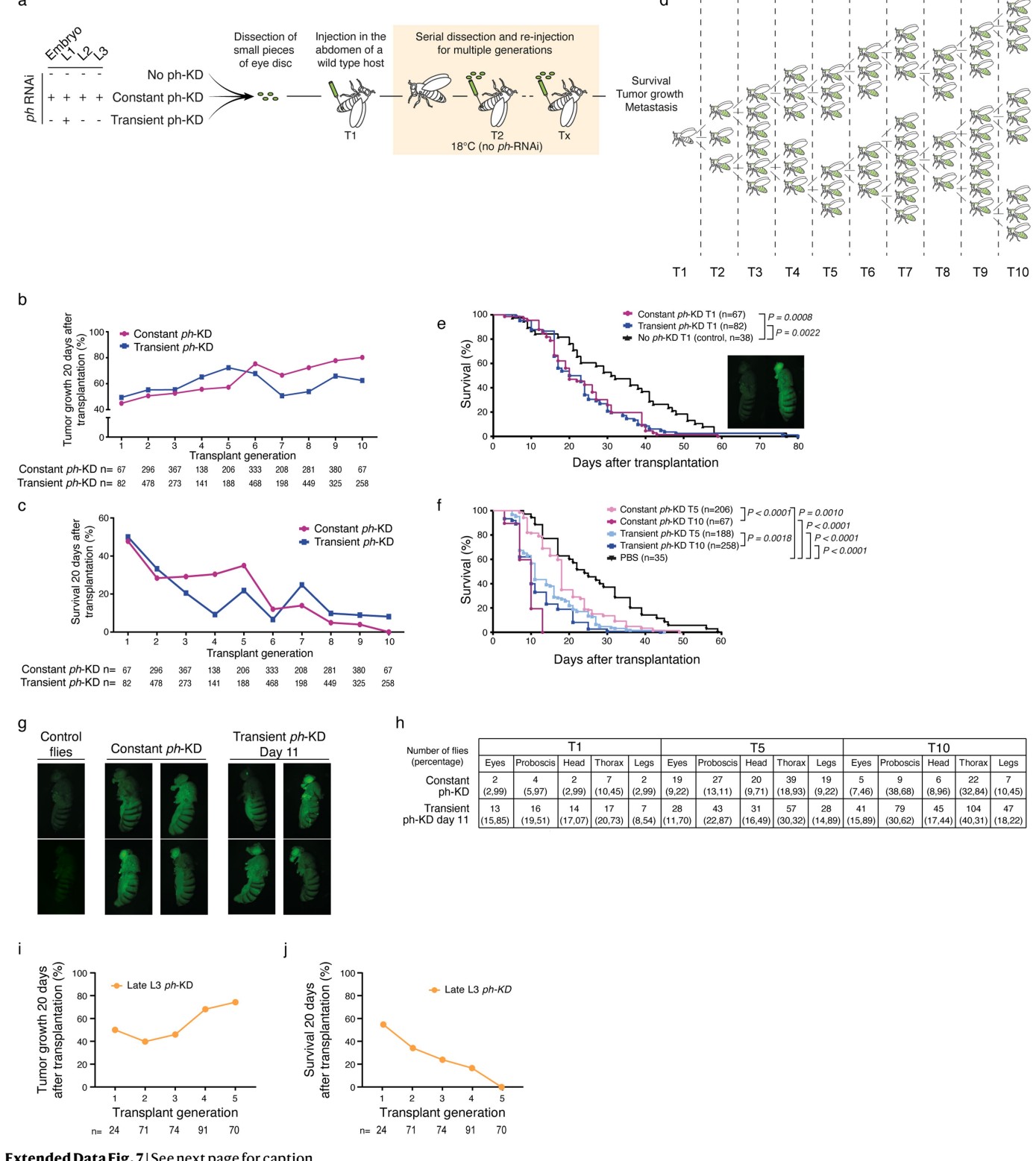

**Extended Data Fig. 7** | See next page for caption.

Extended Data Fig. 7 | Comparative analysis of tumour growth by serial transplantation of constant and transient *ph*-KD EICs. a- Schematic overview of the experimental allograft workflow. Flies of the same genotype were subjected to no *ph*-KD (control, 18 °C), constant *ph*-KD (29 °C) or transient *ph*-KD (24h *ph*-KD at 29 °C during L1 stage). L3 EDs expressing constitutive GFP were dissected from donor larvae and repeatedly allografted into the abdomen of host flies for 10 consecutive rounds until T10 of transplantation (T10≈3 months). All allograft experiments were performed at 18 °C to avoid *ph*-RNAi expression after transplantation. b-c- Tumour growth measured as the percentage of flies showing tumour progression 20 days after transplantation (b) or surviving 20 days after each allograft (c) constant (purple) or transient (blue) *ph*-KD tumours for 10 rounds of transplantation (x-axis). d- Tree representation of the allograft assay. A primary ED tumour derived from constant or transient *ph*-KD is dissected from L3 donor larvae and repeatedly allografted into the abdomen of a female host maintained at 18 °C to prevent re-expression of *ph*-RNAi. Each injected fly is monitored every two days. When the host fly abdomen is completely filled with GFP positive cells, the host is dissected and the tumour cells are injected again into multiple hosts. The procedure was repeated until the tenth generation (T10). e- Host lifespan (x-axis) after the first transplantation (T1) of control (no *ph*-KD, in black), constant (in purple) or transient (in blue) *ph*-KD tumours. Statistical significance was assessed using log-rank test. f- Host lifespan (x-axis) after the fifth (T5) and the tenth (T10) rounds of transplantations of constant (T5 in pink, T10 in purple) or transient (T5 in light blue, T10 in blue) *ph*-KD tumours. Since control (no *ph*-KD) tissues do not grow and cannot be serially transplanted, PBS injections were used as control (in black). Statistical significance was assessed using log-rank test. g- Flies injected with dissected grafts after no *ph*-KD (control), constant or transient *ph*-KD. Only primary tumours generated by constant or transient *ph*-KD can invade the abdomen and surrounding tissues but not EDs resulting from no *ph*-KD (control) conditions. h- In order to score the frequency of metastases, the injected flies were monitored twice a week and the appearance of metastases in the thorax, head, proboscis, eyes and legs were noted for each generation. Values in the table represent the number and percentage (in brackets) of flies with metastases after the 1st, 5th or 10th round of transplantation (T1, T5 and T10, respectively) of constant or transient *ph*-KD tumours. i-j- Tumour growth measured as the percentage of flies showing tumour progression 20 days after transplantation (d) and host fly survival 20 days after allograft of late L3 *ph*-KD (e) for 5 consecutive rounds of transplantation.

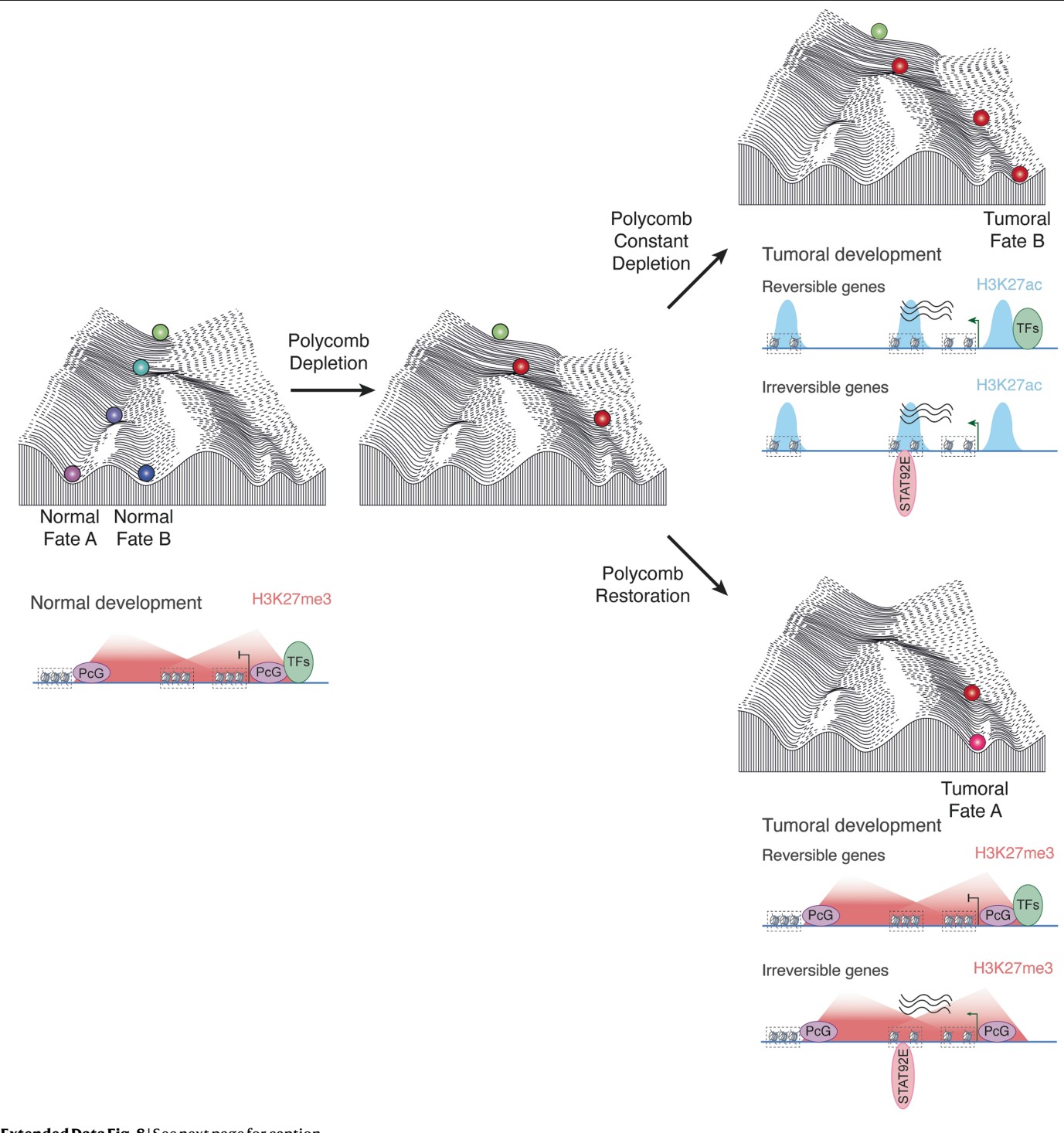

**Extended Data Fig. 8** | See next page for caption.

**Extended Data Fig. 8 | A model explaining the emergence of epigenetically initiated cancers.** The model is based on the well-known Waddington landscape depicting a marble rolling down a slope with multiple choices of trajectories that depend on the hills and valleys encountered on their path. This scheme is a metaphor for the multiple possible cell fates that can arise from a single cell representing the zygote and is frequently used to signify that epigenetic inheritance contributes to the stable transmission of cell fates, once they are determined by intrinsic and extrinsic signals. In the context of this work, we posit that Polycomb components contribute to shaping the landscape and allow for multiple normal cell fates to be established and transmitted through the developmental process. In normal development, the cells (in green) at the top of the hill will move down during differentiation in order to acquire normal fates (left panel). Upon depletion of a Polycomb component, such as the PRC1 subunits PH or PSC, the landscape is modified (center panel). If depletion is stably maintained, the modified landscape forces cells to take a path that is both aberrant and intrinsically stable, inducing cancer formation through loss of cell differentiation, loss of cell polarity and sustained proliferation (upper right panel). If Polycomb protein levels are restored, the landscape returns to its original shape. However, if restoration of the landscape occurs after cells have already chosen an aberrant route (represented by the marble in the middle of the landscape), they will no longer be able to find the healthy trajectory and will be obliged to choose from a limited set of possibilities in a diseased cell space. This may ultimately lead to the maintenance of tumour phenotypes. In addition to the Waddington landscape panels, gene panels are added, representing a putative molecular explanation for the phenomenon described here. The chromatin and functional state of reversible and irreversible genes are shown in each condition. In a physiological condition (left), both categories of genes are bound by Polycomb components and are decorated by repressive histone marks, such as H3K27me3. Upon depletion of Polycomb components such as PH, both the Polycomb complexes and their histone marks are lost and Polycomb target genes acquire active histone mark such as H3K27ac and become transcribed. At irreversible genes, transcriptional activation is dependent on the JAK–STAT signaling pathway transcription factor STAT92E (top right). Upon PRC1 re-establishment, the repressive mark and PH binding is globally recovered. However, chromatin stays open at specific sites that regulate irreversible genes, in which a DNA motif bound by the main JAK–STAT effector STAT92E is enriched. STAT92E target genes include proliferation components and *zfh1*, which encodes a transcription factor that represses transcription of a set of genes involved in cell differentiation. The combined, self-sustaining induction of cell proliferation and loss of differentiation induces tumorigenesis even after restoration of normal levels of Polycomb proteins on their target chromatin (bottom right, see also Fig. 5g).

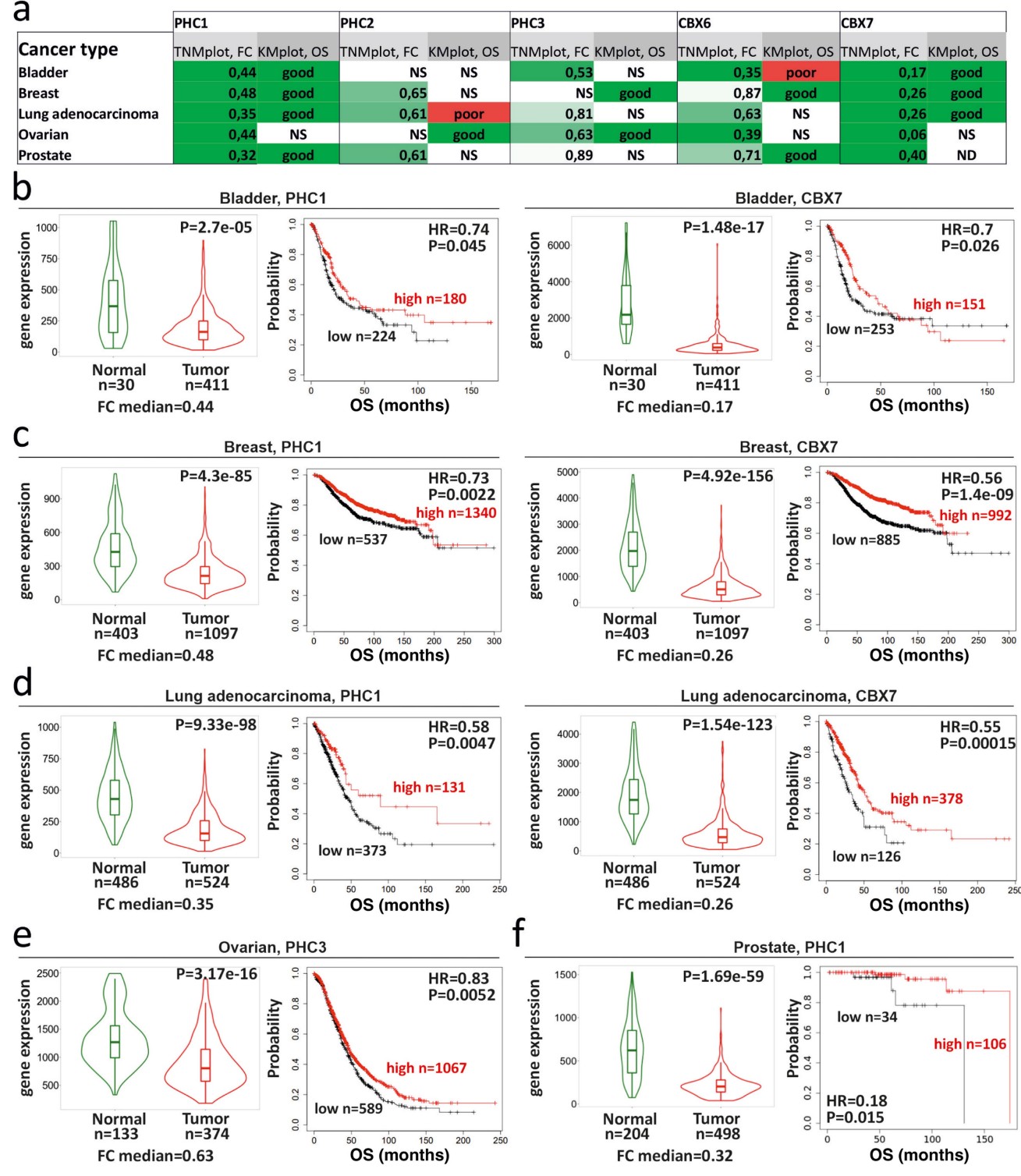

Extended Data Fig. 9 | See next page for caption.

**Extended Data Fig. 9 | Examples of the tumour suppressive role of canonical PRC1 core subunits in different cancer types. a-** Clinical correlations for PRC1 in selected cancer types. Differential gene expression (TNMplot) and clinical prognosis Kaplan-Meier plot (KMplot) results are given for *PHC1, PHC2, PHC3, CBX6, CBX7* genes. TNMplot columns represent the differential gene expression analysis in tumour and matched normal tissues, which was performed using the https://tnmplot.com/ online tool. FC median: Fold change median. Statistical significance was calculated using a two-sided Mann-Whitney U test with a significance level of 0.01. NS – non-significant Mann–Whitney p-value. Green boxes indicate that gene expression is significantly lower in tumour tissues. KMplot columns show the analysis of correlation between overall survival (OS) and levels of gene expression. KMplot analysis was performed using the https://kmplot.com/ online tool. Statistical significance was calculated by a two-sided Cox regression test with a significance level of 0.05. NS – non-significant logrank *p*-values. Green boxes ("Good") indicate cases in which high expression of PRC1 genes in tumours is associated with a better overall patient survival. **b-** Clinical prognosis for PHC1 (left) and CBX7 (right) in bladder cancer. For each gene the TNMplot (Violin plots, left panels) and KM plots (right panels) are shown. **c-** Clinical prognosis of PHC1 (left) and CBX7 (right) in breast cancer. For each gene the TNMplot (Violin plots, left panels) and KM plots (right panels) are shown. **d-** Clinical prognosis of PHC1 (left) and CBX7 (right) in lung adenocarcinoma. For each gene the TNMplot (Violin plots, left panels) and KM plots (right panels) are shown. **e–f** Clinical prognosis of PHC3 (**e**) in ovarian and PHC1 (**f**) in prostate cancer. For each gene the TNMplot (Violin plots, left panels) and KM plots (right panels) are shown. The Violin plots display the range of values from the minimum to the maximum value, with the box representing the values from the first quartile to the third quartile. The median is indicated by the thick line in the center, and the width of the plot, or density, reflects the frequency of the samples. In KMplots, the cohort with low gene expression level is coloured black and the cohort with high gene expression is coloured red. HR: hazard ratio.

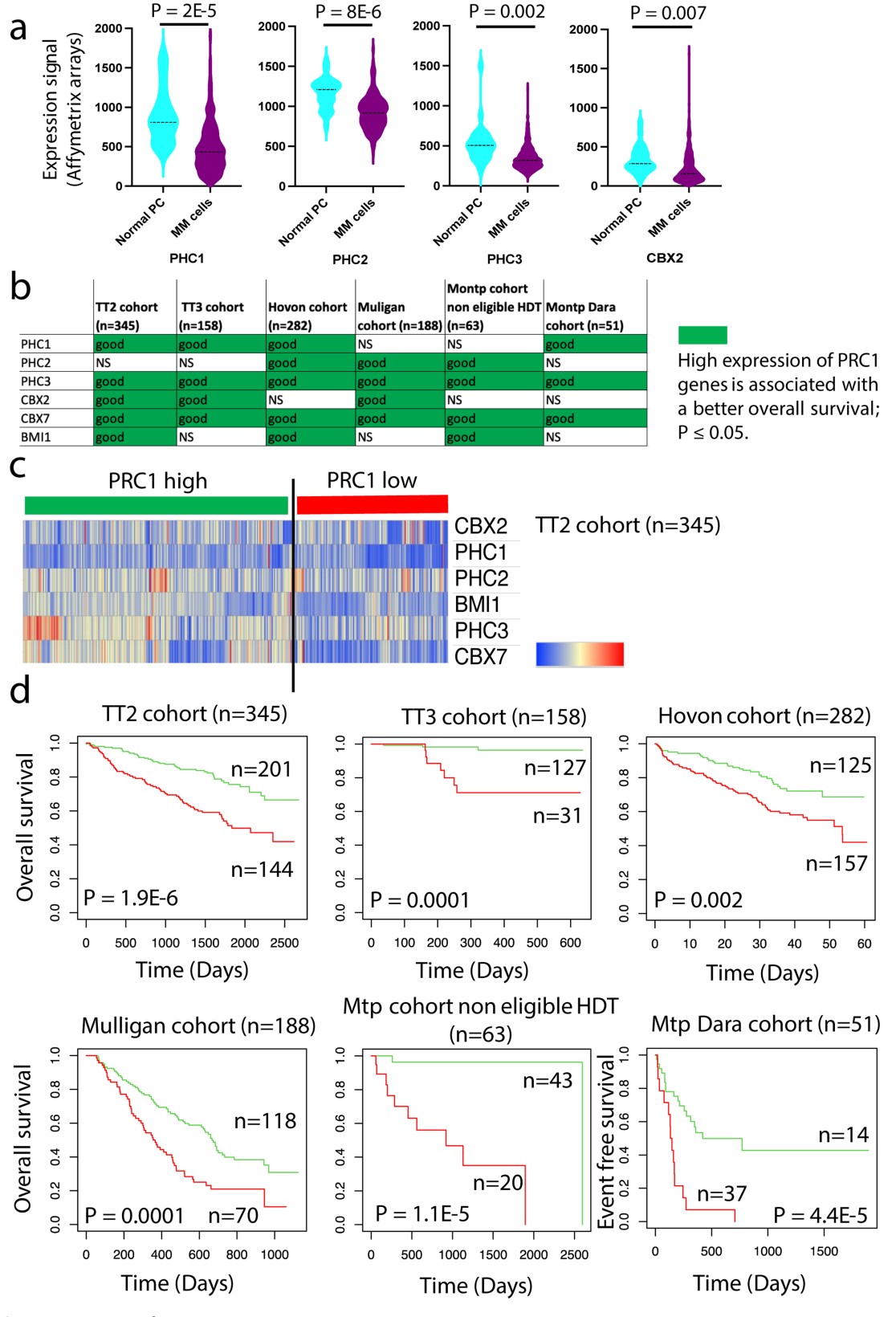

**Extended Data Fig. 10** | See next page for caption.

**Extended Data Fig. 10 | Tumour suppressive role of core PRC1 subunits in Multiple Myeloma. a-** *PHC1, PHC2, PHC3* and *CBX2* gene expression is significantly downregulated in malignant plasma cells (PCs) from patients with Multiple Myeloma (MM cells) compared to normal bone marrow PCs. Affymetrix U133 P gene expression profiles of purified bone marrow PC from 22 healthy donors and purified myeloma PCs from 345 previously untreated patients were compared using publicly available data (Gene Expression Omnibus, accession number GSE2658) from the University of Arkansas for Medical Sciences (UAMS, Little Rock, AR). Statistical difference was assayed using a two-sided Student t test. **b-** Prognostic value of core PRC1 components in MM. The prognostic value of PHC1, PHC2, PHC3, CBX2, CBX7, and BMI1 gene expression was analyzed in 6 independent cohorts of patients with MM using the Maxstat R function and Kaplan Meier survival curves as previously described. Low expression of *PHC1, PHC2, PHC3, CBX2, CBX7* and *BMI1* was associated with significantly shorter overall survival in at least three independent cohorts of MM patients out of the six studied (green colour). The six cohorts included gene expression data of purified MM cells from the TT2, TT3 (accession number E-TABM-1138; GSE2658),

and Hovon (accession number GSE19784) cohorts (345, 158 and 282 newly-diagnosed MM patients treated by high-dose melphalan and autologous hematopoietic stem cell transplantation); the Mulligan cohort (188 patients at relapse treated by proteasome inhibitor in monotherapy; GSE9782); the Mtp cohort non eligible to HDT (63 newly-diagnosed MM patients non eligible to high-dose melphalan and autologous hematopoietic stem cell transplantation) and the Mtp Dara cohort (51 patients at relapse treated by anti-CD38 monoclonal antibody (Daratumumab). **c-** The prognostic information of *PHC1, PHC2, PHC3, CBX2, CBX7* and *BMI1* genes was combined. Patients of the TT2 cohort (n = 345) were ranked according to the increased value of the calculated score and a cluster was defined. **d-** In the TT2 cohort, a maximum difference in overall survival was obtained, using the Maxstat R package, splitting patients into high-risk for 144 patients with the lowest expression of PRC1 genes and low-risk group for the 201 patients with higher PRC1 gene expression. Using the same parameter of the TT2 training cohort, we validated the association between low expression of PRC1 genes and a poor outcome in five other independent cohorts of patients with MM.

# Reporting Summary

## Statistics

For all statistical analyses, confirm that the following items are present in the figure legend, table legend, main text, or Methods section.

| n/a | Confirmed | |
|---|---|---|
| ☐ | ☒ | The exact sample size (*n*) for each experimental group/condition, given as a discrete number and unit of measurement |
| ☐ | ☒ | A statement on whether measurements were taken from distinct samples or whether the same sample was measured repeatedly |
| ☐ | ☒ | The statistical test(s) used AND whether they are one- or two-sided<br>*Only common tests should be described solely by name; describe more complex techniques in the Methods section.* |
| ☒ | ☐ | A description of all covariates tested |
| ☐ | ☒ | A description of any assumptions or corrections, such as tests of normality and adjustment for multiple comparisons |
| ☐ | ☒ | A full description of the statistical parameters including central tendency (e.g. means) or other basic estimates (e.g. regression coefficient) AND variation (e.g. standard deviation) or associated estimates of uncertainty (e.g. confidence intervals) |
| ☐ | ☒ | For null hypothesis testing, the test statistic (e.g. *F*, *t*, *r*) with confidence intervals, effect sizes, degrees of freedom and *P* value noted<br>*Give P values as exact values whenever suitable.* |
| ☒ | ☐ | For Bayesian analysis, information on the choice of priors and Markov chain Monte Carlo settings |
| ☐ | ☒ | For hierarchical and complex designs, identification of the appropriate level for tests and full reporting of outcomes |
| ☐ | ☒ | Estimates of effect sizes (e.g. Cohen's *d*, Pearson's *r*), indicating how they were calculated |

*Our web collection on statistics for biologists contains articles on many of the points above.*

## Software and code

Policy information about availability of computer code

| Data collection | All immunostaining images were acquired using a Leica SP8 confocal microscope. Images of EdU stained-eye discs were acquired using a Zeiss LSM980 Airyscan microscope in 4Y modality. RT-qPCR were performed using LightCycler480. Western Blot images were acquired using the ChemiDoc Imaging Systems, Bio-Rad. To score tumor progression in allografts, flies were imaged every two days using Leica MZ FLIII in order to verify GFP as a readout of tumor growth. Injected Drosophila pictures were taken using Ximea USB 3.1 Gen1 camera with a Sony CMOS-xiCAll sensor. NGS illumina sequencing was performed on a Novaseq 6000. |
|---|---|
| Data analysis | Immunostaining were analysed using Fiji version 1.54h. Airyscan images of EdU stained-eye discs were processed with ZEN (version 3.6 Blue Edition, Zeiss) using default settings. Western Blot gels were analysed using ImageLab software version 6.1 from Bio-Rad. EdU movies were created using Imaris (version 10.1, Oxford Instruments). RT-qPCR were analysed using LightCycler software version 1.5.1. Alignments were performed using Burrows-Wheeler Aligner (BWA, gDNA), bowtie2 (CUT&RUN, ChIP-Seq, ATAC-Seq, v2.3.5.1) and the Rsubread R package (v2.0.1). Peak calling was performed using MACS2 (v2.2.7.1). Differential analysis of gene expression was performed using the DESeq2 R package (v1.26.0). Motif analysis was performed using the i-cisTarget online tool (database v.6.0). Genomic DNA variants were called using the GATK software. For Multiple Myeloma meta analysis, clustering was performed using the Morpheus software and violin plots using GraphPad Prism. Difference in overall survival between groups of patients was assayed with a log-rank test and survival curves plotted using the Kaplan-Meier method (Maxstat R Package). All other in-house bioinformatic analyses of Drosophila data were performed in R version 4.2.0, and full custom code were made publicly available at https://github.com/vloubiere/Parreno_Loubiere_2023. |

For manuscripts utilizing custom algorithms or software that are central to the research but not yet described in published literature, software must be made available to editors and reviewers. We strongly encourage code deposition in a community repository (e.g. GitHub). See the Nature Portfolio guidelines for submitting code & software for further information.

## Data

Policy information about availability of data

All manuscripts must include a data availability statement. This statement should provide the following information, where applicable:
- Accession codes, unique identifiers, or web links for publicly available datasets
- A description of any restrictions on data availability
- For clinical datasets or third party data, please ensure that the statement adheres to our policy

All genomic data produced for this manuscript have been deposited on GEO repository under the public accession number GSE222193. All other data were retrieved from publicly available data as indicated.

## Human research participants

Policy information about studies involving human research participants and Sex and Gender in Research.

| Reporting on sex and gender | Not Applicable |
| --- | --- |
| Population characteristics | Not Applicable |
| Recruitment | Not Applicable |
| Ethics oversight | Not Applicable |

Note that full information on the approval of the study protocol must also be provided in the manuscript.

# Field-specific reporting

Please select the one below that is the best fit for your research. If you are not sure, read the appropriate sections before making your selection.

☒ Life sciences  ☐ Behavioural & social sciences  ☐ Ecological, evolutionary & environmental sciences

For a reference copy of the document with all sections, see nature.com/documents/nr-reporting-summary-flat.pdf

# Life sciences study design

All studies must disclose on these points even when the disclosure is negative.

| Sample size | Sample size was defined in compliance with the gold standards of the field, such that relevant statistical parameters (mean, median...) would get stabilized. ChIP-Seq, CUT&RUN and ATAC-Seq were performed in duplicates, following encode's standards (https://www.encodeproject.org/chip-seq/transcription_factor/#standards; https://www.encodeproject.org/atac-seq/#standards). RNA-Seq were performed in triplicates, following encode's recommandations (https://www.encodeproject.org/data-standards/rna-seq/long-rnas/). For sample sizes of Immunostaining experiments, see Supplementary Table 6, sheet name "All IF sample numbers". For the patients cohorts discussed in the extended data Fig 12, the publicly available TT2 cohort (n=345) enables detection of transcriptional patterns associated with therapy resistance and overall survival in Multiple Myeloma. Furthermore, five other independent cohorts of patients with Multiple Myeloma (n=158, n=282, n=188, n=63 and n=51) were used as validation datasets. The original papers describing each cohort, including discussions on the sample sizes are provided in the legends relative to extended data. |
| --- | --- |
| Data exclusions | For transcriptome differential expression analysis, only the genes with at least 10 counts across all tested conditions were retained, complying with good practices. Similarly, only the genes that were differentially expressed in any of the tested condition were retained for the clustering of differentially expressed genes. For ATAC-Seq data, only the peaks overlapping at least 100 reads across all tested condition were considered, to avoid weak or noisy peaks. Only the peaks showing significant changes in accessibility during tested condition and had a log10(baseMean) value bigger or equal to 1.25 were considered for clustering. |
| Replication | All experiments were performed several times on different days, and only consistent observations were reported (see replicates section for further details regarding biological replicates). For transcriptomic, RT-qPCR and western blot analysis, experiments were performed in biological triplicates. ATAC-seq, Cut&RUN, ChIP-seq and immunostaining experiments were performed in biological duplicates. Each biological replicate was obtained from independent genetic crosses. The only exception was the phospho H2AV staining shown in Figure 1j and Extended Fig.2c, which was performed once, but scoring tissues coming from six independent genetic crosses. This is now indicated in the methods section of the manuscript. |
| Randomization | This study does not require proper randomization protocols. However, collected flies were always randomly selected to perform the different experiments. |
| Blinding | Blinding is not applicable here, since data anonymization is not compatible with quality controls and the identity of control samples must be known in order to perform genomic data analysis. However, reported data are based on unbiased analysis avoiding confirmation bias and/or |

# Reporting for specific materials, systems and methods

We require information from authors about some types of materials, experimental systems and methods used in many studies. Here, indicate whether each material, system or method listed is relevant to your study. If you are not sure if a list item applies to your research, read the appropriate section before selecting a response.

## Materials & experimental systems

| n/a | Involved in the study |
|---|---|
| ☐ | ☒ Antibodies |
| ☒ | ☐ Eukaryotic cell lines |
| ☒ | ☐ Palaeontology and archaeology |
| ☐ | ☒ Animals and other organisms |
| ☒ | ☐ Clinical data |
| ☒ | ☐ Dual use research of concern |

## Methods

| n/a | Involved in the study |
|---|---|
| ☐ | ☒ ChIP-seq |
| ☒ | ☐ Flow cytometry |
| ☒ | ☐ MRI-based neuroimaging |

## Antibodies

**Antibodies used**

Antibodies are described in the Methods section of the manuscript.

The following primary antibodies were used (Antibody, dilution, Provider, Catalogue number) for immunostaining:
- PH, 1:500,
- ELAV, 1:1000, DSHB, 9F8A9,
- ABD-B, 1:1000, DSHB, 1A2E9,
- GFP, 1:500, Invitrogen, A10262
- ZFH1, 1:2000
- Histone H2AvD pS137, 1:500, Rockland, 600-401-914
- Rhodamine phalloidin Alexa Fluor 555, 1:1000, Invitrogen, R415
- Rhodamine phalloidin Alexa Fluor 488, 1:1000, Invitrogen, A12379

The following secondary antibodies were used (Antibody, dilution, Provider, Catalogue number) for immunostaining:
- donkey anti-goat Alexa Fluor 555, 1:1000, Invitrogen, A-21432
- donkey anti-mouse Alexa Fluor 647, 1:1000, Invitrogen, A-31571
- donkey anti-chicken Alexa Fluor 488, 1:1000, Clinisciences, 703-546-155
- donkey anti-rabbit Alexa Fluor 555, 1:1000, Invitrogen, A-31572

The folowing antibodies were used (Antibody, dilution, Provider, Catalogue number) for western blot:
- PH, 1:200, gift from Renato Paro
- ZFH1, 1:2000, gift from Erika Bach
- Beta-tubulin, 1:5000, DSHB, AA12.1

The following HRP-conjugated secondary antibodies were used (Antibody, dilution, Provider, Catalogue number) for western blot:
- goat anti-rabbit, 1:15000, Sigma, A0545
- rabbit anti-mouse, 1:15000, Sigma, A9044

The following antibodies were used (Antibody, dilution, Provider, Catalogue number) for ChIP-seq:
- PH, 1:100, Giacomo Cavalli laboratory

The following antibodies were used (Antibody, dilution, Provider, Catalogue number) for CUT&RUN:
- H3K27me3, 1:100, Active motif, 39155
- H2AK118Ub, 1:100, Cell Signaling Technology, 8240S
- H3K27Ac, 1:100, Active motif, 39133
- IgG, 1:100, Cell Signaling Technology, 2729S

**Validation**

All commercial antibodies are validated for the use of immunofluorescence, western blot, ChIP-seq and CUT&RUN. Data are available on the manufacturer's website. The following antibodies have been validated by the manufacturer, by previous papers and/or in this paper by RNAi knockdown (Antibody, Validation and references):
- PH, immunostaining, PMID: 16530043, ChIP-seq PMID: 27643538, western blot validated in this paper by RNAi knockdown
- ELAV, immunotaining, Elav-9F8A9 was deposited to the DSHB by Rubin, Gerald M. (DSHB Hybridoma Product Elav-9F8A9), PMID: 8033205
- ABD-B, immunostaining, anti-ABD-B (1A2E9) was deposited to the DSHB by Celniker, S. (DSHB Hybridoma Product anti-ABD-B (1A2E9)), PMID: 2575066
- GFP, immunostaining, specificity of chicken-anti-GFP (#A10262, Invitrogen) has been verified in this paper by verifying that it produces signal at the expected molecular weight and only in fly lines and experimental conditions that lead to the presence of GFP.
- ZFH1, immunostaining PMID: 20412771
- Histone H2AvD pS137, validated in PMID: PMID: 29925946 for immunostaining, see also manufacturer's website for references
- Rhodamine phalloidin, immunostaining see manufacturer's website for references

- Beta-tubulin, AA12.1 was deposited to the DSHB by Walsh, C. (DSHB Hybridoma Product AA12.1), PMID: 6363422
- H3K27me3, CUT&RUN see manufacturer's website for references. Additional validation is provided in this manuscript, by verifying that CUT&RUN signal is lost upon knock down of the Polycomb component PH.
- H3K27Ac, CUT&RUN see manufacturer's website for references. Additional validation is provided in this manuscript, by verifying that CUT&RUN signal is gained at many Polycomb target genes that are induced upon knock down of the Polycomb component PH.
- H2AK118ub, CUT&RUN see manufacturer's website for references. Additional validation is provided in this manuscript, by verifying that CUT&RUN signal is lost upon knock down of the Polycomb component PH.

# Animals and other research organisms

Policy information about studies involving animals; ARRIVE guidelines recommended for reporting animal research, and Sex and Gender in Research

| Laboratory animals | For crosses adults were between 2-3 days old when crosses were performed. Larvae were between 9 to 11 days.<br>The following flies strain were used in this study :<br>eyFLP, Act5C (FRT.CD2) Gal4 ; + ; UAS-GFP (BL#64095), Tub-Gal80TS ; TM2/TM6B,Tb (BDSC#7019), UAS-ph RNAi (VDRC#50028), UAS-Psc RNAi (BL#38261), UAS-Su(z)2 RNAi (VDRC#100096), Sna[Sco] / CyO ; Tub-Gal80TS (BL#7018), UAS-white RNAi (BL#33623), UAS-gfp RNAi (BL#9331), UAS-zfh1 RNAi (VDRC#103205), UAS-Stat92E RNAi (VDRC#43866), ey-FLP (BL#5580), Ubi-p63E(FRT.STOP)Stinger (BL#32249), Act5C(FRT.CD2) Gal4 , UAS-RFP/TM3,Sb (BL#30558), His2Av-mRFP (BL#23650) |
|---|---|
| Wild animals | The study did not involve wild animals. |
| Reporting on sex | In order to standardize expression of the RNAi constructs, for all crosses, we used virgin females for all lines expressing GAL4.<br>All Drosophila analyzed in the study were females in order to avoid sex biases. Of note, male flies always showed similar phenotypes. |
| Field-collected samples | The study did not involve samples collected from the field. |
| Ethics oversight | This study was performed under the ethical approval N. n6906C2 of the MINISTÈRE DE L' ENSEIGNEMENT SUPÉRIEUR, DE LA RECHERCHE ET DE L' INNOVATION, issued on April 8, 2020. Drosophila strains used and the age of the populations are reported in the manuscript and the methods section. |

Note that full information on the approval of the study protocol must also be provided in the manuscript.

# ChIP-seq

## Data deposition

☒ Confirm that both raw and final processed data have been deposited in a public database such as GEO.

☒ Confirm that you have deposited or provided access to graph files (e.g. BED files) for the called peaks.

| Data access links<br>*May remain private before publication.* | GSE222193 |
|---|---|
| Files in database submission | For each ChIP-Seq data, bw genomic tracks and peak files are available. For each ATAC-Seq data, bw genomic tracks and peak files are available. For RNA-Seq, bw tracks, normalized counts and differential expression files are available. For gDNA sequencing, all raw SNPs and InDels as well as somatic variants are available. Finally, raw fastq files are available for all experiments. |
| Genome browser session<br>(e.g. UCSC) | A UCSC browser session is available at http://genome-euro.ucsc.edu/s/cavalli/EpiCancer and is mentioned in the Data availability Heading in the manuscript. Furthermore, at: https://www.ncbi.nlm.nih.gov/geo/query/acc.cgi?&acc=GSE222193 users can very easily download all bigwig (BW) format files that can the then simply dispayed in local genome browsers such as IGV or IGB that can be downloaded for free and used on any desktop computer. |

## Methodology

| Replicates | For ChIP-Seq and Cut&Run data, two replicates were used per condition. |
|---|---|
| Sequencing depth | Condition Total reads Uniquely aligned reads<br>H2AK118Ub_PH18_rep1 5,461,997 3,491,769<br>H2AK118Ub_PH18_rep2 10,670,328 6,534,683<br>H2AK118Ub_PH29_rep1 16,931,039 5,085,514<br>H2AK118Ub_PH29_rep2 7,688,530 5,173,802<br>H2AK118Ub_PHD11_rep1 8,223,640 6,108,599<br>H2AK118Ub_PHD11_rep2 8,526,156 6,702,363<br>H2AK118Ub_PHD9_rep1 8,226,806 5,787,468<br>H2AK118Ub_PHD9_rep2 6,836,989 4,212,828<br>H3K27Ac_PH18_rep1 9,502,481 6,423,401<br>H3K27Ac_PH18_rep2 9,455,194 6,557,137<br>H3K27Ac_PH29_rep1 7,933,451 6,020,508<br>H3K27Ac_PH29_rep2 10,928,323 6,094,928 |

H3K27Ac_PHD11_rep1 10,837,602 8,099,891
H3K27Ac_PHD11_rep2 14,222,258 10,293,439
H3K27Ac_PHD9_rep1 11,714,006 7,742,745
H3K27Ac_PHD9_rep2 11,392,314 8,577,451
H3K27me3_PH18_rep1 9,860,407 7,139,641
H3K27me3_PH18_rep2 8,980,660 7,500,962
H3K27me3_PH29_rep1 7,800,260 5,501,836
H3K27me3_PH29_rep2 9,340,779 6,440,368
H3K27me3_PHD11_rep1 9,866,072 6,994,121
H3K27me3_PHD11_rep2 10,130,345 7,451,714
H3K27me3_PHD9_rep1 10,431,545 7,553,174
H3K27me3_PHD9_rep2 9,577,300 5,745,957
IgG_PH18_rep1 9,966,366 5,920,782
IgG_PH18_rep2 10,683,819 4,882,917
IgG_PH29_rep1 10,581,617 4,375,438
IgG_PH29_rep2 12,450,166 6,844,033
IgG_PHD11_rep1 9,081,016 5,772,603
IgG_PHD11_rep2 12,626,730 8,014,611
IgG_PHD9_rep1 8,807,714 4,619,458
IgG_PHD9_rep2 10,756,886 6,785,571
PH_PH18_rep1 11,937,832 7,640,526
PH_PH18_rep2 13,949,018 8,016,532
PH_PH29_rep2 11,382,596 6,293,104
PH_PHD11_rep1 13,554,060 9,296,764
PH_PHD11_rep2 17,309,927 10,419,882
PH_PHD9_rep1 14,598,749 8,370,146
PH_PHD9_rep2 14,144,138 8,201,413
input_PH18_rep1 10,263,837 7,869,566
input_PH18_rep2 14,283,477 10,695,459
input_PH29_rep1 10,939,021 6,402,253
input_PH29_rep2 12,758,267 10,242,311
input_PHD11_rep1 11,380,515 9,000,824
input_PHD11_rep2 15,150,197 12,146,663
input_PHD9_rep1 11,116,399 8,489,153

| Antibodies | The following antibodies were used (Antibody, dilution, Provider, Catalogue number) for ChIP-seq :<br>- PH, 1:100, Giacomo Cavalli laboratory<br><br>The following antibodies were used (Antibody, dilution, Provider, Catalogue number) for CUT&RUN :<br>- H3K27me3, 1:100, Active motif, 39155<br>- H2AK118Ub, 1:100, Cell Signaling Technology, 8240S<br>- H3K27Ac, 1:100, Active motif, 39133<br>- IgG, 1:100, Cell Signaling Technology, 2729S |
|---|---|
| Peak calling parameters | Peak calling was performed using MACS2 using with the following parameters: --keep-dup 1 -g dm -f BAMPE -B --SPMR |
| Data quality | The quality of the data was assessed using PCA on normalized counts and Pearson's correlation coefficients between replicates. Following are the number of significant peaks found for each condition:  H2AK118Ub_PH18 = 728, H2AK118Ub_PH29= 45, H2AK118Ub_PHD11= 217, H2AK118Ub_PHD9= 228, H3K27Ac_PH18=  3836, H3K27Ac_PH29= 4380, H3K27Ac_PHD11= 4871, H3K27Ac_PHD9= 4360, H3K27me3_PH18 = 265, H3K27me3_PH29= 161, H3K27me3_PHD11= 205, H3K27me3_PHD9= 251, PH_PH18= 2855, PH_PH29= 71, PH_PHD11= 2873, PH_PHD9= 2215 |
| Software | Peaks calling was performed using MACS2, and further quantification were performed in R (v4.2.0). All custom code will be made publicly available on github before publication (yet to come). |

