## [Peer Review File · Nature]

Manuscript Title: Transient loss of Polycomb components induces an epigenetic cancer fate

Redactions – unpublished data

Reviewer Comments & Author Rebuttals

Reviewer Reports on the Initial Version:

Referees' comments:

Referee #1 (Remarks to the Author):

This manuscript attempts to address if non-genetic reprogramming is sufficient to induce tumourigenesis which would be an important step forward for the role of epigenetics in tumour initiation. The authors generate a transient RNAi knockdown of PRC1 components and show this is sufficient to induce overgrowth and loss of apico-basal polarity and differentiation markers. They carry out DNA sequencing to explore whether this transient depletion is promoting overgrowth by a genetic mechanism. Transcriptomic analysis of the tumours indicates a cluster of genes that are altered beyond the transient depletion which are associated with cytokine activity and contain direct PcG target genes. Epigenetic analysis explores the restoration of chromatin at altered genes and they propose that transcriptional repressor binding distinguishes irreversibly altered genes. Serial transplantation of the tumours indicates they become more aggressive over time. This is a large body of impressive work but raises a number of important questions and seems somewhat incomplete.

Major comments:

- PH depletion at L3 late does not lead to tumours, perhaps suggesting this requires developmental, non-differentiated tissues and raises the question of relevance to adult tissues? Is transient depletion simply enforcing a developmental state or preventing differentiation during drosophila development rather than inducing an 'epigenetically initiated cancer'?
- In the DNA-sequencing, the authors show a significant amount of mutations (in the thousands) and argue these are not responsible for tumour formation. Where do these mutations come from and what are they relative to? Can they filter out the mutations unique to PH knockdown due to batch effect to really understand which are arising from the knockdown? A simple graph showing absolute number of new mutations arising would provide clarity rather than feature proportions.
- Unclear whether copy number alterations are being induced by transient PH depletion. This could contribute to persistence of tumour formation. This seems an oversight given the importance of copy number alterations in cancer progression.
- Does knockdown of *zfh1* during or even after transient depletion of PH rescue tumour formation, this could provide insight into mechanism.

- Unclear why is the gene set reduced to PH bound genes and how are non-PH target genes being altered, this is not a minority of the geneset for figure 3A/B.
- The authors claim that 'essentially normal chromatin landscape' returns following transient depletion of PH, yet zfh1 H3K27Ac does not return to baseline. This appears to be significant as zfh1 the authors claim is required for the phenotype. This appears an overstatement and would be important to look closely at the 'few exceptions' in detail and why the normal chromatin landscape is not restored. This could provide critical insight into mechanisms downstream of ph that leads to tumours. In the absence of further proof it is unclear that differential transcriptional repressor binding of a small subset is the underlying mechanism.
- In addition, this reviewer would like to see a clearer visualisation/analysis of the relation between the histone marks and gene expression. At present, it is unclear how and if the repressive and activating marks are different between the reversible and irreversible genes, especially in light of the observations on zfh1 as described in the previous point.
- Tumours becoming more aggressive over time could indicate ongoing genomic instability or chromosome instability induced by transient PH depletion.
- Evidence supporting a role of this epigenetic initiation in humans is weak and has been carried out in cancers with widespread genomic driver alterations.

Minor comments:

- Confusing break in axis in extended data figure 1G. In general, breaking of axis do not aid clarity of interpretation.
- Western showing return to normal levels is unclear and quantification of western blot would confirm levels of PH return to baseline.
- Statistical analysis showing over representation of PcG target genes in cluster 2 would avoid potential observed random over representation.
- Lack of follow up on spz3 and Dcr-2 as the evidence these aren't driving a phenotype is lacking.
- It is intriguing that extended data figure 2D is similar to main figure 2E.

Referee #2 (Remarks to the Author):

While genetic mutations in epigenetic regulators are well-known to contribute to cancers, it is not clear whether transient epigenetic changes can lead to cancer development. The authors have addressed this important question using *Drosophila* as a model system. They show that transient depletion of PRC1 is sufficient for tumor development and that the generated tumors are transplantable. They determine the transcriptional changes induced by the transient depletion of PRC1 and define reversible and irreversible genes upon transient PRC1 depletion. They suggest that tumor formation is mediated by failure to repress irreversible genes and provide data supporting this notion by showing that knockdown of zfh1, one of the identified irreversible genes, can partially rescue the PRC1 depletion phenotype.

The authors perform several experiments to understand how the transient depletion of PRC1 results in tumor formation. They hypothesize that weaker binding of PRC1 to the irreversible genes

as well as the binding of transcriptional factors into these genes cause sustained activation of these targets. As a result, aberrant growth is observed, and tumor formation is triggered.

The authors address a very interesting and outstanding question in cancer biology, and their observations suggest that transient epigenetic changes are indeed sufficient for tumor development and initiation. However, we have several comments that we would like the authors to address:

1. Mechanism: The authors write their results show (lines 182-184) "an uncoupling between the impact of transient ph-KD on transcription and chromatin, whereby irreversible transcriptional changes drive tumorigenesis despite the reestablishment of an essentially normal chromatin landscape at Polycomb target genes." The question is how do the authors imagine this? Do the authors think that the *zfh1* gene is expressed when bound by PH (PRC1), PRC2 and enriched for H3K27me3 and H2AK118ub1? The idea that a gene is expressed while bound by Polycomb group proteins and carrying the Polycomb modifications is not supported by the literature. If the authors believe this is the case, they should provide data supporting this point. An alternative hypothesis is that the *zfh1* gene expression is expressed in few cells that remain positive for H3K27ac. If this is the case, then it would be important to show that PH is expressed in the cells that also express *zfh1*.

2. Related to the first point, Fig. 1B shows the WB result for PH level after transient depletion. In line 74 the authors write that the PH protein level is as returning to the normal level following transient depletion. However, the data shows a slightly decreased level of PH in the transient KD samples. This could indicate that PH expression is not re-established in all cells, which may therefore be the cells that contribute to tumor development and maintenance.

3. PH is part of the PRC1 complex that catalyzes H2K118Ub. Therefore, the authors should include H2K118Ub screen shots in Figure 3d. In Figure 4, the authors show ChIP-seq plots for Polycomb and PcG recruiter proteins. To get a comprehensive view of the role of these proteins in regulating reversible and irreversible genes, it would be highly relevant to show the binding profiles of these proteins on irreversible and reversible genes as done in Figure 3d for PH, H3K27me3 and H3K27ac.

4. In Extended Data Figure 4c, the authors shows that ZF1 levels are upregulated more in transient KD samples compared to the constant KD sample. Could the authors please comment on this observation?

5. Fig 2e-h show the partial rescue achieved by *zfh1* knockdown (KD) in the PRC1 depletion system. Did the authors test the rescue capacity of other identified irreversible target genes such as *upd2*, *upd3* and *Ets21C*? Can the combination of KD of several of these targets enhance the rescue phenotype?

6. For the rescue experiment with *zfh1* KD (Fig 2e-h), the authors did not use their thermo-sensitive strategy. Can transient KD of *zfh1* be sufficient to get a similar rescue phenotype?

Additionally, for the rescue experiment, a negative control could be included such as depletion of a selected gene from Cluster 1 since the authors state that upregulation of these genes is an indirect effect of PRC1 depletion (Line 134).

7. In line 206, it is stated that motifs of JAK-STAT pathways downstream effectors are enriched in irreversible genes. Since *zfh1* KD gives only a partial rescue, can KD of *kayak* or *Jra* alone or in combination with *zfh1* give better rescue results?

8. Differential analysis of H3K27me3 and H2AK118ub binding upon constant or transient ph-KD (Fig 3e and Extended Data Fig 5f) indicates that there are only 15 regions with reduced H3K27me3

levels and no regions with reduced H2AK118ub upon transient PRC1 depletion even though there are target genes upregulated (Fig 2a-d). How do the authors explain this observation? Does it mean the binding of PRC1 plays an essential role in the repression of these genes rather than the posttranslational modification?

9. All the RNA-seq, ChIP-Seq and Cut&Run experiments have been performed at a later stage of development (8-11 days AEL). However, to be able to identify primary targets leading to tumour initiation, analysis at the earlier stage can be performed in a time-course manner.

Referee #3 (Remarks to the Author):

Content

Non-genetic mechanisms may also play a role in the initiation of cancer, as significant epigenomic alterations can contribute to tumorigenesis. In this study using *Drosophila*, the authors conclude that temporarily disrupting transcriptional silencing mediated by Polycomb Group proteins leads to an irreversible switch to a cancer cell fate, even without driver mutations. This is linked to the activation of genes associated with tumorigenesis, such as JNK and JAK-STAT signaling pathways and the ZEB1 oncogene. The authors suggest that this altered cell fate can be inherited epigenetically and contributes to cancer development.

Impact

The current view in the field is that tumors arise from an accumulation of permanent mutations. This paper contradicts this notion and shed a completely new light onto tumor ontogenesis. It would have important implications for clinical approaches to diagnostics and treatment, as well as experimental approaches to their molecular analysis. This manuscript would therefore be suitable for publication in *Nature*.

General assessment

I would like to extend my congratulations to the authors for producing such a great manuscript. The data presented in this study is of outstanding quality, the scientific approach is thorough and the extensive controls provide a robust foundation for the authors' claims - while excluding alternative scenarios. I truly enjoyed reviewing it.

Major concern 1

However, I do have one major concern regarding the interpretation of the role of Polycomb, JNK, and JAK/STAT signaling in this process. My question is, to what extent is the origin of this heritable phenotype dependent on Polycomb? Is it possible that this is simply a heritable state that can also be induced by non-epigenetic JNK/JAK activating mutations and tumors? Addressing this issue will significantly strengthen the authors' conclusions and provide a more comprehensive understanding of the mechanisms underlying this phenomenon.

I will try to express this concern in more detail below:

Pinal et al. (2018) demonstrated in their study published in *Nature Communications* that transient JNK activation in conjunction with apoptosis inhibition can lead to the development of a heritable tumor state. (Please cite and reference this paper.) However, Pinal et al only controlled the transient nature of JNK activation. As they used *dronc* mutant discs to suppress apoptosis, the tissue gained apoptosis resistance also during the subsequent recovery phase. Therefore, their model differs from the current study in that the perpetuation of the JNK/JAK activated tumor phenotype may be linked to continuous apoptosis resistance. Nonetheless, their findings raise an important question regarding the potential for a heritable tumor state to be induced by other (non-epigenetic) JNK/JAK activating mutations? This idea is particularly relevant in light of the authors'

observations that Ph-tumors activate JNK and JAK/STAT signaling and demonstrate a high degree of immortality and apoptosis resistance. Therefore, it is possible that JNK/JAK activation in general, rather than transient Polycomb deregulation, is a central feature of a heritable tumor phenotype. The authors' own conclusions may support this theory:

Line 180 ff These results rule out a differential reprogramming of Polycomb-dependent histone marks upon PH depletion or their differential re-establishment upon reinstating PH as general explanations of the difference between reversible and irreversible genes. They also highlight a surprising uncoupling between the impact of transient ph-KD on transcription and chromatin, whereby irreversible transcriptional changes drive tumorigenesis despite the reestablishment of an essentially normal chromatin landscape at Polycomb target genes.

Line 205 ff This approach indicates that irreversible genes' REs are enriched for motifs associated to downstream effectors of the JAK-STAT pathway, including kayak (AP-1/Fos), Jra (Jun) and zfh1, the latter one showing a positive effect on tumor growth (Fig. 2e-g). Together, these results indicate a tight equilibrium, whereby less potent recruitment of PRC1 at the promoter of irreversible genes, coupled with the binding of TFs related to the JAK-STAT pathway, whose cognate motifs are not enriched in the vicinity of reversible genes, might prevent the re-silencing of irreversible genes upon PRC1 recovery.

Hence, I would love to see at least one experiment that demonstrates that a transient induction of a non-PRC1/2 tumor does (or does not) cause a heritable tumor state. Either outcome could help strengthen conclusions about the role of PcG or about the role of JNK/JAK in heritable tumor states. A 'straight-forward' experiment may be:

- transient activation of a UAS RasV12, UAS scrib RNAi model
- transient activation of a UAS Zfh1 UAS hepACT or UAS Zfh1, UAS scrib RNAi model

The later option may actually help to better define the role of Zfh1 in heritable tumor states. It is acting strongly anti-apoptotic, so it may cooperate with activated JNK in tumor induction.

Major concern 2

Please be more clear and precise about the distinction of JNK (kay, jra) and JAK/STAT signaling signatures and the effects they have on tumor growth. It is clear from the literature that both pathways are strongly co-regulated and active in tumor development. However, kay and jra are effectors of JNK signaling and should not be presented as JAK/STAT effectors.

Major concern 3

The authors of the study have introduced their model really well and make every effort to provide a number of relevant controls that demonstrate control over 'transient expression'. They have rigorously tested the initial knock-down as well as the reversal of knock-down at transcript and protein levels. Throughout the manuscript, they introduce highly relevant control experiments (for example to test for mutagenicity of PhKDs) to exclude alternative scenarios. This thorough characterization provides an excellent foundation for all their conclusions.

However, there are two points that require further clarification:

1) At what timepoint after knock-down does Ph come back? How long are these tumors without PH?

To judge the degree of Ph independence, it would be important to provide some approximate measure of when Ph levels are restored. Could the authors conduct an anti-Ph immunostaining time course after a transient L1 induction at day 3, 5, 7?

2) The numbers presented in Ext Data Fig 3b suggest that all PRC1 components, and especially the two other potent PRC1 tumor suppressors Psc and Su(z)2, are mildly decreased in day 9 and day 11 flies. Could you please comment on this? Especially since your constant PhKD only reduces Ph by -0.3/-0.5.

Minor comments

The paper is well-written and mostly easy to follow and understand. However, at times, the writing can be dense, with sentences containing multiple ideas or lacking necessary information. While it is important to keep the manuscript short, sharpening the focus of certain sentences would enhance readability. Additionally, providing more information in the text and legends, when appropriate, would be beneficial.

Some more details below:

Introduction

While the first paragraph of the introduction focuses on epigenetic driver mutations, it is not until the second paragraph that the actual problem addressed in the study is described: epigenetic changes occurring in the absence of driver mutations. To more effectively introduce the focus of the paper, it may be helpful to incorporate some of the compelling examples from the discussion into the introduction. This may orient readers to the central questions addressed more quickly.

Results

- Line 65

One sentence to introduce ph-p, ph-d before Line 65 would help to ease the reader into the genetic set-up and model.

- Line 70: As expected, 100% of EDs collected at the third Larval stage (L3) after constant PH depletion EDs are transformed into tumors (Figure 1c,d; Extended Data Fig. 1d-f). No actual quantifications are shown here. Rephrase the sentence!

- Line 71

Why are your pupal hatching rates so low (30-60%) for wKD controls?

- Line 99

Please provide more information about the graph in Figure 1 H (also Ext Data Fig 4A) in the figure legend. I can guess but I do not completely understand how to read this graph. What is the set size? Why is one bar red? The axis label as 'number of mutated genes' does not make much sense (?).

- Line 117 As expected, both systems are hardly distinguishable at 18°C (no RNAi), as well as upon transient w-KD (Figure 2a; Extended Data Fig. 3a). Briefly mention which assay you use to ease the reader into the data.

- Line 137 Cluster 5 includes many TFs that are bound by both PRC1 and PRC2 and correspond to canonical PcG targets such as *en*, *eve*, *wg*, *Ubx* and *Scr*. *Wg* is not a TF. Remove or rephrase.

- Line 164 As expected, reversible genes (n= 68) show no expression changes after transient ph-KD, whereas irreversible genes (n= 61) remain upregulated (Extended Data Table 2, Figure 3a). Is this not just simply a consequence of your filtering the data rather than an expectation?

- Line 161 ff

I found the text and figure legend associated with Figure 3 to be the most challenging to understand, and would appreciate some clarification and simplification in the writing. Specifically, it would be helpful to expand the information in the figure legends for Figure 3, as it took some time for me to infer the details. From the text, for example, I had difficulty discerning whether the analysis was referring solely to PcG target genes or to all regulated genes, which made it hard to follow specific conclusions. To improve clarity and accessibility, I recommend revising the text and figure legend associated with Figure 3.

- Line 187 ff

The conclusion drawn in Figure 4 is that reversible genes exhibit increased binding by PRC1 components and repressive transcription factors. Yet, H3K27me3 levels appear to not play a strong role. This leads to the question of how much PRC2 is required in this process, given the apparent lack of H3K27me3 involvement. To provide context for this finding, it would be useful to briefly point to the previous literature on the differences between PRC1 and PRC2, and their respective roles in regulating gene expression.

- Line 202 ff

Why does pho show up enriched at reversible genes (Fig 4E) but its motif is enriched at irreversible genes (Fig 4 G)? The same discrepancy is true for NF-YA and CTCF between Fig 4F and Fig 4G. What am I missing? Could you comment on this please?

- Line 205 This approach indicates that irreversible genes' REs are enriched for motifs associated to downstream effectors of the JAK-STAT pathway, including kayak (AP-1/Fos), Jra (Jun) and zfh1, the latter one showing a positive effect on tumor growth (Fig. 2e-g).

Actually, kayak (AP-1/Fos), Jra (Jun) are effectors of JNK signaling. That JNK and JAK/STAT are tightly co-regulated in tumors is clear but this sentence here is a misrepresentation of these classical tumor-promoting pathways.

- Line 218 Importantly, this system induces EICs with similar penetrance, morphological and transcriptional defects compared to the previous one (Extended Data Fig. 6b-e; Extended Data Table 3).

Please provide more information for non-specialists in the figure legends to understand the graphs for Ext Data Fig 6b and enable interpretation. What are the numbers in the plots? What is a standardized residual?

- Line 223 Constant ph-KD tumors were able to expand and invade a fraction of injected host flies at restrictive temperature (Figure 5b).

Rephrase sentence. Tumors do not invade host. Tumors invade distant sites/organs within host.

Discussion

- Line 251 On the other hand, preferential binding of JAK-STAT related TFs in the vicinity of irreversible genes coupled with weaker PRC1 binding at their promoter might foster transcription after transient perturbation of PcG, dampening their re-repression, and resulting in a self-sustaining aberrant cell state that stimulates tumor progression (Extended Data Fig. 8).

Please rephrase to include JNK-related TFs, because this is really what kayak, jra or Ets21C are.

Author Rebuttals to Initial Comments:

Nature manuscript 2023-01-01119 Reviewer response document

Summary of the changes introduced in the revised version of the manuscript

The three reviewers gave a positive assessment of our submitted manuscript version but also raised important points and suggested experiments in order to improve it. We summarize below the main areas in which we have improved the manuscript.

1- Insight into the mechanisms generating epigenetic tumors.

We provide critical evidence showing that transient depletion of PRC1 induces the JAK-STAT pathway, including its transcription factor STAT92E, as well as the transcriptional repressor, ZFH1. Together, these two factors induce specific changes in chromatin accessibility that support irreversible transcriptional changes, which resist Polycomb reestablishment and foster cell proliferation, compromised cell polarity and cell differentiation. Therefore, a multi-step pathway initiated by the dysregulation of Polycomb function leads to a self-sustaining oncogenesis process.

2- Excluding the possibility that genomic alterations are responsible for tumorigenesis.

We sequenced six more tumors and did extensive sequence analysis to exclude mutations as a cause for tumorigenesis. Furthermore, we showed the absence of high rate of DNA damage in epigenetic tumors and we analysed tissues as early as 24 hours after the onset of PH depletion, which we found induce over-proliferation of cells throughout the whole tissue and not just of sporadic cell clones. These results are inconsistent with oncogenic mutations of a cell clone as the cause of tumorigenesis and instead suggest that derailment of gene expression in all cells trigger Epigenetically Initiated Cancers.

3- Strengthening the tumour rescue assays

We strengthened the rescue assays by proving that the depletion of ZFH1 but also of the JAK/STAT pathway transcription factor STAT92E, can rescue tumors. This is seen not only in constant depletion conditions, but also upon transient depletion of these components. Furthermore, we provided new data showing that chromatin accessibility and gene expression is rewired upon ZFH1 or STAT92E depletion, substantially healing the transcriptional defects induced by PH depletion.

4- Investigate to what extent is the origin of this heritable phenotype dependent on PRC1.

Our new results indeed converge towards a multistep model in which the transient disruption of PcG-mediated silencing is sufficient to irreversibly activate the JAK-STAT pathway and its downstream effector STAT92E, whose binding prevents the re-repression of irreversible genes. Therefore, although Polycomb dysregulation is the initial trigger, at least two downstream transcription factors drive oncogenesis. Intriguingly, we also found that transient depletion of another tumor suppressor, the scribble gene, can induce cancers, although less efficiently than PH depletion. This opens the way to future investigation and suggests that Epigenetically Initiated Cancers might result from different triggering events.

Main Figure changes and new datasets provided in order to respond to the reviewer comments:

Figure 1: The WB shown in panel b was changed to a new one, which was used to now quantify the signal (Extended Data Fig. 1d). Panels f,g,h related to SNV/InDels were replaced with a EdU staining (f), an improved set of analyses regarding gDNA alterations (g-i) and the quantification of γ H2Av foci/cell before and after Ionizing Radiation (j).

Figure 2: The new panel c shows a refined analysis of the over-representation of direct PcG targets in all RNA-Seq clusters. Old panels c-d were accordingly moved to d-e and old panels e-h were moved to Figure 5 a-b.

Figure 3: Old panel a was not necessary and was replaced with the old panel b. Old panel c was replaced with an improved analysis and moved to Extended Data Fig. 6b. New panels b and c show the number of irreversible and reversible PcG target genes that overlap H2K27me3 domains or H3K27Ac peaks, respectively. Panel d was updated to include the H2AK118Ub mark deposited by PRC1. Old panel g was moved to f and new panels g-h show a refined analysis of H3K27me3 and H3K27Ac fold changes after constant or transient *ph*-KD.

Figure 4: The analyses shown in the previous version of Figure 4 have been replaced by our new dataset and analyses (see the point-by-point response below). The revised version of Figure 4 is therefore entirely new and introduces novel key findings, derived from the analysis of newly generated ATAC-Seq data.

Figure 5: The old Figure 5 was moved to Figure 6 and left unchanged. Revised panels a-b correspond to the old panels e-h from Figure 2 – showing the impact of concomitant *zfh1*-KD on tumor growth – to which we added similar results using concomitant *Stat92E*-KD. New panels c-h introduce novel key findings derived from the analysis of newly generated RNA-Seq and ATAC-Seq data.

Figure 6: Corresponds to the old figure 5 without modifications.

New datasets: To address reviewers' comments, we newly generated the following datasets:

- Genomic DNA: 8 samples consisting of 6 independent EIC samples and 2 control tissues (no *ph*-KD).
- ATAC-Seq: 6 conditions (2 biological replicates/condition), including no *ph*-KD (control), transient *ph*-KD, constant *ph*-KD, constant *gfp+w*-KD, constant *gfp+ph*-KD and constant *zfh1+ph*-KD.
- RNA-Seq: 4 conditions (3 biological replicates/condition), including constant *gfp+w*-KD, constant *gfp+ph*-KD, constant *Stat92E+ph*-KD and constant *zfh1+ph*-KD.

New Extended Data Figure 3 shows new data suggesting that DNA repair is normal in EICs and no increase in aneuploidies can be detected.

New Extended Data Figure 7 shows that the transient depletions of STAT92E or of ZFH1, in addition to the transient depletion of PH, are sufficient to partially restore normal eye disc development.

Several changes have been made to other **Extended Data Figures** in order to improve the analysis or present individual panels that were either present in the original submission or are new.

Old **Extended Data Table 3** was moved to **Extended Data Table 5**.

New **Extended Data Table 3** now reports the ATAC-seq peaks obtained by comparing control samples to constant and transient *ph*-KD, see methods, text and **new Figure 4**.

New **Extended Data Table 4** reports the analysis of the transcriptomes after *zfh1+w*-KD, *Stat92E+w*-KD, *gfp+ph*-KD, *zfh1+ph*-KD and *Stat92E+ph*-KD, all were compared to temperature-matched *gfp+w*-KD. See methods, text and **new Figure 5**.

Point-by-point response to the comments of each reviewer

This revision required a relatively complex new experimental work and data analyses, which resulted in several changes in figures and text. To facilitate the navigation through the document, we wrote our responses in blue. For each point, in addition to the response text we pasted the corresponding figure panels and figure legends that are used in the revised manuscript, such that reviewers do not need to go the manuscript to track the changes. Furthermore, key sentences that we introduced in the manuscript to address reviewer's issues are transcribed in green in our response text. Finally, we introduced some Reviewer Figures, which present additional data that are not introduced in the manuscript in order to avoid adding unnecessary complexity for the readers, but which can help reviewers in their assessment. Finally, we insert below a "Table of contents", reporting each reviewer questions as

subheadings that are hyperlinked to the corresponding pages. We hope that this will help the reviewers reading through our responses.

Table of Contents

SUMMARY OF THE CHANGES INTRODUCED IN THE REVISED VERSION OF THE MANUSCRIPT	1
POINT-BY-POINT RESPONSE TO THE COMMENTS OF EACH REVIEWER	2
REFEREE #1 (REMARKS TO THE AUTHOR):.....	4
MAJOR COMMENTS:	4
REVIEWER#1 MAJOR COMMENT 1.	4
REVIEWER#1 MAJOR COMMENT 2.	5
REVIEWER#1 MAJOR COMMENT 3.	7
REVIEWER#1 MAJOR COMMENT 4.	8
REVIEWER#1 MAJOR COMMENT 5.	11
REVIEWER#1 MAJOR COMMENT 6.	12
REVIEWER#1 MAJOR COMMENT 7.	15
MINOR COMMENTS:	16
REFEREE #2 (REMARKS TO THE AUTHOR):.....	18
REVIEWER#2 MAJOR COMMENT 1.	18
REVIEWER#2 MAJOR COMMENT 2.	20
REVIEWER#2 MAJOR COMMENT 3.	21
REVIEWER#2 MAJOR COMMENT 4.	25
REVIEWER#2 MAJOR COMMENT 5.	25
REVIEWER#2 MAJOR COMMENT 6.	27
REVIEWER#2 MAJOR COMMENT 7.	30
REVIEWER#2 MAJOR COMMENT 8.	30
REVIEWER#2 MAJOR COMMENT 9.	31
REFEREE #3 (REMARKS TO THE AUTHOR):.....	33
REVIEWER#3 MAJOR CONCERN 1	33
REVIEWER #3 MAJOR CONCERN 2	41
REVIEWER #3 MAJOR CONCERN 3	42
CONCERN 3, SUB-POINT 1)	42
CONCERN 3, SUB-POINT 2)	42
MINOR COMMENTS.....	43
REVIEWER RESPONSE REFERENCES.....	47

Referee #1 (Remarks to the Author):

This manuscript attempts to address if non-genetic reprogramming is sufficient to induce tumourigenesis which would be an important step forward for the role of epigenetics in tumour initiation. The authors generate a transient RNAi knockdown of PRC1 components and show this is sufficient to induce overgrowth and loss of apico-basal polarity and differentiation markers. They carry out DNA sequencing to explore whether this transient depletion is promoting overgrowth by a genetic mechanism. Transcriptomic analysis of the tumours indicates a cluster of genes that are altered beyond the transient depletion which are associated with cytokine activity and contain direct PcG target genes. Epigenetic analysis explores the restoration of chromatin at altered genes and they propose that transcriptional repressor binding distinguishes irreversibly altered genes. Serial transplantation of the tumours indicates they become more aggressive over time. This is a large body of impressive work but raises a number of important questions and seems somewhat incomplete.

We thank the reviewer for the positive comments and hope that the following point-by-point response correctly addresses pending questions.

Major comments:

Reviewer#1 major comment 1.

PH depletion at L3 late does not lead to tumours, perhaps suggesting this requires developmental, non-differentiated tissues and raises the question of relevance to adult tissues? Is transient depletion simply enforcing a developmental state or preventing differentiation during drosophila development rather than inducing an ‘epigenetically initiated cancer’?

Losing the ability to differentiate is one of the hallmarks of cancer. In this regard, our system is relevant to understand how tumors can arise during development, such as pediatric posterior fossa ependymoma, showing impaired differentiation without any clear driver mutations. After a 24h *ph*-KD at the late L3 stage, the tissue looks indeed more normal compared to earlier stages, as pointed out by the reviewer (Extended Data Fig. 1g). However, when injected in the abdomen of adult flies, these tissues were immortal and grew into neoplastic tumors until they eventually kill the host (new Extended Data Fig. 9d-e):

Extended Data Fig. 9:

d-e- Tumor growth measured as a percentage of flies showing tumoral progression 20 days after transplantation (d) and survival of host flies 20 days after allograft of late L3 *ph*-KD (e) during 5 consecutive rounds of transplantation.

Consistently, we performed RNA-Seq on these tissues where 17/30 irreversible and 19/42 reversible genes that are direct PcG targets are already significantly upregulated (without any cutoff on log2FoldChange, these numbers increase to 21/30 and 29/42, respectively):

Reviewer Figure #1

Over-representation of significantly upregulated genes (adjust $pval < 0.05$) after 24h *ph-KD* at the late L3 stage with an additional cutoff on \log_2 FoldChange (>1 , on the left) or not (right).

Even if we could not track the further development of these tissues in pupae because of the technical difficulties for their dissection, all pupae died with severe disruption in their head structure, suggesting that tumorigenesis might be responsible for lethality. These data suggest that tumors dependent on PRC1 depletion might develop during an extended time window. Our system is not designed for the specific assessment of adult tumours. However, adult cancers can be generated in flies¹, and it would be interesting in future studies to explore whether transient perturbations might be sufficient to drive neoplastic transformation in adult tissues.

Reviewer#1 major comment 2.

In the DNA-sequencing, the authors show a significant amount of mutations (in the thousands) and argue these are not responsible for tumour formation. Where do these mutations come from and what are they relative to? Can they filter out the mutations unique to PH knockdown due to batch effect to really understand which are arising from the knockdown? A simple graph showing absolute number of new mutations arising would provide clarity rather than feature proportions.

We thank the reviewer for this question, which inspired us a new series of experiments and representations. We performed gDNA sequencing for two extra batches of tumors (for a total of four) that were dissected after one, four or six days of recovery at 18°C following a 24h *ph-KD* at the early L3 stage (eL3 *ph-KD* 1d, 2d, 6d recov). Following the reviewer's idea, we added an upset plot in Extended Data 3a showing the number of SNV/InDels found in each sample as well as the overlap between the different samples:

Extended Data Fig. 3: *ph-KD* does not induce the accumulation of mutations or aneuploidy.

a- SNV/InDels overlaps between all sequenced gDNA samples. Each vertical bar corresponds to an intersection (corresponding samples are shown below) and horizontal bars (bottom left) show the total number of SNV/InDels found in each sample. Only intersections containing ≥ 40 SNV/InDels are shown and SNV/InDels that are specific to one sample are indicated in orange (68.1% of all detected SNV/InDels).

This panel shows that 68% of SNV/InDels are specific to one sample (in orange) and we detected less SNV/InDels in five of the new tumors (eL3 *ph-KD*) than in the “No *ph-KD*.6” control sample. The “Transient *ph-KD* d9.1” tumor sample also contains less mutations than the “No *ph-KD*.2” control. Moreover, a substantial fraction of the SNV/InDels that are shared between multiple samples are also found in the “no *ph-KD*.2” control sample and are generally

shared between the first six rows, which correspond to the samples of our first series of experiments. Thus, many of these correspond to variants with low allelic frequencies. Consistently, the **new Figure 1 g** shows that 92.8% of the detected SNV/InDels have an allelic frequency smaller than 0.2, making them unlikely to drive the whole-tissue tumours that arise upon the depletion of PH (see also below and **new Figure 1f**, in which we describe the fact that cells start overproliferating across the whole tissue within 24 hours from the depletion).

Figure 1: Transient PRC1 depletion is sufficient to initiate tumors.
g- Distribution of somatic SNV/InDel allelic frequencies detected across all samples.

Therefore, following the reviewer's advice, we focused on the SNV/InDels that are only found in tumor samples and not in any of the control samples, with an allelic frequency higher than 0.2. This approach drastically decreased the number of confident SNV/InDels down to a total of ~2,000 across the 12 tumor samples, of which 89.4% remain specific to one sample:

Figure 1: Transient PRC1 depletion is sufficient to initiate tumors.
h- Fraction of tumor samples in which each SNV/InDel, gene with deleterious SNV/InDels, Structural Variants (SVs) and Copy Number Variation (CNVs) were found.

Most importantly, none of the detected SNV/InDels are shared between the 12 tumor samples. Although the precise genomic coordinates of SNV/InDels found in different samples might differ, it was important to eliminate the possibility that different deleterious SNV/InDels end up affecting the same genes in different samples, but 85.1% of the genes containing deleterious SNV/InDel(s) are only found in one sample, and none is shared between more than 3 tumor samples. Consistently, SNV/InDels detected in control samples or in at least two tumor samples had similar features distribution, with no bias towards exons (**new Figure 1i**):

Figure 1: Transient PRC1 depletion is sufficient to initiate tumors.
i- Feature distribution of SNV/InDels that were found in any of the control samples (no ph-KD, left bar) or shared between at least two tumor samples (right bar).

Altogether, these results indicate that *ph*-KD is not associated with the massive accumulation of SNV/InDels, and their inconsistency across samples argue against specific genetic events that might repeatedly drive *ph*-dependent tumorigenesis. One possible explanation why we do not detect an increase in mutations might be that, at early stages at least, only few jackpot cells mutate and initiate tumorigenesis. However, a new EdU incorporation experiment that we added in the **new Figure 1f** shows that, as early as 24h after a transient depletion of PH, there is a massive EdU incorporation throughout the tissue, excluding the possibility that tumors arise from a single or few cells of origin. To orthogonally assess whether DNA damages increase after transient *ph*-KD, we also performed γ H2Av staining before and after ionizing radiation (IR). Before IR, the number of γ H2Av foci was only slightly increased after transient *ph*-KD (**new Figure 1j**), which might simply reflect the higher fraction of cycling cells that are found throughout the tissue 24h after transient *ph*-KD (**Figure 1j**):

Figure 1: Transient PRC1 depletion is sufficient to initiate tumors.

f- EdU stainings (in green) imaged 0h (left) and 24h (right) after 24h of w-KD (control, top) or *ph*-KD (bottom). Scale bars: 100 μ m. j- Number of γ H2Av foci per cell before (0') and after (30', 480') irradiation in control (no *ph*-KD, on the left) or transient *ph*-KD EDs (right). t.test: *****p*val<1e-5.

Between 30min and 480min after irradiation, transient tumor cells showed similar recovery compared to the control (no *ph*-KD), suggesting that they efficiently repair their DNA to avoid the massive accumulation of mutations. All these results are further discussed in the section entitled “**A transient epigenetic perturbation is sufficient to initiate tumors**” of the revised manuscript (line 64).

Reviewer#1 major comment 3.

Unclear whether copy number alterations are being induced by transient PH depletion. This could contribute to persistence of tumour formation. This seems an oversight given the importance of copy number alterations in cancer progression.

We thank the reviewer for this suggestion. We analyzed copy number variations (CNVs) and Structural Variants (SVs) that are specifically found in any of the 12 tumor samples (**new Figure 1h**):

Figure 1: Transient PRC1 depletion is sufficient to initiate tumors.

h- Fraction of tumor samples in which each SNV/InDel, gene with deleterious SNV/InDels, Structural Variants (SVs) and Copy Number Variation (CNVs) were found.

93.4% of SVs and 78.7% of CNVs are specific to one unique sample and none was shared between all tumor samples.

We noted that 12 CNVs were shared between more than 6 samples. Of these, 10 showed $\geq 80\%$ overlap with a CNV detected in control samples, suggesting that they are false positives whose start and end genomic coordinates called by the CNVnator software only differ by a small offset. The two remaining CNVs are intergenic deletions: one 600bp deletion on chromosome X and one of 800bp on chr2L. Their closest genes are *SkpD*, *lncRNA:CR44885* and *His-Psi:CR31614*, *Lamp1*, respectively. The *CR44885* lncRNA has no known function, *CR31614* is a pseudogene and neither *SkpD* (predicted to be part of SCF ubiquitin ligase complex) nor *Lamp1* (involved in lipophagy) have known role in tumorigenesis. None of these genes were found to be differentially expressed after Transient *ph*-KD.

Hence, we did not find any evidence that CNV or SV play a role in driving *ph*-dependent tumorigenesis. Consistently, we looked at the karyotype of cells after transient *ph*-KD and found no sign of aneuploidy (**new Extended Data Fig. 3d**):

Extended Data Fig. 3: *ph*-KD does not induce the accumulation of mutations or aneuploidy.

d- Representative karyotypes (left) and quantification of chromosome abnormalities in EDs after no *ph*-KD (control, top) and transient *ph*-KD (bottom). Schematic representation shows the position of the satellites stained. Scale bars: 1mm (c, d).

Reviewer#1 major comment 4.

Does knockdown of *zfh1* during or even after transient depletion of PH rescue tumour formation, this could provide insight into mechanism.

We thank the reviewer for proposing this interesting experiment. We developed a new driver line allowing for transient double RNAi and found that concomitant *zfh1*-KD decreases the overgrowth of the tissue and partially restores differentiation (**new Extended Data Figure 7**):

Extended Data Fig. 7: Transient *Stat92E*-KD and *zfh1*-KD are sufficient to substantially rescue transient *ph*-KD tumor defects a-c DAPI (grey, a), F-actin (red, b) and ELAV (magenta, c) staining of EDs after transient *gfp*-KD (control), *Stat92E*-KD and *zfh1*-KD in the presence of a concomitant, transient depletion of white (*w*-KD, control, three first columns) or *ph* (*ph*-KD, last three columns). DAPI staining was used to assess the overgrowth of the different tissues, F-actin for apico-basal polarity defects and the neuronal marker ELAV for differentiation defects. d- ED sizes quantified as overall DAPI staining area for the different conditions, showing that transient *Stat92E*-KD and *zfh1*-KD decreased ED overgrowth following transient *ph*-KD. Two-sided Wilcoxon test: ***p*val<1e-2, ****p*val<1e-3, *****p*val<1e-5. Scale bars: 100 μm (a), 10 μm (b, c).

Regarding the mechanism by which *zfh1* upregulation fosters *ph*-dependent tumorigenesis, we performed new ATAC-Seq experiments after no *ph*-KD and transient *ph*-KD. These data showed that the *zfh1* DNA binding motif is associated with decreased accessibility after transient *ph*-KD (new Figure 4f,h):

Figure 4: Chromatin accessibility changes underlie reversible and irreversible transcriptional changes. f- Linear model t-values of DNA binding motifs associated with increased (positive t-values) or decreased (negative t-values) accessibility after transient (x axis) or constant *ph*-KD (y axis). Only motifs with a significant *p*.value<1e-5 in at least one of the two linear models are shown. g- Fold changes at ATAC-Seq peaks (y axis) upon constant (left) or transient (right) *ph*-KD, depending on the number of caudal (*cad*) motifs they contain (x axis). Two-sided Wilcoxon test: N.S- not significant, *****p*val<1e-5. h- Fold changes at ATAC-Seq peaks (y axis) upon transient *ph*-KD, depending on the number of *Stat92E* (left, in orange) or *zfh1* (right, in blue) motifs they contain (x axis). Two-sided Wilcoxon test: ***p*val<1e-2, *****p*val<1e-5.

This result suggests that *zfh1* acts by reducing the accessibility of a subset of regulatory elements (such as enhancers and/or promoters), consistent with its known role of transcriptional repressor^{2,3}. To further understand ZFH1 mode of action, we built on the fact that concomitant *zfh1*-KD also rescues *ph*-KD to perform RNA-Seq and ATAC-Seq after constant *zfh1+ph*-KD (new Figure 5):

Figure 5: Tumor development requires STAT92E and ZFH1.

a- DAPI (top, in grey) and ELAV (bottom, in magenta) staining of EDs after constant *gfp+w*, *Stat92E+w*, *zfh1+w*, *gfp+ph*, *Stat92E+ph* and *zfh1+ph-KD* (see top labels). ELAV is used as a marker of terminally differentiated neurons. Scale bars: 100 μm (DAPI), 10 μm (ELAV). *b*- ED area distributions, as measured using DAPI stained tissues (number of measured EDs is reported in brackets). *c*- Number of differentially expressed genes after *gfp+ph* (tumors), *Stat92E+ph* and *zfh1+ph-KD*. Transitions between upregulated (orange), unaffected (grey) and downregulated (blue) states are indicated by thin lines of the same respective colors. *d*- Number of ATAC-Seq peaks showing significant accessibility changes after *gfp+ph* or *zfh1+ph-KD*. Transitions between increased (orange), unaffected (grey) and decreased (blue) states are indicated by thin lines of the same respective colors. *e*- Fold changes at ATAC-Seq peaks between *zfh1+ph* and *gfp+ph-KD*, depending on the number of ZFH1 motifs they contain (*x* axis). Two-sided Wilcoxon test: *****p*val $<1e-5$. *f*- RNA-Seq fold changes upon *gfp+ph-KD* (*x* axis) of genes associated to ATAC-Seq peaks that are decreased (in blue), unaffected (in grey) or increased (in orange) after *zfh1+ph-KD* compared to *gfp+ph-KD* (*y* axis). *g*- Top enriched GO terms for genes associated to ATAC-Seq peaks containing at least one ZFH1 motifs and showing significantly increased accessibility after *zfh1+ph-KD* compared to *gfp+ph-KD*.

zfh1-KD rescues the normal expression of about 50% of the genes that are differentially expressed after constant *ph*-KD (including a majority of down-regulated genes, new Figure 5c). *zfh1*-KD also rescued a majority of the ATAC-Seq peaks showing decreased accessibility upon *ph*-KD (new Figure 5d). This “re-opening” was correlated with the presence of ZFH1 DNA binding motifs (Figure 5e), consistent with its association to reduced accessibility after transient *ph*-KD (new Figure 4h). In order to link accessibility to transcriptional changes, we assigned ATAC-Seq peaks that were increased upon *zfh1+ph*-KD compared to *ph*-KD (and likely correspond to regulatory elements that are abnormally closed by *zfh1* in tumors) to the closest TSS (+/- 25kb). Interestingly, the assigned genes were significantly downregulated after constant *ph*-KD and were implicated in ED development differentiation and neuron differentiation (Figure 5f,g).

Altogether, these results suggest a model where the aberrant expression of *zfh1* – which translates into higher protein levels after transient *ph*-KD (Extended Data Fig. 5c,d) – fosters the aberrant repression of genes responsible for the normal development and differentiation of ED, preventing differentiation and in turn stimulating neoplastic growth. These new results

are discussed in the paragraph entitled “STAT92E and ZFH1 are required for EIC development” (line 257).

Reviewer#1 major comment 5.

Unclear why is the gene set reduced to PH bound genes and how are non-PH target genes being altered, this is not a minority of the gene set for figure 3A/B.

Tumor transcriptomes typically correspond to a mix of direct and indirect effects, the latter ones being a delayed consequence of the former. Two possible ways to disentangle direct effects from indirect ones are to 1/ focus on genes that are bound by the protein of interest (and are therefore more likely to be directly affected) and 2/ measure the transcriptome as quickly as possible after the perturbation.

To tackle the primary transcriptional defects triggered by *ph*-KD, we now performed RNA-Seq right after a 24h *ph*-KD at the late L3 stage. We found that 44% of these early responsive genes that were already upregulated were covered with the PcG-mediated H3K27me3 repressive mark (18 times more than expected):

Reviewer Figure #2

Enrichment of PcG target genes (for which at least 50% of the gene body is covered by PcG-mediated H3K27me3 repressive mark) in genes that are significantly upregulated, downregulated or remain unaffected after 24h of *ph*-KD at the late L3 stage. One sided fisher test, **** p val<1e-5.

Thus, primary transcriptional defects indeed seem to consist in the derepression of silenced PcG target genes. In Figure 3, our goal was to focus on the most direct, primary effects that allowed the derailment of the transcriptome, and this is why we focused only looked at irreversible and reversible genes that are covered with the H3K27me3 repressive mark (in control tissue), to understand whether their distinct behaviors could be explained by differences in surrounding PcG landscape.

To better explain our logic, we revised the manuscript by stating that “**Interestingly, the upregulated clusters show stronger and significant over-representation of PcG target genes covered with H3K27me3 – the canonical mark of PcG-mediated transcriptional repression – in control EDs (Figure 2c). This suggests that their upregulation is a direct consequence of compromised PcG repression, although they retain distinct patterns.**” (line 141). The **new Figure 2c** now shows the preferential enrichment of Irreversible and Reversible clusters for PcG target genes (of note, we also renamed the RNA-Seq clusters after their characteristic patterns, Fig. 2b):

Figure 2: EICs show irreversible transcriptional changes.

a- Number of differentially expressed genes after no ph-KD (control), constant and transient ph-KD. Transitions between upregulated (orange), unaffected (grey) and downregulated (blue) states are indicated by thin lines of the same respective colors. b- Clustering of differentially expressed genes after constant or transient ph-KD. c- Over-representation of direct PcG target genes ($\geq 50\%$ of the gene body overlaps a H3K27me3 repressive domain in control condition). One-sided Fisher's exact test: *** p val $<1e-3$, **** p val $<1e-5$.

Reviewer#1 major comment 6.

The authors claim that 'essentially normal chromatin landscape' returns following transient depletion of PH, yet zfh1 H3K27Ac does not return to baseline. This appears to be significant as zfh1 the authors claim is required for the phenotype. This appears an overstatement and would be important to look closely at the 'few exceptions' in detail and why the normal chromatin landscape is not restored. This could provide critical insight into mechanisms downstream of ph that leads to tumours. In the absence of further proof it is unclear that differential transcriptional repressor binding of a small subset is the underlying mechanism.

We thank the reviewer for highlighting this critical point, which led to important clarifications and new key experiments. First, we want to clarify what we mean by "essentially normal" chromatin landscape. When using peak calling to define PH peaks, H3K27me3 and H2AK118Ub domains (the two canonical marks of PcG repression deposited by PRC2 and PRC1, respectively) we observed that the majority were lost upon constant depletion but recovered after transient ph-KD (new Extended Data Figure 6b):

Extended Data Fig. 6: PcG epigenetic landscape is re-established after transient ph-KD.

b- PH peaks, H3K27me3 (PRC2-mediated repressive mark) and H2AK118Ub (PRC1-mediated repressive mark) domains overlaps after no ph-KD (control), constant or transient ph-KD. Each vertical bar corresponds to an intersection (corresponding conditions are shown below) and horizontal bars (bottom left) show the total number of peaks/domains detected in each sample. To avoid weak and noisy peaks/domains, we focused on domains containing at least one PH peak and on PH peaks overlapping H3K27me3 domains in control sample.

Consistently, most PcG-target irreversible and reversible genes lose the H3K27me3 repressive mark and gain its activating counterpart H3K27Ac upon constant depletion, but return to baseline after transient ph-KD:

Figure 3: PcG repressive landscape is restored after transient *ph*-KD.

b- Number of irreversible (pink) and reversible (green) genes overlapping a H3K27me3 domain ($\geq 50\%$ of the gene body) after no *ph*-KD (control), constant or transient *ph*-KD. *c*- Number of irreversible (pink) and reversible (green) genes overlapping at least one H3K27Ac peak (in the gene body or up to 2.5kb upstream of the TSS) after no *ph*-KD (control), constant or transient *ph*-KD.

Hence, a binary classification based on the presence/absence of PcG features (PH, H3K27me3, H2AK118Ub) after transient *ph*-KD is not sufficient for the systematic identification of irreversible versus reversible genes. Nevertheless, the reviewer correctly points out that few irreversible loci show irreversible H3K27Ac peaks after transient *ph*-KD, which we indicated line 195: “Nevertheless, we noted some exceptions, such as the *zhf1* gene which retains low but significantly higher levels of H3K27Ac compared to control tissues upon transient depletion of PH (Figure 3d), suggesting that a fraction of irreversible loci might retain small quantitative differences”. However, out of 30 irreversible genes, only five show such irreversible H3K27Ac peaks, out of which three recovered detectable H3K27me3 levels (*zhf1*, *Ets21C* and *CG12688*) after transient *ph*-KD:

Reviewer Figure #3

Screenshot of H3K27me3 (top) and H3K27Ac CUT&RUNs tracks at loci acquiring irreversible H3K27Ac peaks after transient *ph*-KD.

The 25 remaining irreversible genes had no detectable H3K27Ac peaks after transient *ph*-KD despite being irreversibly upregulated, such as the *upd* locus showed in Figure 3d. Consistently, the differential analysis of H3K27me3 and H3K27Ac in Figure 3e,f shows that these two marks are massively affected after constant *ph*-KD compared to transient *ph*-KD, and now highlighted the peaks/domains overlapping the *upd* and *zhf1* loci:

Figure 3: PcG repressive landscape is restored after transient *ph*-KD.

e-f- For H3K27me3 domains (e) and H3K27Ac peaks (f) detected across all conditions, fold changes after constant (left) or transient (right) *ph*-KD are shown. Significant changes are highlighted using a color code (see color legend).

zfh1 indeed appears to be an outlier for its significant increase in H3K27Ac after transient *ph*-KD, contrasting with the *upd1-3* irreversible locus. We then classified H3K27me3 domains based on whether they contain irreversible or reversible genes, which are generally found in distinct domains (see new Extended Data Fig. 6d). We only found small differences between reversible and irreversible, the latter ones showing slightly lower H3K27me3 fold changes and a reciprocal difference in H3K27Ac fold change (new Figure 3g,h):

Figure 3: PcG repressive landscape is restored after transient *ph*-KD.

g- H3K27me3 fold changes at H3K27me3 domains that are found in control sample (no *ph*-KD) and overlap irreversible (pink) or reversible (green) genes. All H3K27me3 domains are shown as a reference (grey). Two-sided Wilcoxon test: N.S-

not significant, * $pval < 5e-2$, ** $pval < 1e-2$, *** $pval < 1e-3$, **** $pval < 1e-5$. h- H3K27Ac fold changes at H3K27Ac peaks overlapping H3K27me3 domains that are found in control sample (no *ph*-KD) and overlap irreversible (pink) or reversible (green) genes. All H3K27Ac peaks overlapping control H3K27me3 domains are shown as a reference (grey). Two-sided Wilcoxon test: N.S- not significant, **** $pval < 1e-5$.

Thus, we think it is fair to state line 207 that “differences in Polycomb-dependent histone marks upon PH depletion or their differential re-establishment upon reinstating PH are unlikely to explain the difference between reversible and irreversible genes”. Consistent with this conclusion, we did not find any significant difference in terms of H3K27me3, H2AK118Ub or H3K27Ac levels between these two groups, neither in no *ph*-KD (control condition) nor after transient *ph*-KD (new Extended Data Fig. 6a):

Extended Data Fig. 6: PcG epigenetic landscape is re-established after transient *ph*-KD.

a- PH ChIP-Seq (top), H3K27me3 (2nd row), H2AK11Ub (3rd row) and H3K27Ac (bottom) CUT&RUN average tracks, anchored at the TSS of PcG-bound irreversible (in pink), reversible (in green) and unaffected genes (in grey) after no *ph*-KD (control, on the left), constant (middle) or transient *ph*-KD (right). For each condition, the average signal is shown (solid line) \pm standard error (shaded area) and the distance to the TSS is shown on the x axis. The signal was quantified at the regions highlighted with dotted lines (see corresponding box plots on the right). Two-sided Wilcoxon test: * $pval < 0.05$; N.S- Not significant.

To summarize, we think that the PcG epigenetic landscape and its potential alterations after transient *ph*-KD are not sufficient to explain why certain genes remain irreversibly upregulated and thus we posit that the action of specific TFs that perturb their re-repression might be involved. We discuss this line 213 in the new paragraph entitled “Specific chromatin opening sites mark reversible and irreversible loci” (and corresponding new Figure 4), where we used ATAC-Seq to show that irreversible transcriptional changes are mainly supported by STAT92E, the key effector TF of the JAK-STAT pathway.

Reviewer#1 major comment 7.

In addition, this reviewer would like to see a clearer visualisation/analysis of the relation between the histone marks and gene expression. At present, it is unclear how and if the repressive and activating marks are different between the reversible and irreversible genes, especially in light of the observations on *zfh1* as described in the previous point.

Please refer to the last panel of the previous point. We did not detect any significant difference between the two groups.

Reviewer#1 major comment 8. Tumours becoming more aggressive over time could indicate ongoing genomic instability or chromosome instability induced by transient PH depletion.

Please see answers to points 2 and 3: we did not find any evidence for ongoing genomic instability, at least before transplantation. To assess whether genomic instability might increase after several rounds of allografts, we tried to FACS the GFP+ cells from the allograft to sequence their genomic DNA, but this approach appeared to be technically challenging and we did not manage to have enough samples. Hence, we cannot rule out that tumors that have been transplanted multiple times acquired genomic instability, and future studies will have to be carried out in order to address this interesting possibility. To clarify that genetic mutations might be involved in secondary tumour progression over time we added a note at line 315: **“The fact that tumors become progressively more aggressive over time might suggest that they acquire secondary modifications, either genetic or epigenetic, that increase their aggressiveness.”**

Reviewer#1 major comment 9. Evidence supporting a role of this epigenetic initiation in humans is weak and has been carried out in cancers with widespread genomic driver alterations.

We certainly agree and the jury is still out on whether epigenetic perturbations can initiate a human cancer. We made sure in the revised introduction to single out posterior fossa ependymoma, which is the best characterized example of a tumour for which no driver mutation has been found yet, but even there we do not know whether the cause for the tumour is epigenetic, we simply have no genetic driver identified. We believe however that, albeit challenging, this open question is a very important one for future cancer biology research.

Minor comments:

- 1) Confusing break in axis in extended data figure 1G. In general, breaking of axis do not aid clarity of interpretation.

This panel was moved to Extended Data Fig. 1f, and we updated it by adding dotted lines to delineate where the y axis is cut. We need to have an axis break because, without it, it would be difficult to see the hatching rate levels for the constant *ph*-KD condition:

- 2) Western showing return to normal levels is unclear and quantification of western blot would confirm levels of PH return to baseline.

Thanks to the reviewer’s comment, we now repeated this WB in triplicate and show the corresponding quantification in the **new extended Data Fig. 1d**:

Extended Data Fig. 1: Transient PH depletion generates neoplastic tumors that persist after PH recovery.

d- Quantification of the Western blot shown in Fig. 1b. Error bars represent the standard error of the mean for three independent experiments. Dunnet’s test: ns= not significant, **pval<0.01.

- 3) Statistical analysis showing over representation of PcG target genes in cluster 2 would

avoid potential observed random over representation.

We thank the reviewer for encouraging us to clarify this important point. Following up on our response to point #5, we now added a new panel to assess the overrepresentation of direct PcG targets in each cluster (**new Figure 2c**):

Figure 2: EICs show irreversible transcriptional changes.

c- Over-representation of direct PcG target genes ($\geq 50\%$ of the gene body overlaps a H3K27me3 repressive domain in control condition). One-sided Fisher's exact test: *** $pval < 1e-3$, **** $pval < 1e-5$.

When relevant, we reduced the set of genes to the ones that are PcG-bound – defined as having more than half of the gene body covered by the H3K27me3 repressive mark – which are the primary candidates to trigger a derailment of the transcriptome. For example, we did this distinction when looking at the GO terms of gene clusters or when focusing on genes involved in the JAK-STAT pathway (**new Figure 2d-e**):

Figure 2: EICs show irreversible transcriptional changes.

d- Representative GO terms enriched for each gene cluster, further stratified for being direct PcG targets (left) or not (right). The full chart is available in Extended Data 5b. e- Transcriptional fold changes of genes involved in the JAK-STAT signaling pathway upon ph-KD. Direct PcG targets (+) are indicated in the right column.

● 4) Lack of follow up on spz3 and Dcr-2 as the evidence these aren't driving a phenotype is lacking.

We did not follow up on these genes because none were previously linked to tumorigenesis nor showed high allelic frequencies: *Dcr-2* allelic frequencies ranged between 4.3% and 12.5 % (depending on the sample), and *spz3* between 10% and 24.6%. Moreover, no SNV/InDel was detected in these two genes in any of our 6 new tumor samples (generated via *ph-KD* at the early L3 stage). Please also refer to response to point #2 for a detailed description of gDNA sequencing results.

● 5) It is intriguing that extended data figure 2D is similar to main figure 2E.

Extended data Fig 2D shows the size of ED after constant or transient *Psc/Su(z)12-KD*, another canonical member of the PRC1 complex, while the previous version of the main Fig. 2E (**now shown in the main Fig. 5b**) shows the size of ED after concomitant expression of *gfp+w* (control), *zfh1+w*, *gfp+ph* and *zfh1+ph* constant RNAi. Even if the plots might look visually similar, they come from independent datasets and the distributions are indeed different.

Referee #2 (Remarks to the Author):

While genetic mutations in epigenetic regulators are well-known to contribute to cancers, it is not clear whether transient epigenetic changes can lead to cancer development. The authors have addressed this important question using *Drosophila* as a model system. They show that transient depletion of PRC1 is sufficient for tumor development and that the generated tumors are transplantable. They determine the transcriptional changes induced by the transient depletion of PRC1 and define reversible and irreversible genes upon transient PRC1 depletion. They suggest that tumor formation is mediated by failure to repress irreversible genes and provide data supporting this notion by showing that knockdown of *zfh1*, one of the identified irreversible genes, can partially rescue the PRC1 depletion phenotype.

The authors perform several experiments to understand how the transient depletion of PRC1 results in tumor formation. They hypothesize that weaker binding of PRC1 to the irreversible genes as well as the binding of transcriptional factors into these genes cause sustained activation of these targets. As a result, aberrant growth is observed, and tumor formation is triggered.

The authors address a very interesting and outstanding question in cancer biology, and their observations suggest that transient epigenetic changes are indeed sufficient for tumor development and initiation. However, we have several comments that we would like the authors to address:

We would like to thank the reviewers for their appreciation of the work and for raising insightful points that gave us the chance to perform crucial experiments. We hope that they will find our responses convincing.

Reviewer#2 major comment 1.

Mechanism: The authors write their results show (lines 182-184) “an uncoupling between the impact of transient *ph*-KD on transcription and chromatin, whereby irreversible transcriptional changes drive tumorigenesis despite the reestablishment of an essentially normal chromatin landscape at Polycomb target genes.” The question is how do the authors imagine this? Do the authors think that the *zfh1* gene is expressed when bound by PH (PRC1), PRC2 and enriched for H3K27me3 and H2AK118ub1? The idea that a gene is expressed while bound by Polycomb group proteins and carrying the Polycomb modifications is not supported by the literature. If the authors believe this is the case, they should provide data supporting this point. An alternative hypothesis is that the *zfh1* gene expression is expressed in few cells that remain positive for H3K27ac. If this is the case, then it would be important to show that PH is expressed in the cells that also express *zfh1*.

We thank the reviewers for highlighting this intriguing aspect. When looking at the *zfh1* locus, we indeed see that some H3K27Ac remain after transient *ph*-KD, but also that a substantial amount of the H3K27me3 assembles into a detectable domain (Figure 3d):

Figure 3: PcG repressive landscape is restored after transient *ph-KD*.

d- Screenshot of PH ChIP-Seq, H3K27me3, H2AK118Ub and H3K27Ac CUT&RUNs tracks at representative irreversible (left) or reversible (right) loci under the indicated conditions (left).

However, ZFH1 staining shows that the protein increases in most of the cells (Extended Data Fig. 5d), precluding the idea of a small subset of cells expressing the corresponding gene:

Extended Data Fig. 5: The ZFH1 protein is irreversibly increased after transient *ph-KD*

d- ZFH1 immunostaining (in red) after no *ph-KD* (control), constant or transient *ph-KD*. Tissues were counterstained with DAPI (in blue). Scale bars: 10 μ m

Besides the few irreversible loci showing persistent active marks after transient *ph-KD* (such as *zfh1*), most do not: the *upd* locus stays significantly upregulated in the presence of detectable levels of H3K27me3 and no H3K27Ac peaks (see Fig. 3d panel on the previous page). In previous publications, we and others have shown that the presence of PRC1 was not incompatible with low levels of ongoing transcription (Loubiere et al., Science Advances 2020). Despite showing significant fold changes compared to control, we wish to emphasize that the levels of transcription (as measured by FPKMs) are relatively low at irreversible genes compared to the whole distribution of FPKMs after transient *ph-KD*:

Figure to reviewer

RNA-Seq fold change compared to temperature-matched control (\log_2 , left) and FPKMs (right) after transient *ph-KD* for all genes (in white) or PcG-bound genes that are Unaffected (in grey), irreversible (in pink) or reversible (in green).

Together, these observations suggest that weak levels of transcription are compatible with the presence of PcG marks. In the future, it would be interesting to study the detailed underlying mechanism, a feat that extends beyond the scope of the current manuscript.

Reviewer#2 major comment 2.

Related to the first point, Fig. 1B shows the WB result for PH level after transient depletion. In line 74 the authors write that the PH protein level is as returning to the normal level following transient depletion. However, the data shows a slightly decreased level of PH in the transient KD samples. This could indicate that PH expression is not re-established in all cells, which may therefore be the cells that contribute to tumor development and maintenance.

To address this point, we performed the WB shown in old Figure 1b in triplicate and show the corresponding quantification in the new Extended Data Fig 1d:

Extended Data Fig. 1: Transient PH depletion generates neoplastic tumors that persist after PH recovery.

d- Quantification of the Western blot shown in Fig. 1b. Error bars represent the standard error of the mean for three independent experiments. Dunnet's test: ns= not significant, **pval<0.01. e- Western blot showing PH protein from EDs subjected to 24h white-KD (w-KD, control) or *ph*-KD followed by 0h, 24h, 48h and 72h of recovery at 18°C (see bottom axis). This time course illustrates the acute depletion and quick recovery of the PH protein after *ph*-KD.

We also performed a time course experiment to measure PH levels after 0h, 24h, 48h and 72h of recovery at 18 °C following a 24h *ph*-KD at the early L3 stage and show that, after 48h of recovery, the PH protein is back at levels that are comparable to white-KD (w-KD) control tissues (see Extended Data Fig. 1e above).

Furthermore, PH immunostaining did not show any evidence of cell-to-cell variation after transient *ph*-KD, including in cells located in the middle of the tumour mass (Extended Data Fig. 1c):

Extended Data Fig. 1: Transient PH depletion generates neoplastic tumors that persist after PH recovery.

c- PH immunostaining (in red) after no *ph*-KD (control), constant or transient *ph*-KD. Tissues were counterstained with DAPI (in blue).

Altogether, these results argue against substantially lower PH levels after transient *ph*-KD or the existence of a subset of “non-recovering” cells. This is consistent with the fact that we observed over-proliferation throughout the whole tissue already 24h after a 24h *ph*-KD, which would be difficult to reconcile with the idea of a subset of cells driving the tumor (new Fig. 1f):

Figure 1: Transient PRC1 depletion is sufficient to initiate tumors.

f- EDU staining (in green) imaged 0h (left) and 24h (right) after 24h of w-KD (control, top) or *ph*-KD (bottom).

Nevertheless, we do believe that it would be highly interesting to study whether these tumours show increased cell-to-cell variations compared to normal tissues, an interesting area of study for future research.

Reviewer#2 major comment 3.

PH is part of the PRC1 complex that catalyzes H2K118Ub. Therefore, the authors should include H2K118Ub screen shots in Figure 3d. In Figure 4, the authors show ChIP-seq plots for Polycomb and PcG recruiter proteins. To get a comprehensive view of the role of these proteins in regulating reversible and irreversible genes, it would be highly relevant to show the binding profiles of these proteins on irreversible and reversible genes as done in Figure 3d for PH, H3K27me3 and H3K27ac.

Following reviewers' advice, we added the H2AK118Ub tracks in the revised version of the Figure 3d:

Figure 3: PcG repressive landscape is restored after transient *ph*-KD.
 d- Screenshot of PH ChIP-Seq, H3K27me3, H2AK118Ub and H3K27Ac CUT&RUNs tracks at representative irreversible (left) or reversible (right) loci under the indicated conditions (left).

In the previous version of our manuscript, we did not have ATAC-Seq data to focus on the regulatory elements that might dictate irreversible versus reversible transcriptional changes. Therefore, we could only focus on 26 irreversible and 32 reversible PH-bound promoters, indicating a slight yet significant difference in PH binding but also for the PcG recruiter proteins COMBGAP, PHO, SPPS and TRL (old figure 4). During the revision process, we performed ATAC-Seq after no *ph*-KD (control), constant or transient *ph*-KD and saw that the difference in PH binding that we reported was reflected by comparable differences in accessibility:

Figure to reviewer: PH differences at the promoter of PcG-bound irreversible/reversible promoters reflect differences in accessibility
 PH ChIP-Seq (left) and ATAC-Seq average tracks in no *ph*-KD (control) tissues, anchored at the TSS of irreversible (in purple) and reversible PH-bound promoters (in green) The average signal is shown (line) ± standard error (transparent polygon) and the distance to the TSS is shown on the x axis. The signal was quantified at the regions highlighted with dotted lines (see corresponding box plots on the right, depicting the median (line), upper and lower quartiles (box) ±1.5x interquartile range (whiskers); outliers not shown). Two-sided wilcoxon test, **p*val<0.05; N.S- Not significant

Because accessibility is known to confound ChIP-Seq analyses, we aimed at assessing whether stronger PH/PRC1 binding would be a *bona fide* feature of reversible gene loci using a more robust approach. To do so, we assigned PH peaks to their closest TSS – with a

maximum distance of 25kb – which allowed the identification of 91 PH peaks in the vicinity of PcG-bound Irreversible genes and 113 peaks assigned to PcG-bound reversible genes (see new Extended Data Fig. 6e):

Extended Data Fig. 6: PcG epigenetic landscape is essentially re-established after transient *ph*-KD

e- Average PH ChIP-Seq signal around PH peak summits (x axis) after no *ph*-KD (top), constant (middle) or transient *ph*-KD (top). PH peaks were stratified based on the closest TSS with a maximum of 25kb distance: peaks assigned to irreversible and reversible genes are shown in pink and in green, respectively. For each condition, the average signal is shown (solid line) \pm standard error (shaded area) and the distance to the TSS is shown on the x axis. The signal was quantified at the regions highlighted with dotted lines (see corresponding box plots on the right). Two-sided Wilcoxon test: N.S.- Not significant.

With this increased set of peaks assigned to irreversible and reversible genes, we did not detect substantial differences in PH levels. We acknowledge this point at line 214 of the revised manuscript: “The analysis of PH binding levels at PH peaks located ± 25 kb from the transcription start sites (TSS) of reversible (n= 113) and irreversible (n= 91) genes revealed that the two groups of binding sites are associated with similar levels of PH, both in control EDs (no *ph*-KD) and after transient *ph*-KD (see Extended Data Fig. 6e and Methods). This is consistent with the levels of H3K27me3 and H2AK118Ub repressive marks, which are again similar (Extended Data Fig. 6a)”:

Extended Data Fig. 6: PcG epigenetic landscape is re-established after transient *ph*-KD.

a- PH ChIP-Seq (top), H3K27me3 (2nd row), H2AK11Ub (3rd row) and H3K27Ac (bottom) CUT&RUN average tracks, anchored at the TSS of PcG-bound irreversible (in pink), reversible (in green) and unaffected genes (in grey) after no *ph*-KD (control, on the left), constant (middle) or transient *ph*-KD (right). For each condition, the average signal is shown (solid line) \pm standard error (shaded area) and the distance to the TSS is shown on the x axis. The signal was quantified at the regions highlighted with dotted lines (see corresponding box plots on the right). Two-sided Wilcoxon test: **p*val<0.05; N.S- Not significant.

In light of these new results, scrutinizing differences in PcG recruiters binding seemed unlikely to provide an explanation for the difference between reversible and irreversible genes. Instead, we performed ATAC-Seq after no *ph*-KD (control), constant and transient *ph*-KD to assess whether specific Transcription Factors (TFs) might support irreversible transcriptional changes.

This way, we indeed identified a set of irreversible ATAC-Seq peaks, showing increased accessibility after Transient *ph*-KD compared to control tissues (**new Figure 4a**):

Figure 4: Chromatin accessibility changes underlie reversible and irreversible transcriptional changes.
a- Clustering of ATAC-Seq peaks showing significant changes after constant or transient *ph*-KD. **b-** Over-representation of genes associated to irreversible (top), reversible (middle) or decreased (bottom) ATAC-Seq peaks, for each of the six RNA-seq clusters defined in Figure 2b. One-sided Fisher's exact test: N.S- not significant, * p val $<5e-2$, *** p val $<1e-3$, **** p val $<1e-5$. **c-** Fraction of TSS-distal peaks per cluster (>1kb). **d-** Screenshot of ATAC-Seq tracks after no *ph*-KD (control, on top), constant (middle) or transient (bottom) *ph*-KD, at the irreversibly upregulated *upd3* gene (left) and the reversibly upregulated *Ubx* gene (right). **e-** Normalized enrichment scores of the DNA binding motifs found at each cluster of ATAC-Seq peaks (± 250 bp, x axis). **f-** Linear model *t*-values of DNA binding motifs associated with increased (positive *t*-values) or decreased (negative-*t* values) accessibility after transient (x axis) or constant *ph*-KD (y axis). Only motifs with a significant *p*.value $<1e-5$ in at least one of the two linear models are shown. **g-** Fold changes at ATAC-Seq peaks (y axis) upon constant (left) or transient (right) *ph*-KD, depending on the number of caudal (*cad*) motifs they contain (x axis). Two-sided Wilcoxon test: N.S- not significant, *** p val $<1e-5$. **h-** Fold changes at ATAC-Seq peaks (y axis) upon transient *ph*-KD, depending on the number of Stat92E (left, in orange) or *zfh1* (right, in blue) motifs they contain (x axis). Two-sided Wilcoxon test: ** p val $<1e-2$, **** p val $<1e-5$.

Interestingly, irreversible peaks are specifically enriched in the vicinity of irreversible genes – while a set of reversible peaks are over-represented around reversible loci – and are typically found more than 1kb away from the closest TSS (Figure 3c-d). Together, these results suggest that we identified a set of transcriptional enhancers that get activated after transient *ph*-KD and boost the transcription of irreversible genes from a distance, preventing their repression.

DNA sequence analysis of these putative regulatory regions identified a strong enrichment for the DNA binding motifs of STAT92E (the key effector TF of the aberrantly activated JAK-STAT pathway), JRA and KAY (the *Drosophila* homologs of the AP-1 transcriptional activator

downstream of the JNK pathway), as shown in the **new Figure 4e**. Linear regression analyses of ATAC-Seq fold changes showed that *Stat92E* and *zfh1* DNA binding motifs are among the best predictors of increased *versus* decreased accessibility after transient *ph*-KD, respectively (**Figure 4f,h**). On the other hand, increased accessibility at reversible peaks was associated with DNA binding motifs associated to HOX genes (**Figure 4e-g**), which are mostly upregulated after constant *ph*-KD and are likely dispensable for growth of EICs (**revised Extended Data Fig. 4b**):

Extended Data Fig. 4: Transcriptional defects after constant or transient *ph*-KD.

b- Transcriptional fold changes after no *ph*-KD (control), constant or transient *ph*-KD (see x axis) of PcG core components, Hox genes (the canonical targets of PcG repression), key genes that regulate eye disc development and JNK pathway core members.

These new results are described in detail in a **new paragraph entitled “Specific chromatin opening sites mark reversible and irreversible loci”** (line 213), in which we conclude that the “recruitment of AP-1 and STAT92E at irreversible peaks could maintain irreversible genes in an activate state, potentially by maintaining chromatin open at their cis-regulatory regions” (line 249).

Reviewer#2 major comment 4.

In Extended Data Figure 4c, the authors shows that ZF1 levels are upregulated more in transient KD samples compared to the constant KD sample. Could the authors please comment on this observation?

The reviewers are correct in this point, which we confirmed both when analyzing RNA-seq and in immunostaining experiments (in the revised version, Extended Data Fig. 4 was moved to Extended Data Fig. 5). Since little is known on the regulation of *zfh1* transcription, it is not possible to explain how differential upregulation is achieved yet. One possibility is that, among the massive number of genes that are upregulated after constant *ph*-KD, specific transcriptional repressors – yet to be identified – might dampen *zfh1* transcription.

Reviewer#2 major comment 5.

Fig 2e-h show the partial rescue achieved by *zfh1* knockdown (KD) in the PRC1 depletion system. Did the authors test the rescue capacity of other identified irreversible target genes such as *upd2*, *upd3* and *Ets21C*? Can the combination of KD of several of these targets enhance the rescue phenotype?

We thank the reviewers for proposing these interesting experiments. We tested whether concomitant *Ets21C*-KD would rescue the phenotype, but a double RNAi did not reveal substantial rescue, as the tissue still showed an aberrant morphology and overgrowth:

Figure to reviewer: concomitant *Ets21C* KD does not rescue the phenotype of *ph*-KD tumor
 DAPI staining of EDs after constant *gfp*+*w*-KD (left), *Ets21C*+*w*-KD (2nd column), *gfp*+*ph*-KD (3rd column) and *Ets21C*+*ph*-KD (right).

upd genes were previously shown to be implicated in the growth of tumors driven by the mutation of another PRC1 member, *Psc-Su(z)2*, as noted in a previous paper by Classen et al⁴: “a deletion removing all three *upd* genes partially rescued the pupal lethality induced by the presence of *Psc-Su(z)2* eye tumors and caused a mild but significant reduction in tumor size”. Importantly, *upd* genes, the upstream ligands of the JAK-STAT pathway, are direct PcG targets irreversibly upregulated after transient *ph*-KD, as we tried to clarify in the revised version of Figure 2e:

Figure 2: EICs show irreversible transcriptional changes.
 e- Transcriptional fold changes of genes involved in the JAK-STAT signaling pathway upon *ph*-KD. Direct PcG targets (+) are indicated in the right column.

We therefore set out to assess the role of the JAK/STAT downstream effector, the *STAT92E* TF, which would demonstrate that the aberrant activation of the JAK-STAT pathway is a main driver of the tumor phenotype. Unlike *upd* genes, *Stat92E* is not a direct PcG target – consistent with the fact that its activation happens at the post-translational level – and directly testing its contribution to the phenotype was crucial. Importantly, we found that *Stat92E*-KD significantly reduces the growth of constant *ph*-KD tissues and partially restores ED differentiation (new Figure 5a-b):

Figure 5: Tumor development requires *STAT92E* and *ZFH1*.
 a- DAPI (top, in grey) and *ELAV* (bottom, in magenta) staining of EDs after constant *gfp*+*w*, *Stat92E*+*w*, *zfh1*+*w*, *gfp*+*ph*,

Stat92E+ph and *zfh1+ph-KD* (see top labels). ELAV is used as a marker of terminally differentiated neurons. Scale bars: 100 μm (DAPI), 10 μm (ELAV). b- ED area distributions, as measured using DAPI stained tissues (number of measured EDs is reported in brackets).

To assess whether this rescue would still happen after transient *ph*-KD, we developed a new genetic system allowing for transient double RNAi. The results obtained with this system show that *Stat92E*-KD is also sufficient to reduce tumor growth after transient *ph*-KD and to partially rescue tissue differentiation (new extended Data Fig. 7):

Extended Data Fig. 7: Transient *Stat92E*-KD and *zfh1*-KD are sufficient to substantially rescue transient *ph*-KD tumor defects a-c DAPI (grey, a), F-actin (red, b) and ELAV (magenta, c) staining of EDs after transient *gfp*-KD (control), *Stat92E*-KD and *zfh1*-KD in the presence of a concomitant, transient depletion of white (*w*-KD, control, three first columns) or *ph* (*ph*-KD, last three columns). DAPI staining was used to assess the overgrowth of the different tissues, F-actin for apico-basal polarity defects and the neuronal marker ELAV for differentiation defects. d- ED sizes quantified as overall DAPI staining area for the different conditions, showing that transient *Stat92E*-KD and *zfh1*-KD decreased ED overgrowth following transient *ph*-KD. Two-sided Wilcoxon test: ***p*val<1e-2, ****p*val<1e-3, *****p*val<1e-5. Scale bars: 100 μm (a), 10 μm (b, c).

Together, these new results highlight two main facts. First, not all irreversible genes that are direct PcG target seem to play a major role in PRC1-dependent tumorigenesis, as exemplified by *Ets21C*. Systematic investigations will be needed to identify the most relevant ones. Second, the downstream consequences of the irreversible upregulation of a subset of genes, such as the *upd* ligands, have key functional implications: although the *Stat92E* gene is not a PcG target, its aberrant activation is required for tumour growth. The **results of these new experiments are presented in the new paragraphs entitled “Specific chromatin opening sites mark reversible and irreversible loci” and “STAT92E and ZFH1 are required for EIC development”** (as well as corresponding new Fig. 4 and 5).

Reviewer#2 major comment 6.

For the rescue experiment with *zfh1* KD (Fig 2e-h), the authors did not use their thermo-sensitive strategy. Can transient KD of *zfh1* be sufficient to get a similar rescue phenotype? This was an intriguing experiment that we introduced in the revised version of the manuscript (see the new Extended Data Fig. 7 at the end of the previous point). Indeed, concomitant *zfh1*-KD is sufficient to significantly reduce overgrowth and partially restore differentiation after transient *ph*-KD.

Additionally, for the rescue experiment, a negative control could be included such as depletion of a selected gene from Cluster 1 since the authors state that upregulation of these genes is an indirect effect of PRC1 depletion (Line 134).

Before discussing the result of the experiment that we did in order to address this intriguing point, let us present some new analysis. First, we now show in the **new Figure 2c** that the transient-specific genes from cluster 1 are significantly enriched for direct PcG target genes (for which at least 50% of the gene body is covered with the H3K27me3 repressive mark), although the enrichments are lower than for the reversible and irreversible clusters (of note, we renamed the RNA-Seq clusters to directly reflect their characteristic patterns, and “Cluster 1” became “transient-specific”, Fig. 2b):

Figure 2: EICs show irreversible transcriptional changes.

a- Number of differentially expressed genes after no ph-KD (control), constant and transient ph-KD. Transitions between upregulated (orange), unaffected (grey) and downregulated (blue) states are indicated by thin lines of the same respective colors. b- Clustering of differentially expressed genes after constant or transient ph-KD. c- Over-representation of direct PcG target genes ($\geq 50\%$ of the gene body overlaps a H3K27me3 repressive domain in control condition). One-sided Fisher's exact test: *** $pval < 1e-3$, **** $pval < 1e-5$.

Thus, we modified the text accordingly, line 141: “Interestingly, the upregulated clusters show stronger and significant over-representation of PcG target genes covered with H3K27me3 – the canonical mark of PcG-mediated transcriptional repression – in control EDs (Figure 2c). This suggests that their upregulation is a direct consequence of compromised PcG repression, although they retain distinct patterns”.

{REDACTED}

{REDACTED}

{REDACTED}

Reviewer#2 major comment 7.

In line 206, it is stated that motifs of JAK-STAT pathway downstream effectors are enriched in irreversible genes. Since *zfh1* KD gives only a partial rescue, can KD of *kayak* or *Jra* alone or in combination with *zfh1* give better rescue results?

We thank the reviewer for encouraging us to further investigate the role of JAK-STAT downstream effectors in tumor growth. In the previous version of our manuscript, we mistakenly qualified *kayak* and *Jra* of downstream effectors of the JAK-STAT pathway: they act downstream of the JNK pathway. We apologize for this mistake and corrected the manuscript accordingly. Of note, KAY and JRA DNA binding motifs (the Drosophila homolog of the AP-1/Fos transcriptional activator) are strongly enriched at Irreversible ATAC-Seq peaks (see **new Figure 4e**) and future studies should address their impact on tumor growth. However, combining a *kayak*-KD or *Jra*-KD together with *zfh1*-KD and *ph*-KD is extremely challenging technically, as it hits the limit of Drosophila genetics' capabilities. It requires to combine at least three different RNAi in one strain which, if we count the required controls targeting *gfp* or the *white* gene, would necessitate at least three complicated and possibly fragile stocks. We tried to establish these strains but they were not viable. Hence, we could not assess this point.

Instead, we focused on the JAK-STAT pathway, for which we identified direct PcG target whose upregulation likely explains its aberrant activity (namely *upd* ligands). We **improved Fig. 2e** to only show the core components of the pathway, their respective functions and whether they are direct PcG targets (covered with the H3K27me3 repressive mark):

Figure 2: EICs show irreversible transcriptional changes.

e- Transcriptional fold changes of genes involved in the JAK-STAT signaling pathway upon *ph*-KD. Direct PcG targets (+) are indicated in the right column.

The downstream effector of the JAK-STAT pathway is the STAT92E TF, whose motif is the most strongly enriched at irreversible ATAC-Seq peaks. In the revised version of the manuscript, we present a series of experiment indicating that STAT92E plays a key role in supporting the aberrant transcription of irreversible genes after transient *ph*-KD and that *Stat92E*-KD significantly reduces tumor growth and partially rescues the differentiation of the tissue (see response to point #3, **new Figure 4 and Figure 5**).

Reviewer#2 major comment 8.

Differential analysis of H3K27me3 and H2AK118ub binding upon constant or transient *ph*-KD (Fig 3e and Extended Data Fig 5f) indicates that there are only 15 regions with reduced H3K27me3 levels and no regions with reduced H2AK118ub upon transient PRC1 depletion even though there are target genes upregulated (Fig 2a-d). How do the authors explain this observation? Does it that mean the binding of PRC1 plays an essential role in the repression of these genes rather than the posttranslational modification?

For now, we don't have a clear molecular understanding of how ongoing transcription happens

in the presence of repressive marks (see response to point #1). However, we now found ATAC-Seq sites with increased accessibility after transient *ph*-KD that lay in the vicinity of irreversible genes. They are enriched for the DNA binding motifs of the STAT92E TF, the key effector of the JAK-STAT pathway, which might therefore support irreversible transcriptional changes (see response to previous points #3 and #7). Consistent with STAT92E acting as a transcriptional activator in *ph*-dependent tumours, we show in the new Fig. 5 that concomitant *Stat92E*-KD reduces overgrowth and partially restores differentiation after constant *ph*-KD (new Fig. 5a-b) but also brings about 50% of the genes that were upregulated back to control levels (new Fig. 5c):

Figure 5: Tumor development requires STAT92E and ZFH1.

a- DAPI (top, in grey) and ELAV (bottom, in magenta) staining of EDs after constant *gfp+w*, *Stat92E+w*, *zfh1+w*, *gfp+ph*, *Stat92E+ph* and *zfh1+ph-KD* (see top labels). ELAV is used as a marker of terminally differentiated neurons. Scale bars: 100 μm (DAPI), 10 μm (ELAV). b- ED area distributions, as measured using DAPI stained tissues (number of measured EDs is reported in brackets). c- Number of differentially expressed genes after *gfp+ph* (tumors), *Stat92E+ph* and *zfh1+ph-KD*. Transitions between upregulated (orange), unaffected (grey) and downregulated (blue) states are indicated by thin lines of the same respective colors.

Among the “rescued genes” that were upregulated after constant *gfp+ph-KD* and recovered normal levels after *Stat92E+ph-KD* (i.e. switched to “unaffected”), irreversible genes were strongly over-represented, further suggesting that STAT92E aberrantly activates them after constant *ph*-KD:

Reviewer Figure #6: *Stat92E* RNAi rescues the expression of a subset of irreversible genes

Over-representation of “rescued genes” (upregulated after constant *gfp+ph-KD* and recovering normal levels after *Stat92E+ph-KD*) among the different RNA-Seq clusters from Figure 2b. One-sided Wilcoxon test, *****p*val<1e-5

Reviewer#2 major comment 9.

All the RNA-seq, ChIP-Seq and Cut&Run experiments have been performed at a later stage of development (8-11 days AEL). However, to be able to identify primary targets leading to

tumour initiation, analysis at the earlier stage can be performed in a time-course manner. This is an interesting point. Regarding ChIP-Seq and CUT&RUN data, it is challenging to dissect enough tissues for all these conditions at early time point because the EDs are still small. {REDACTED}

{REDACTED}

Referee #3 (Remarks to the Author):

Content

Non-genetic mechanisms may also play a role in the initiation of cancer, as significant epigenomic alterations can contribute to tumorigenesis. In this study using *Drosophila*, the authors conclude that temporarily disrupting transcriptional silencing mediated by Polycomb Group proteins leads to an irreversible switch to a cancer cell fate, even without driver mutations. This is linked to the activation of genes associated with tumorigenesis, such as JNK and JAK-STAT signaling pathways and the ZEB1 oncogene. The authors suggest that this altered cell fate can be inherited epigenetically and contributes to cancer development.

Impact

The current view in the field is that tumors arise from an accumulation of permanent mutations. This paper contradicts this notion and shed a completely new light onto tumor ontogenesis. It would have important implications for clinical approaches to diagnostics and treatment, as well as experimental approaches to their molecular analysis. This manuscript would therefore be suitable for publication in *Nature*.

General assessment

I would like to extend my congratulations to the authors for producing such a great manuscript. The data presented in this study is of outstanding quality, the scientific approach is thorough and the extensive controls provide a robust foundation for the authors' claims - while excluding alternative scenarios. I truly enjoyed reviewing it.

We thank the reviewer very much for their kind words and hope that the revised version of the manuscript will clarify key pending questions.

Reviewer#3 Major concern 1

However, I do have one major concern regarding the interpretation of the role of Polycomb, JNK, and JAK/STAT signaling in this process. My question is, to what extent is the origin of this heritable phenotype dependent on Polycomb? Is it possible that this is simply a heritable state that can also be induced by non-epigenetic JNK/JAK activating mutations and tumors? Addressing this issue will significantly strengthen the authors' conclusions and provide a more comprehensive understanding of the mechanisms underlying this phenomenon.

We thank the reviewer for pointing out this key aspect of the present study and to give us the opportunity to clarify key points.

The phenotype described in the current manuscript is polycomb-dependent in essence, as it was triggered by a transient perturbation of PRC1. The term "Epigenetically Initiated Cancer" was coined to reflect the fact that the phenotype can be propagated after the initial trigger has disappeared, i.e. the transient depletion of *ph-p* and *ph-d*, not because it was initiated via depleting epigenetic factors.

However, the reviewer is making a very good point: although the initial trigger is the transient perturbation of the PcG machinery, what are the mechanisms supporting the maintenance of the tumors' transcriptome? In particular, would they boil down to activation of the JAK/STAT and JNK pathways and thus, are the EICs essentially the same as tumors induced by mutations activating JAK/STAT and/or JNK pathways? This is particularly interesting and, during the revision process, ATAC-Seq experiments in the different conditions as well as new rescue experiments taught us a lot about the mechanisms underlying EIC development and maintenance. We used these experiments to address the reviewers' comments as discussed below.

I will try to express this concern in more detail below:

Pinal et al. (2018) demonstrated in their study published in *Nature Communications* that transient JNK activation in conjunction with apoptosis inhibition can lead to the development of a heritable tumor state. (Please cite and reference this paper.)

We thank the reviewer for suggesting this and have now cited and referenced this significant paper.

However, Pinal et al only controlled the transient nature of JNK activation. As they used dronc mutant discs to suppress apoptosis, the tissue gained apoptosis resistance also during the subsequent recovery phase. Therefore, their model differs from the current study in that the perpetuation of the JNK/JAK activated tumor phenotype may be linked to continuous apoptosis resistance. Nonetheless, their findings raise an important question regarding the potential for a heritable tumor state to be induced by other (non-epigenetic) JNK/JAK activating mutations? This idea is particularly relevant in light of the authors' observations that Ph-tumors activate JNK and JAK/STAT signaling and demonstrate a high degree of immortality and apoptosis resistance. Therefore, it is possible that JNK/JAK activation in general, rather than transient Polycomb deregulation, is a central feature of a heritable tumor phenotype. The authors' own conclusions may support this theory:

Line 180 ff These results rule out a differential reprogramming of Polycomb-dependent histone marks upon PH depletion or their differential re-establishment upon reinstating PH as general explanations of the difference between reversible and irreversible genes. They also highlight a surprising uncoupling between the impact of transient ph-KD on transcription and chromatin, whereby irreversible transcriptional changes drive tumorigenesis despite the reestablishment of an essentially normal chromatin landscape at Polycomb target genes.

Besides few exceptions (such as the *zfh1* loci that acquires few irreversible H3K27Ac peaks after transient *ph*-KD), the PcG landscape is indeed remarkably recovered after a transient *ph*-KD (see Fig. 3 and Extended Data Fig. 6). Although transient *ph*-KD was the initial trigger that allowed the derailment of the transcriptome, finding the mechanisms allowing the maintenance of this acquired state was of key relevance and we have now new data that provide critical insights concerning this mechanism, as discussed below.

Line 205 ff This approach indicates that irreversible genes' REs are enriched for motifs associated to downstream effectors of the JAK-STAT pathway, including kayak (AP-1/Fos), Jra (Jun) and *zfh1*, the latter one showing a positive effect on tumor growth (Fig. 2e-g). Together, these results indicate a tight equilibrium, whereby less potent recruitment of PRC1 at the promoter of irreversible genes, coupled with the binding of TFs related to the JAK-STAT pathway, whose cognate motifs are not enriched in the vicinity of reversible genes, might prevent the re-silencing of irreversible genes upon PRC1 recovery.

Before discussing the role of JNK and JAK-STAT effector TFs, we would like to update the reviewer on the PRC1 binding at reversible vs irreversible genes that we reported in the previous version of the manuscript. At that time, we did not have ATAC-Seq data in our different conditions. Therefore, we could only scrutinize 26 irreversible and 32 reversible PH-bound promoters to understand the mechanisms that might support irreversible transcriptional changes, which came with limited statistical power (old figure 4). During the revision process we performed ATAC-Seq in all conditions, and saw that the difference in PH binding that we reported was reflected by comparable differences in local accessibility after no *ph*-KD (control):

Reviewer Figure #9: PH differences at the promoter of PcG-bound irreversible/reversible promoters reflect differences in accessibility

PH ChIP-Seq (left) and ATAC-Seq average tracks in no *ph*-KD (control) tissues, anchored at the TSS of irreversible (in purple) and reversible PH-bound promoters (in green). The signal was quantified at the regions highlighted with dotted lines (see corresponding box plots on the right).

Since accessibility is a confounding parameter in ChIP-Seq, we took a more robust approach to assess if less potent PRC1 binding is a major hallmark of irreversible loci. PH peaks were assigned to the closest TSS (maximum distance of 25kb) and we compared the peaks assigned to irreversible (n= 91) and reversible (n= 113) PcG-bound genes (**new Extended Data Fig. 6d**):

Extended Data Fig. 6: PcG epigenetic landscape is re-established after transient *ph*-KD.

e- Average PH ChIP-Seq signal around PH peak summits (x axis) after no *ph*-KD (top), constant (middle) or transient *ph*-KD (top). PH peaks were stratified based on the closest TSS with a maximum of 25kb distance: peaks assigned to irreversible and reversible peaks are shown in pink and in green, respectively. For each condition, the average signal is shown (solid line) \pm standard error (shaded area) and the distance to the TSS is shown on the x axis. The signal was quantified at the regions highlighted with dotted lines (see corresponding box plots on the right). Two-sided Wilcoxon test: N.S- Not significant.

Consistent with the comparable levels of H3K27me3 and H2AK118Ub between the two groups (in control and after transient *ph*-KD, **new Extended Data Fig. 6a**), PH levels were also similar and we modified the text accordingly. Therefore and thank to the reviewer concern, we can conclude that differential PH binding is not a major general determinant of the identity of reversible vs irreversible genes, despite the fact that at some irreversible loci PH binding is indeed relatively weak.

Most importantly, concerning JNK and JAK-STAT effector TFs, newly generated ATAC-Seq data identified 446 irreversible peaks with increased accessibility after transient *ph*-KD (**new Figure 4a**):

Figure 4: Chromatin accessibility changes underlie reversible and irreversible transcriptional changes.

a- Clustering of ATAC-Seq peaks showing significant changes after constant or transient *ph*-KD. b- Over-representation of genes associated to irreversible (top), reversible (middle) or decreased (bottom) ATAC-Seq peaks, for each of the six RNA-seq clusters defined in Figure 2b. One-sided Fisher's exact test: N.S- not significant, * p -val $<5e-2$, *** p -val $<1e-3$, **** p -val $<1e-5$. c- Fraction of TSS-distal peaks per cluster (>1kb). d- Screenshot of ATAC-Seq tracks after no *ph*-KD (control, on top), constant (middle) or transient (bottom) *ph*-KD, at the irreversibly upregulated *upd3* gene (left) and the reversibly upregulated *Ubx* gene (right). e- Normalized enrichment scores of the DNA binding motifs found at each cluster of ATAC-Seq peaks (± 250 bp, x axis). f- Linear model *t*-values of DNA binding motifs associated with increased (positive *t*-values) or decreased (negative-*t* values) accessibility after transient (x axis) or constant *ph*-KD (y axis). Only motifs with a significant *p*-value $<1e-5$ in at least one of the two linear models are shown. g- Fold changes at ATAC-Seq peaks (y axis) upon constant (left) or transient (right) *ph*-KD, depending on the number of caudal (*cad*) motifs they contain (x axis). Two-sided Wilcoxon test: N.S- not significant, **** p -val $<1e-5$. h- Fold changes at ATAC-Seq peaks (y axis) upon transient *ph*-KD, depending on the number of *Stat92E* (left, in orange) or *zfh1* (right, in blue) motifs they contain (x axis). Two-sided Wilcoxon test: ** p -val $<1e-2$, **** p -val $<1e-5$.

Irreversible peaks are significantly enriched in the vicinity of irreversible genes (while a set of reversible peaks are over-represented around reversible loci) and are typically found more than 1kb away from the closest TSS, suggesting that they correspond to transcriptional enhancers (new Figure 4b-d). They are strongly and specifically enriched for the DNA binding motifs of *Stat92E* (the TF of the JAK-STAT pathway) and of *Jra* and *kay* (the TFs of the JNK pathway, new Figure 4e). Furthermore, *Stat92E* DNA binding motifs were associated with increased accessibility after transient *ph*-KD (new Figure 4f,h), suggesting that sustained STAT92E binding might maintain transcription of irreversible genes upon restoration of normal PH levels.

Consistently, concomitant *Stat92E*-KD significantly reduces growth and partially restores differentiation after constant (new Figure 5a-b) *ph*-KD, and brings a substantial fraction of differentially expressed genes to control levels (new Fig. 5c):

Figure 5: Tumor development requires STAT92E and ZFH1.

a- DAPI (top, in grey) and ELAV (bottom, in magenta) staining of EDs after constant *gfp+w*, *Stat92E+w*, *zfh1+w*, *gfp+ph*, *Stat92E+ph* and *zfh1+ph-KD* (see top labels). ELAV is used as a marker of terminally differentiated neurons. Scale bars: 100 μm (DAPI), 10 μm (ELAV). b- ED area distributions, as measured using DAPI stained tissues (number of measured EDs is reported in brackets). c- Number of differentially expressed genes after *gfp+ph* (tumors), *Stat92E+ph* and *zfh1+ph-KD*. Transitions between upregulated (orange), unaffected (grey) and downregulated (blue) states are indicated by thin lines of the same respective colors.

Importantly, we additionally set up a new genetic system allowing for reversible double RNAi, and found that *Stat92E*-KD also reduces growth and partially restores differentiation in such transient depletion context (new Extended Data Fig. 7):

Extended Data Fig. 7: Transient *Stat92E*-KD and *zfh1*-KD are sufficient to substantially rescue transient *ph*-KD tumor defects a-c DAPI (grey, a), F-actin (red, b) and ELAV (magenta, c) staining of EDs after transient *gfp*-KD (control), *Stat92E*-KD and *zfh1*-KD in the presence of a concomitant, transient depletion of white (*w*-KD, control, three first columns) or *ph* (*ph*-KD, last

three columns). DAPI staining was used to assess the overgrowth of the different tissues, F-actin for apico-basal polarity defects and the neuronal marker ELAV for differentiation defects. d- ED sizes quantified as overall DAPI staining area for the different conditions, showing that transient *Stat92E-KD* and *zfh1-KD* decreased ED overgrowth following transient *ph-KD*. Two-sided Wilcoxon test: ***pval*<1e-2, ****pval*<1e-3, *****pval*<1e-5. Scale bars: 100 μ m (a), 10 μ m (b, c).

Altogether, these results converge towards a multistep model in which a transient disruption of PcG-mediated silencing is sufficient to irreversibly activate the JAK-STAT pathway and its downstream effector STAT92E (as well as, possibly, JNK signaling and its *Jra* and *kay* TFs, although we have no functional evidence supporting their role), whose binding prevents the re-repression of irreversible genes. For a detailed description of these new results, we invite the reviewer to read our new paragraph entitled “**Specific chromatin opening sites mark reversible and irreversible loci**” (line 213), where we conclude that the “**recruitment of AP-1 and STAT92E at irreversible peaks could maintain irreversible genes in an activate state, potentially by maintaining chromatin open at their cis-regulatory regions.**” (line249).

Thus, our results are compatible with the model that the reviewer brings up, at least regarding the JAK-STAT pathway. We note, however, that we could not definitely prove that the aberrant activation of this pathway would be sufficient to drive these tumors. Firstly, in the absence of a transient perturbation of the PcG machinery, it is unclear why *upd* genes (coding for the ligands of the JAK-STAT pathway) would get upregulated in the first place. Secondly, our work does not allow to formally address the functional role of the JNK pathway (whose core members are not upregulated, **new Extended Data Fig. 4b**). {REDACTED}

{REDACTED}

{REDACTED}

As a final note to this important reviewer issue, we want to stress that, even if the tumor phenotype was maintained *via* the aberrant activation of signaling pathway, this would still fit our definition of an “epigenetically initiated” state, since it can be maintained and propagated independently of the initial trigger and of any irreversible genetic event (see Fig. 1). {REDACTED}

Hence, I would love to see at least one experiment that demonstrates that a transient induction of a non-PRC1/2 tumor does (or does not) cause a heritable tumor state. Either outcome could help strengthen conclusions about the role of PcG or about the role of JNK/JAK in heritable tumor states. A 'straight-forward' experiment may be:

- transient activation of a UAS RasV12, UAS scrib RNAi model
 - transient activation of a UAS Zfh1 UAS hepACT or UAS Zfh1, UAS scrib RNAi model
- UAS-hopTum + UAS-hepACT + actGal4 + gal80Ts

The later option may actually help to better define the role of Zfh1 in heritable tumor states. It is acting strongly anti-apoptotic, so it may cooperate with activated JNK in tumor induction.

{REDACTED}

{REDACTED}

{REDACTED} Regarding the mechanism by which *zfh1* upregulation fosters *ph*-dependent tumorigenesis, we performed ATAC-Seq after no *ph*-KD and transient *ph*-KD and found the *zfh1* DNA binding motif to be associated with decreased accessibility in the tumor samples (**new Figure 4f,h**):

Figure 4: Chromatin accessibility changes underlie reversible and irreversible transcriptional changes.
f- Linear model *t*-values of DNA binding motifs associated with increased (positive *t*-values) or decreased (negative *t*-values) accessibility after transient (*x* axis) or constant *ph*-KD (*y* axis). Only motifs with a significant *p*-value $< 1e-5$ in at least one of the two linear models are shown. **g-** Fold changes at ATAC-Seq peaks (*y* axis) upon constant (left) or transient (right) *ph*-KD, depending on the number of caudal (*cad*) motifs they contain (*x* axis). Two-sided Wilcoxon test: N.S- not significant, $****p < 1e-5$. **h-** Fold changes at ATAC-Seq peaks (*y* axis) upon transient *ph*-KD, depending on the number of Stat92E (left, in orange) or *zfh1* (right, in blue) motifs they contain (*x* axis). Two-sided Wilcoxon test: $**p < 1e-2$, $****p < 1e-5$.

This result suggested that *zfh1* might act by reducing the accessibility of a subset of regulatory elements (such as enhancers and/or promoters), consistent with its known role of transcriptional repressor^{2,3}. To further understand its mode of action, we performed RNA-Seq and ATAC-Seq after constant *zfh1+ph*-KD. Our data show that *zfh1*-KD rescues the normal expression levels of about 50% of the genes that are differentially expressed after constant *gfp+ph*-KD (including a majority of down-regulated genes, **new Figure 5c**):

Figure 5: Tumor development requires STAT92E and ZFH1.
a- DAPI (top, in grey) and ELAV (bottom, in magenta) staining of EDs after constant *gfp+w*, *Stat92E+w*, *zfh1+w*, *gfp+ph*, *Stat92E+ph* and *zfh1+ph*-KD (see top labels). ELAV is used as a marker of terminally differentiated neurons. Scale bars: 100 μm (DAPI), 10 μm (ELAV). **b-** ED area distributions, as measured using DAPI stained tissues (number of measured EDs is

reported in brackets). c- Number of differentially expressed genes after *gfp+ph* (tumors), *Stat92E+ph* and *zfh1+ph-KD*. Transitions between upregulated (orange), unaffected (grey) and downregulated (blue) states are indicated by thin lines of the same respective colors. d- Number of ATAC-Seq peaks showing significant accessibility changes after *gfp+ph* or *zfh1+ph-KD*. Transitions between increased (orange), unaffected (grey) and decreased (blue) states are indicated by thin lines of the same respective colors. e- Fold changes at ATAC-Seq peaks between *zfh1→+→ph* and *gfp→-→ph-KD*, depending on the number of ZFH1 motifs they contain (x axis). Two-sided Wilcoxon test: *****pval*<1e-5. f- RNA-Seq fold changes upon *gfp+ph-KD* (x axis) of genes associated to ATAC-Seq peaks that are decreased (in blue), unaffected (in grey) or increased (in orange) after *zfh1+ph-KD* compared to *gfp+ph-KD* (y axis). g- Top enriched GO terms for genes associated to ATAC-Seq peaks containing at least one ZFH1 motifs and showing significantly increased accessibility after *zfh1+ph-KD* compared to *gfp+ph-KD*.

zfh1-KD also brought a large majority of the ATAC-Seq peaks of decreased accessibility upon *ph-KD* back to control levels (new Figure 5d). This chromatin “re-opening” was correlated with the presence of ZFH1 DNA binding motifs (new Figure 5e), consistent with its association to reduced accessibility after transient *ph-KD* (new Figure 4h). In order to link accessibility to transcriptional changes, we assigned ATAC-Seq peaks that were increased upon *zfh1+ph-KD* compared to *ph-KD* (which we hypothesize to correspond to regulatory elements that are abnormally closed by *zfh1* in tumors) to the closest TSS (+/- 25kb). Interestingly, the corresponding genes were significantly downregulated after constant *ph-KD* and were implicated in ED development differentiation and neuron differentiation (new Figure 5f,g). Altogether, these results suggest a model in which the aberrant expression of *zfh1* – which translates into higher protein levels after transient *ph-KD* (old Ext Data Fig. 4c,d, corresponding to the new Extended Data Fig. 5c,d) – fosters the aberrant repression of genes responsible for the normal development and differentiation of ED, preventing differentiation and in turn stimulating neoplastic growth. These results are discussed in the paragraph entitled “STAT92E and ZFH1 are required for EIC development” (line 257).

Reviewer #3 Major concern 2

Please be more clear and precise about the distinction of JNK (*kay*, *jra*) and JAK/STAT signaling signatures and the effects they have on tumor growth. It is clear from the literature that both pathways are strongly co-regulated and active in tumor development. However, *kay* and *jra* are effectors of JNK signaling and should not be presented as JAK/STAT effectors. We apologize for this confusion and corrected the text accordingly. We also revised Figure 2e to focus on the core member of the JAK-STAT pathway, highlighting their respective roles and whether they are PcG targets or not:

Figure 2: EICs show irreversible transcriptional changes.

e- Transcriptional fold changes of genes involved in the JAK-STAT signaling pathway upon *ph-KD*. Direct PcG targets (+) are indicated in the right column.

In the revised version of our manuscript, we mostly focused on this pathway because we could link its aberrant activation to primary transcriptional defects. Although we did not see any transcriptional defect regarding the core members of the JNK pathway, we show their fold changes in the new Extended Data Fig. 4b (see above) and future studies should address their functional relevance in this system.

Reviewer #3 Major concern 3

The authors of the study have introduced their model really well and make every effort to provide a number of relevant controls that demonstrate control over 'transient expression'. They have rigorously tested the initial knock-down as well as the reversal of knock-down at transcript and protein levels. Throughout the manuscript, they introduce highly relevant control experiments (for example to test for mutagenicity of PhKDs) to exclude alternative scenarios. This thorough characterization provides an excellent foundation for all their conclusions. However, there are two points that require further clarification:

Concern 3, sub-point 1)

At what timepoint after knock-down does Ph come back? How long are these tumors without PH?

To judge the degree of Ph independence, it would be important to provide some approximate measure of when Ph levels are restored. Could the authors conduct an anti-PH immunostaining time course after a transient L1 induction at day 3, 5, 7?

We agree that a better characterization of the system's kinetic behaviour is important. While we are not able to dissect eye disc tissues at the L1 stage, which consist of only few cells, we know that similar tumors are obtained upon transient depletion of PH at the early L3 stage. We therefore performed a time course after a 24h KD at the early L3 stage and 0h, 24h, 48h and 75h of recovery at 18°C. The data shown below are now presented in the **new Ext. Data Fig. 1**:

Extended Data Fig. 1: Transient PH depletion generates neoplastic tumors that persist after PH recovery.

e- Western blot showing PH protein from EDs subjected to 24h white-KD (w-KD, control) or ph-KD followed by 0h, 24h, 48h and 72h of recovery at 18°C (see bottom axis). This time course illustrates the acute depletion and quick recovery of the PH protein after ph-KD. f- Hatching rate after constant or transient ph-KD.

After 48h of recovery, PH reaches similar levels to those found in the control *white*-KD sample, suggesting that complete restoration is reached somewhere between 24 and 48h after the end of the *ph*-KD.

Concern 3, sub-point 2)

The numbers presented in Ext Data Fig 3b suggest that all PRC1 components, and especially the two other potent PRC1 tumor suppressors Psc and Su(z)2, are mildly decreased in day 9 and day 11 flies. Could you please comment on this? Especially since your constant PhKD only reduces Ph by -0.3/-0.5.

Psc and *Su(z)2* are paralogous genes that are functionally redundant, and removing one of them is not sufficient to induce tumors. In our system, only the reduction of *Su(z)2* is significant ($p_{adj} = 0.0016$) and rather modest (-0.37) after transient *ph*-KD (d11). Hence, although we cannot totally exclude reduced *Su(z)2* levels could favor the development of the tumor, it is unlikely to be the primary cause.

On the other hand, *ph* reduction measured by RNA-Seq should be taken with a grain of salt, as is always the case for depletions made by using RNAi. The best measures of PH depletion are those analyzing protein levels, in particular WB and staining, which show an almost complete loss of the protein 24h *ph*-KD (see previous point).

Minor comments

The paper is well-written and mostly easy to follow and understand. However, at times, the writing can be dense, with sentences containing multiple ideas or lacking necessary information. While it is important to keep the manuscript short, sharpening the focus of certain sentences would enhance readability. Additionally, providing more information in the text and legends, when appropriate, would be beneficial.

Some more details below:

Introduction

While the first paragraph of the introduction focuses on epigenetic driver mutations, it is not until the second paragraph that the actual problem addressed in the study is described: epigenetic changes occurring in the absence of driver mutations. To more effectively introduce the focus of the paper, it may be helpful to incorporate some of the compelling examples from the discussion into the introduction. This may orient readers to the central questions addressed more quickly.

It is indeed important to drive the reader to the question asked as quickly as possible. We found it difficult to shorten the first paragraph but in the revised version of the manuscript we merged the second paragraph with the first in order to emphasize that the role of epigenetics in cancer induction and progression is the central point addressed by this manuscript.

Results

- Line 65

One sentence to introduce *ph-p*, *ph-d* before Line 65 would help to ease the reader into the genetic set-up and model.

We thank the reviewer for this suggestion and considered how to best do it. The only place where we would be able to define the two homologs would be at line #59-60 where we speak of PRC1 subunits, but this would make the introduction statement more convoluted and gain only 5 lines. Therefore, we left the text as it was but are open to suggestions.

- Line 70: As expected, 100% of EDs collected at the third Larval stage (L3) after constant PH depletion EDs are transformed into tumors (Figure 1c,d; Extended Data Fig. 1d-f).

No actual quantifications are shown here. Rephrase the sentence!

We actually did quantify the penetrance for an experiment in which we performed 150 tissue preparations in each of three biological replicates, for control *white-KD* as well as constant or transient *ph-KD* followed by restoration, at day 9 or day 11. These are the conditions which we used for RNA-seq experiments. All tissues of the constant or transient *ph-KD* were severely malformed and overgrown, suggesting that the phenotype is completely penetrant. We now state this in the manuscript.

- Line 71

Why are your pupal hatching rates so low (30-60%) for *wKD* controls?

This is due to the fact that we are using 29°C for the RNAi in lines that contain several transgenes that are used to elicit the depletion. In order to make sure that the depletion is robust, we enforced a temperature that is at least of 29°C at all times during the depletion. This likely involves times at which the temperature actually crosses this threshold by as much as one degree Celsius. These conditions cause the relatively low viability also of the control lines, although no tumors are ever detected in them.

- Line 99

Please provide more information about the graph in Figure 1 H (also Ext Data Fig 4A) in the

figure legend. I can guess but I do not completely understand how to read this graph. What is the set size? Why is one bar red? The axis label as ‘number of mutated genes’ does not make much sense (?).

Previous Fig. 1H is not shown anymore, but the new Extended Data Fig. 3a is very similar, and we improved the corresponding caption as follows: “**SNV/InDels overlaps between all sequenced gDNA samples. Each vertical bar corresponds to an intersection (corresponding samples are shown below) and horizontal bars (bottom left) show the total number of SNV/InDels find in each sample. Only intersections containing ≥ 40 SNV/InDels are shown and SNV/InDels that are specific to one sample are indicated in orange (68.1% of all detected SNV/InDels)**”. All panels showing similar “upset plots” were modified accordingly (such as the old Extended Data Fig. 4a, which moved to Extended Data Fig. 5a).

- Line 117 As expected, both systems are hardly distinguishable at 18°C (no RNAi), as well as upon transient w-KD (Figure 2a; Extended Data Fig. 3a).

Briefly mention which assay you use to ease the reader into the data.

Extended Data Fig. 3a moved to Extended Data Fig. 4a, and the corresponding caption was updated as follows: “**Principal Component Analysis (PCA) of RNA-Seq normalized counts. Each dot corresponds to one biological replicate. Close distance between samples mirrors their similarity, showing that control samples (no w-KD, no ph-KD) and transient w-KD are highly similar**”.

- Line 137 Cluster 5 includes many TFs that are bound by both PRC1 and PRC2 and correspond to canonical PcG targets such as *en*, *eve*, *wg*, *Ubx* and *Scr*.

Wg is not a TF. Remove or rephrase.

We rephrased the sentence accordingly line 144: “**The “reversible” cluster includes canonical PcG target genes such as *en*, *eve*, *wg*, *Ubx* and *Scr***”.

- Line 164 As expected, reversible genes ($n= 68$) show no expression changes after transient ph-KD, whereas irreversible genes ($n= 61$) remain upregulated (Extended Data Table 2, Figure 3a).

Is this not just simply a consequence of your filtering the data rather than an expectation?

In the previous version of the manuscript, the “as expected” was referring to the fact that clustering was based on the RNA-Seq log₂ fold changes. Hence, it was trivial to compare the fold changes of the two groups, as this was the initial discriminant used to segregate them. Because this was a circular argument, we removed this statement from the revised version as well as the old panel from Fig. 3a. Instead, we think that the **new Fig. 3a** (corresponding to the old Fig. 3b) showing the absolute transcriptional levels of the two groups in the different conditions (FPKMs) is more informative.

- Line 161 ff

I found the text and figure legend associated with Figure 3 to be the most challenging to understand, and would appreciate some clarification and simplification in the writing. Specifically, it would be helpful to expand the information in the figure legends for Figure 3, as it took some time for me to infer the details. From the text, for example, I had difficulty discerning whether the analysis was referring solely to PcG target genes or to all regulated genes, which made it hard to follow specific conclusions. To improve clarity and accessibility, I recommend revising the text and figure legend associated with Figure 3.

The Figure 3 was substantially updated and the caption rephrased accordingly, trying to improve its clarity.

- Line 187 ff

The conclusion drawn in Figure 4 is that reversible genes exhibit increased binding by PRC1 components and repressive transcription factors. Yet, H3K27me₃ levels appear to not play a strong role. This leads to the question of how much PRC2 is required in this process, given

the apparent lack of H3K27 me3 involvement. To provide context for this finding, it would be useful to briefly point to the previous literature on the differences between PRC1 and PRC2, and their respective roles in regulating gene expression.

As previously discussed (see response to point #1), we extensively revised this point. Using a more robust approach, we found that PRC1 binding levels at reversible and irreversible genes are similar, even if a minor difference in H3K27me3 and H3K27Ac levels can be detected (see **new Fig 3g,h**).

- Line 202 ff

Why does pho show up enriched at reversible genes (Fig 4E) but its motif is enriched at irreversible genes (Fig 4 G)? The same discrepancy is true for NF-YA and CTCF between Fig 4F and Fig 4G. What am I missing? Could you comment on this please?

As explained in the response to point #1, newly generated ATAC-Seq data allowed us to precisely map 446 irreversible regulatory elements associated to irreversible transcriptional changes, which drastically increased our statistical power. This improved approach did not identify any motif associated to key PcG recruiters, consistent with the fact that PH binding and PcG marks are comparable between irreversible and reversible genes, with the *zfh1* locus behaving like an outlying exception (see response to point #1).

- Line 205 This approach indicates that irreversible genes' REs are enriched for motifs associated to downstream effectors of the JAK-STAT pathway, including kayak (AP-1/Fos), Jra (Jun) and *zfh1*, the latter one showing a positive effect on tumor growth (Fig. 2e-g). Actually, kayak (AP-1/Fos), Jra (Jun) are effectors of JNK signaling. That JNK and JAK/STAT are tightly co-regulated in tumors is clear but this sentence here is a misrepresentation of these classical tumor-promoting pathways.

Thanks for pointing this out. We corrected the text accordingly, line 240: **“On the other hand, irreversible peaks are enriched for Jra and kay motifs, the Drosophila homologs of AP-1, which are the main TFs of the oncogenic JNK signaling pathway”**.

- Line 218 Importantly, this system induces EICs with similar penetrance, morphological and transcriptional defects compared to the previous one (Extended Data Fig. 6b-e; Extended Data Table 3).

Please provide more information for non-specialists in the figure legends to understand the graphs for Ext Data Fig 6b and enable interpretation. What are the numbers in the plots? What is a standardized residual?

The caption of the Fig. was revised to improve clarity and explain the different points highlighted here.

- Line 223 Constant ph-KD tumors were able to expand and invade a fraction of injected host flies at restrictive temperature (Figure 5b).

Rephrase sentence. Tumors do not invade host. Tumors invade distant sites/organs within host.

Thanks for pointing this out. We corrected the text accordingly, line 297: **“Constant *ph*-KD primary tumors grew in a high fraction of the injected host flies within 20 days of transplantation (Figure 6b)”**.

Discussion

- Line 251 On the other hand, preferential binding of JAK-STAT related TFs in the vicinity of irreversible genes coupled with weaker PRC1 binding at their promoter might foster transcription after transient perturbation of PcG, dampening their re-repression, and resulting in a self-sustaining aberrant cell state that stimulates tumor progression (Extended Data Fig. 8). Please rephrase to include JNK-related TFs, because this is really what kay, jra or Ets21C are.

We improved the manuscript to more precisely highlight the role of AP-1, line 248: **“On the other hand, recruitment of AP-1 and STAT92E at irreversible peaks could maintain**

irreversible genes in an activate state, potentially by maintaining chromatin open at their cis-regulatory regions”.

Reviewer Response References

- 1 Gong, S., Zhang, Y., Tian, A. & Deng, W. M. Tumor models in various *Drosophila* tissues. *WIREs Mech Dis* **13**, e1525 (2021). <https://doi.org:10.1002/wsbm.1525>
- 2 Leatherman, J. L. & Dinardo, S. Zfh-1 controls somatic stem cell self-renewal in the *Drosophila* testis and nonautonomously influences germline stem cell self-renewal. *Cell Stem Cell* **3**, 44-54 (2008). <https://doi.org:10.1016/j.stem.2008.05.001>
- 3 Postigo, A. A., Ward, E., Skeath, J. B. & Dean, D. C. zfh-1, the *Drosophila* homologue of ZEB, is a transcriptional repressor that regulates somatic myogenesis. *Mol Cell Biol* **19**, 7255-7263 (1999). <https://doi.org:10.1128/MCB.19.10.7255>
- 4 Classen, A. K., Bunker, B. D., Harvey, K. F., Vaccari, T. & Bilder, D. A tumor suppressor activity of *Drosophila* Polycomb genes mediated by JAK-STAT signaling. *Nat Genet* **41**, 1150-1155 (2009). <https://doi.org:10.1038/ng.445>
- 5 Bunker, B. D., Nellimoottil, T. T., Boileau, R. M., Classen, A. K. & Bilder, D. The transcriptional response to tumorigenic polarity loss in *Drosophila*. *Elife* **4** (2015). <https://doi.org:10.7554/eLife.03189>

Reviewer Reports on the First Revision:

Referees' comments:

Referee #1 (Remarks to the Author):

The authors have generated a larger body of additional work fully addressing the comments. They have carried out further genomic profiling and analysis to explore the role of genetic mutations. Additional data developing new driver lines and epigenetic profiling has provided insight into mechanism which has substantially strengthened the manuscript. A schematic showing the proposed multistep model of epigenetically initiated cancer could be an informative addition. This paper is a strong addition to the literature on cancer initiation and will be of significant interest to the community.

Referee #2 (Remarks to the Author):

The authors have performed an extensive amount of work and provided excellent responses addressing our comments to the manuscript. We believe that the authors convincingly have provided data showing that transient epigenetic changes can initiate cancer using *Drosophila* as a model system. These data are in line with and extend previous observations in human pediatric ependymoma, which is believed to develop through epigenetic changes in the cell of origin. The role of somatic mutations in epigenetic genes is firmly established in cancer, whereas transient epigenetic changes, i.e. without somatic changes are not. These type of changes as drivers of cancer are most likely developmentally restricted, however, this does not affect the impact of the submitted manuscript. Therefore, we recommend publication of the manuscript.

Referee #3 (Remarks to the Author):

I am generally happy with the extensive revision of the manuscript, both with respect to my concerns and the concerns raised by reviewers 1 and 2. The new data provided strengthens and sharpens the conclusion, and the reorganisation of the data helps the manuscript flow. I am still curious about the role of STAT as sufficient to induce these tumours (the experiments I asked for are only partially done and not included in the manuscript). However, I do agree with the authors that these experiments are somewhat beyond the scope and that the main focus of the current manuscript should be on epigenetically induced tumours, rather than on the details of permanent STAT/Zfh1 driven reprogramming of the transcriptional landscape.

Referee #3 (Remarks on code availability):

I cannot comment on coding tools.